# Implicit Geometry of Next-token Prediction:
# From Language Sparsity Patterns to Model Representations

**Yize Zhao**[†]**, Tina Behnia**[†]**, Vala Vakilian & Christos Thrampoulidis**
Department of Electrical & Computer Engineering
University of British Columbia
{zhaoyize,tina.behnia,vaalaa,cthrampo}@ece.ubc.ca

## Abstract

Next-token prediction (NTP) over large text corpora has become the go-to paradigm to train large language models. Yet, it remains unclear how NTP influences the mapping of linguistic patterns to geometric properties of the resulting model representations. We frame training of large language models as soft-label classification over sparse probabilistic label vectors, coupled with an analytical approximation that allows unrestricted generation of context embeddings. This approach links NTP training to rank-constrained, nuclear-norm regularized optimization in the logit domain, offering a framework for analyzing the geometry of word and context embeddings. In large embedding spaces, we find that NTP implicitly favors learning logits with a sparse plus low-rank structure. While the sparse component captures the co-occurrence frequency of context-word pairs, the orthogonal low-rank component, which becomes dominant as training progresses, depends solely on the sparsity pattern of the co-occurrence matrix. Consequently, when projected onto an appropriate subspace, representations of contexts that are followed by the same set of next-tokens collapse—a phenomenon we term subspace-collapse. We validate our theory on synthetic and small-scale real language datasets. Finally, we outline potential research directions aimed at deepening the understanding of NTP's influence on the learning of linguistic patterns and regularities.

## 1 Introduction

Next-token prediction (NTP) is the preferred training paradigm for state-of-the-art language models. The process, elegantly simple, uses a large training corpus to minimize, for each context $z_{<t} \in \mathcal{V}^{t-1}$ of $t-1$ preceding tokens, the cross-entropy (CE) loss between the model's predicted conditional probability distribution over potential next tokens from a vocabulary $\mathcal{V}$ and the one-hot encoded actual next token $z_t \in \mathcal{V}$. The model's conditional distribution is defined through a softmax map applied to logits $\ell_{<t}(W, \theta) = W h_\theta(z_{<t})$, which are generated by mapping *context embeddings* $h_\theta(z_{<t}) \in \mathbb{R}^d$—the neural network's $d$-dimensional representations of contexts $z_{<t}$—using a matrix $W \in \mathbb{R}^{|\mathcal{V}| \times d}$ of *word embeddings*.

Rooted in the foundational works of Shannon (1948) and inspired by the "co-occurrence statistics" and "distributional hypothesis" (Harris, 1954), the NTP paradigm suggests that a word's meaning is defined by its context. This principle underlies both classical vector-space models (Schütze et al., 2008) and neural language models (Turian et al., 2010; Bengio et al., 2000; Baroni et al., 2014; Bengio & Bengio, 2000; Mikolov et al., 2013b), propelling the development of today's sophisticated large language models (Radford et al., 2018; 2019; Brown et al., 2020). However, a fundamental question remains:

> *How does the NTP learning objective shape the relationship between*
> *language statistics and the geometry of model representations?*

This inquiry, which we term the *implicit geometry* of NTP—so named because the NTP objective does not explicitly impose any such relationship—explores how distances and angles within the neural network's $d$-dimensional representational space correlate with linguistic patterns at the end of training. We postulate that understanding this implicit

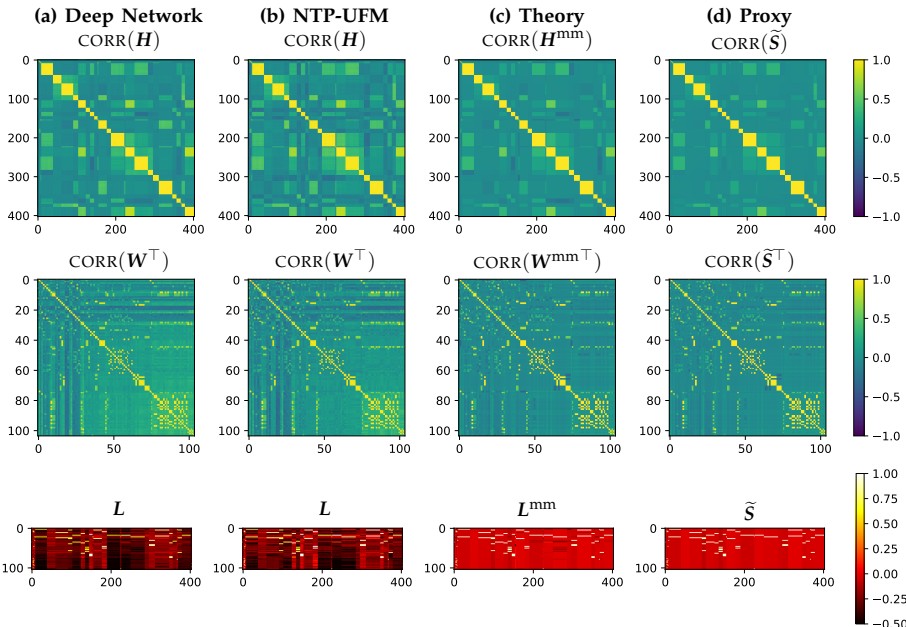

**Figure 1:** A 4-layer transformer (TF) trained on the `Simplified TinyStories` dataset. **(a)** Cosine similarity of TF's context and word embeddings $\mathrm{CORR}(\boldsymbol{H})$ and $\mathrm{CORR}(\boldsymbol{W}^\top)$ at the end of training (when NTP loss converges to its empirical entropy lower-bound). **(b)** $\mathrm{CORR}(\boldsymbol{H})$ and $\mathrm{CORR}(\boldsymbol{W}^\top)$ found by training the log-bilinear model (NTP-UFM) of Eq. (1) on the soft-labels $\boldsymbol{P}$ and support sets $\boldsymbol{S}$ of the original training set. **(c)** Geometry of context and word embeddings as specified by our analysis. $\boldsymbol{H}^{\mathrm{mm}}$ and $\boldsymbol{W}^{\mathrm{mm}}$ are determined by the right/left singular factors (Claim **(C2)**) of the low-rank component $\boldsymbol{L}^{\mathrm{mm}}$ of logits (Claim **(C1)**). **(d)** An easy to compute heuristic proxy for the embeddings' geometry based on the training support set $\boldsymbol{S}$ (Proxy **(P)**). Details in Secs. 1.2 and 5.

geometry is key to understanding functional principles of large language models, since NTP is used for training across diverse architectures, from LSTMs to transformers and state-space models. Specifically, exploring how optimization under NTP shapes representations of words and contexts, which empirically mirror complex human-like patterns, not only fosters scientific interest but could also enhance model interpretability and explainability. Further, revealing how implicit geometry correlates with language statistics could lead to refined training and inference methods, addressing challenges like statistical imbalances in language data. Conversely, understanding how state-of-the-art models internalize language to form representations might also enhance our grasp of language itself.

This paper develops an analytical framework to characterize the implicit geometry induced by NTP training on language datasets. Drawing inspiration from seminal studies on the geometry of deep model representations in image recognition (Papyan et al., 2020), our framework distinguishes itself by not concentrating on specific architectures such as transformers, which have been the focus of previous studies on language representations. Instead, we assume that the model has adequate representation capacity and undergoes effective optimization, making it possible to minimize the NTP loss to its entropy lower-bound. This approach isolates the influence of NTP itself—rather than architectural nuances—in shaping the implicit geometry of the language model. Our analysis also identifies the key role of the sparsity pattern in language statistics (Thrampoulidis, 2024). Concretely, we show that the recurrence of only a few tokens as next-tokens in particular contexts leads to an implicit bias towards a matrix of logits that develops a *sparse plus low-rank* structure during training. The sparse component of this matrix captures the probabilities of co-occurring words and contexts, while the dominant low-rank component encodes the sparsity pattern of the co-occurrence matrix. Overall, this framework introduces a novel perspective, markedly distinct from traditional analyses of word representations like those in the Word2Vec model (Mikolov et al., 2013b; Levy & Goldberg, 2014).

## 1.1 Methodology

Our methodology integrates three foundational modeling concepts as follows:

Firstly, we frame NTP as soft-label classification with CE loss applied to *sparse* probabilistic label vectors (Thrampoulidis, 2024). This isolates *m distinct* contexts, which could be repeated multiple times throughout the training corpus, and assigns to each a sparse $V = |\mathcal{V}|$-dimensional conditional-probability label vector $\hat{p}_j$, reflecting the frequency of each token following context $j \in [m]$. The sparsity of $\hat{p}_j$ indicates that certain tokens, which we refer to as *off-support* tokens for the specific $j$-th context, never follow this context.

The second concept facilitates a tractable analysis of context embeddings by assuming expressive (enough) neural networks can produce unconstrained embeddings $h_j \in \mathbb{R}^d$, independent of the architecture's specific complexities (Yang et al., 2017; Mixon et al., 2020). This redefines NTP as a minimization, over word and context embedding matrices $W$ and $H$, of the NTP loss across a training set $\mathcal{T}$ determined by the matrix of conditional probabilities $P = P(\mathcal{T}) = [\hat{p}_1, \hat{p}_2, \ldots, \hat{p}_m] \in [0, 1]^{V \times m}$, leading to the following log-bilinear model:

$$\min_{W \in \mathbb{R}^{V \times d}, H \in \mathbb{R}^{d \times m}} \mathcal{L}_{\text{NTP}}(WH; P) . \qquad (1)$$

This way, our goal to study word-word, context-context, and word-context geometric relationships becomes that of characterizing the Gram matrices $G_W = WW^\top$ and $G_H = H^\top H$, as well as the logit matrix $L = WH$ at the minimizers of Eq. (1). This task is complicated by the non-convex nature of the minimization and the sparsity of the probabilistic label vectors. Specifically, as we show, the sparsity of $P$ may lead to multiple minimizers, potentially making geometric characterization ambiguous.

To address this, we leverage a third concept: focusing on specific minimizers identified through the *regularization path* (Rosset et al., 2003). This entails following the solution trajectory of the empirical risk minimization in Eq. (1) as an additive ridge regularization for $W$ and $H$ diminishes to zero. For the purpose of comparing the analysis outcomes to our numerical evaluations, we interpret the prediction obtained from the regularization path analysis as a proxy for the solution found by gradient-based optimization when the NTP loss approaches its empirical entropy lower bound.

## 1.2 Summary of findings

Building on the above methodology, our analytical framework leads to the following results:

**Formulation in logit space.** Sec. 3 presents an equivalent formulation of the NTP objective $\mathcal{L}_{\text{NTP}}(WH; P) + \lambda\|W\|^2 + \lambda\|H\|^2$ into the logit space, given in terms of a rank-constrained and nuclear-norm regularized minimization. From this, word and context matrices can be obtained through matrix factorization of logits $L$.

**Logit Convergence.** Focusing on $d \geq V$[1], in Sec. 4, we demonstrate that the logit matrix $L_\lambda$ for regularization $\lambda \to 0$ (which is proxy for training iteration $k \to \infty$), behaves for some $R(\lambda) \to \infty$ as shown by the following claim:

**(C1) Logits' sparse plus low-rank decomposition:** As $\lambda \to 0$,

$$L_\lambda \quad \approx \quad \underbrace{L^{\text{in}}}_{\substack{\textbf{sparse} \text{ component:} \\ \text{sets in-support token probabilities}}} \quad + \quad R(\lambda) \cdot \underbrace{L^{\text{mm}}}_{\substack{\textbf{low-rank} \text{ component:} \\ \text{separates in- from off-support tokens}}} , \qquad (2)$$

where the orthogonal components $L^{\text{in}}$ and $L^{\text{mm}}$ have distinct operational roles.

$L^{\text{in}}$ encodes information regarding frequencies of in-support tokens, inheriting the sparsity of the data matrix $P$. In contrast, $L^{\text{mm}}$ depends only on the sparsity pattern $S \in \{0, 1\}^{V \times m}$ of $P$. Thus, its value is guided by both the observed 'company' of each context (in-support tokens), but not their frequency, and, by the 'company' it lacks (off-support tokens). The role of this second component is to maximize the logit margin between in- and off-support tokens, measured with respect to the low-rank promoting nuclear norm. This decomposition in (2) also shows the logits become unbounded in norm when projected on the subspace of $L^{\text{mm}}$.

---

[1]This does *not* constrain $d$ relative to the number of distinct contexts $m$, which could be significantly larger than the vocabulary size $V$. Also, despite large embedding dimensions, our results still reveal how the geometry of learned word embeddings encodes fine-grained similarities between words as materialized in the patterns of language data.

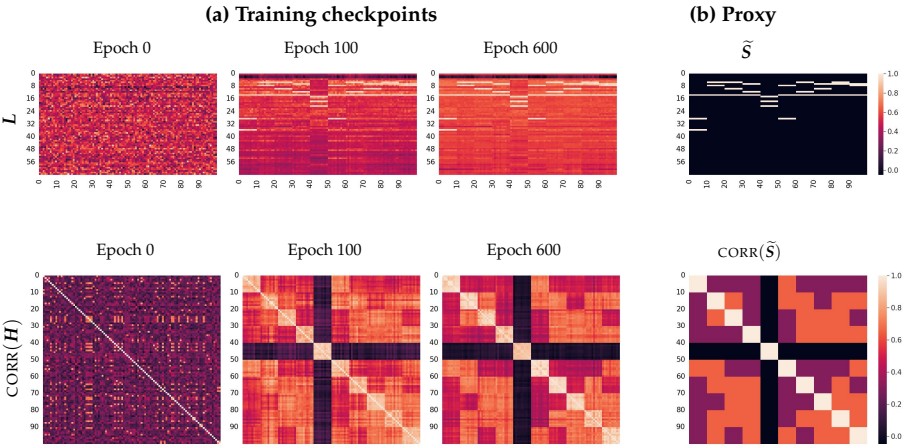

**Figure 2:** Similar to Fig. 1, this time on a 12-layer TF trained on a subset of 100 stories from the `TinyStories` dataset. Here, computing the theoretical prediction $H^{\mathrm{mm}}$ is computationally expensive. Thus, we compare the embeddings geometry with the Proxy **(P)**. Details in Sec. 5.

**Context and word embeddings' geometry.** Similar to logits, word/context embeddings $W$ and $H$ also grow in norm as $\lambda \to 0$. This occurs in a way that simultaneously guarantees the resulting logit matrix abides by Claim **(C1)**, which leads to the following additional claims:

**(C2) Directional convergence:** Word and context embedding matrices converge *directionally* to matrices $W^{\mathrm{mm}} := U\Sigma^{1/2}R$, $H^{\mathrm{mm}} := R^\top \Sigma^{1/2} V^\top$, where $R$ is a rotation matrix and $U\Sigma V^\top$ is the singular value decomposition (SVD) of the low-rank max-margin component $L^{\mathrm{mm}}$ in Eq. (2). Concretely, letting $\overline{A} := A/\|A\|$ for any matrix $A$:

$$\overline{G_W} \to U\overline{\Sigma}U^\top \quad \text{and} \quad \overline{G_H} \to V\overline{\Sigma}V^\top. \tag{3}$$

**(C3) Subspace collapse:** Since $L^{\mathrm{mm}}$ depends only on the sparsity patterns of next-token distributions (not their frequencies), a consequence of **(C2)** is a property termed *subspace collapse*: embeddings $h_j$, $h_{j'}$ of contexts $j \neq j' \in [m]$ that are followed by the same set of next-tokens (although their frequencies may differ), converge to the same limiting direction, i.e., for all $j, j' \in [m]$

$$\mathrm{support}(\hat{p}_j) = \mathrm{support}(\hat{p}_{j'}) \implies \cos(h_j, h_{j'}) = h_j^\top h_{j'} / (\|h_j\| \|h_{j'}\|) \to 1. \tag{4}$$

**(C4) Soft-label interpolation:** When projected on the subspace of in-support tokens, the logits $WH$ interpolate the corresponding soft-labels ensuring that the NTP loss reaches the entropy lower bound. Concretely, for all $j \in [m]$,

$$(w_z - w_{z'})^\top h_j = \log(\hat{p}_{j,z}/\hat{p}_{j,z'}), \quad \forall z, z' \in \mathrm{support}(\hat{p}_j). \tag{5}$$

Since the subspace of in-support tokens is orthogonal to the subspace of $L^{\mathrm{mm}}$, the subspace collapse **(C3)** does *not* prevent logits $w_z^\top h_j$ and $w_z^\top h_{j'}$ of in-support tokens $z \in \mathrm{support}(\hat{p}_j) = \mathrm{support}(\hat{p}_{j'})$ to interpolate (in the sense of Eq. (5)) potentially different conditional probabilities (soft-labels) $\hat{p}_{j,z}$ and $\hat{p}_{j',z}$, respectively.

**Experiments.** Sec. 5 presents experiments on controlled settings that validate our findings. An example is shown in Fig. 1: The experiment involves training a 4-layer transformer (TF) on a training set extracted and curated from the `TinyStories` dataset (Eldan & Li, 2023), with the following characteristics: a vocabulary size $V = 104$ and a total number of contexts $n \approx 3050$, out of which $m \approx 400$ are distinct. We choose TF embedding dimension $d = 128$ and train until the empirical entropy lower bound is reached within an order of $10^{-4}$. The leftmost Panel (a) depicts the geometry of the context embeddings $H$ (Top), word embeddings $W$ (Middle), and logits $L$ (Bottom) learned by the TF at the end of training. Panel (b) compares these with the same quantities learned by the log-bilinear model in Eq. (1) evaluated on the same training dataset. The apparent resemblance of the patterns validates that the proposed log-bilinear analysis model is a good proxy for the

TF model. Panel (c) compares the TF output to our analytical predictions: Comparing TF's context/word embeddings and logits to $H^{\mathrm{mm}}/W^{\mathrm{mm}}$ and $L^{\mathrm{mm}}$ confirms Claims **(C2)** and **(C1)**. The bright yellow regions in the Gram matrices of embeddings help visualize the subspace collapse Claim **(C3)**. See Sec. 5 for additional details and verification of Claim **(C4)**.

Finding the analytical prediction **(C2)**, illustrated in Panel (c), requires solving a semi-definite program for $L^{\mathrm{mm}}$ and computing its SVD. In large scales, this can be computationally prohibitive, motivating the following heuristic proxy for the embedding geometry.

> **(P) Empirical proxy for directional convergence:** Let $\widetilde{S} := (\mathbb{I}_V - \frac{1}{V}\mathbb{1}_V\mathbb{1}_V^\top)S$ be the (column-wise) centered support-set matrix. The directional component of word and context embedding matrices can be well-approximated by $\widetilde{S}$ as follows:
>
> $$W^{\mathrm{mm}}W^{\mathrm{mm}\top} \approx \widetilde{S}\widetilde{S}^\top \quad \text{and} \quad H^{\mathrm{mm}\top}H^{\mathrm{mm}} \approx \widetilde{S}^\top\widetilde{S}. \tag{6}$$

Panel (d) of Fig. 1 suggests proxy **(P)** as a good enough approximator of the embeddings structure that simply only depends on the sparsity pattern $S$ of the next-tokens' conditional probability matrix.

A second example supporting proxy **(P)** is shown in Fig. 2: Here, we train a larger TF on a subset of the `TinyStories`, where computing $L^{\mathrm{mm}}$ and its SVD is expensive. Instead, we observe that the structure of the support set closely captures the structure of the learned embeddings: Along the diagonal blocks, where contexts with the exact same support sets are situated, the context embeddings align closely (subspace collapse in Claim **(C3)**). Conversely, when support sets have zero intersection (dark entries in Panel (b)), context embeddings exhibit low correlation. This finding is consistent with classical intuitions of the "distributional assumption" of words/contexts and is formalized by our framework via the implicit bias of NTP to promote low-rank logits subject to margin conditions for in/off-support tokens.

## 2 Related work

We pinpoint three main related research areas; see App. A for more in-depth discussion.

First, our research conceptually mirrors the seminal work by Levy & Goldberg (2014) who framed the Skip-Gram with Negative Sampling (SGNS) training objective of Word2Vec (Mikolov et al., 2013b;a) as weighted matrix factorization. Specifically for large $d$, they demonstrated that SGNS implicitly factorizes the pointwise mutual information (PMI) matrix. Their analysis relies on the fact that Word2Vec architecture already is a log-biliniear model, while the log-bilinear model in (1) is only used by us as an analytical proxy for more complex architectures. Also, different to them, we focus on NTP, which employs softmax instead of sigmoids as in SGNS. (However, our analysis applies also to SGNS; see App. A.4). More importantly, we contribute a fresh perspective on this line of inquiry by: (i) confronting the sparsity of probabilistic labels head-on, identifying that it leads to diverging embeddings, a scenario where setting the loss gradient to zero, as in Levy & Goldberg (2014), is infeasible; (ii) examining embeddings through the lens of the regularization path—a surrogate for gradient descent optimization— which unveils that embeddings emerge from the factorization of $L^{\mathrm{in}} + RL^{\mathrm{mm}}$, where $L^{\mathrm{in}}$ captures frequencies akin to the PMI, while the directional component $L^{\mathrm{mm}}$, becoming dominant as weights diverge, reflects the explicit sparsity patterns of the context-word co-occurrences. We envision this fresh perspective could similarly motivate further research leveraging the geometric insights of embeddings to uncover linguistic phenomena such as the linear relationships underlying word analogies (Mikolov et al., 2013a; Pennington et al., 2014).

Second, our exploration of NTP's regularization path is inspired by the *implicit bias/regularization* research on the preferred solutions of optimizers like GD in overparameterized systems. Ji & Telgarsky (2020) showed that GD's trajectory in linear one-hot encoding models aligns with the regularization path, providing a lens for examining GD dynamics. Thrampoulidis (2024) recently extended this analysis to NTP, framing it as sparse soft-label classification. By lifting their assumption of fixed context embeddings, we delve into the more complex non-convex domain, establishing explicit connections between the implicit geometry and linguistic patterns, which we also verify empirically.

Finally, our research intersects with the study of *neural-collapse (NC)*, which delves into the geometry of last-layer features and weights in deep networks trained in the interpolating regime (Papyan et al., 2020), utilizing the unconstrained festures model (UFM) for analysis (Mixon et al., 2020; Fang et al., 2021; Zhu et al., 2021). Our work is particularly aligned with Thrampoulidis et al. (2022): We also examine the UFM's regularization path but within the NTP framework, extending their one-hot classification findings, which can be seen as special cases of ours. Additionally, our findings have some parallels with the multilabel NC geometry explored in Li et al. (2023); Fisher et al. (2024). However, our setting is more general raising stringent assumptions on the label distribution. For detailed comparison, see App. A.

## 3 Formulation

**Notations.** Throughout, lowercase and uppercase bold letters (e.g., $\boldsymbol{a}$ and $\boldsymbol{A}$) represent vectors and matrices, respectively. We use $\overline{\boldsymbol{a}}$ and $\overline{\boldsymbol{A}}$ to denote vectors/matrices normalized by their Euclidean norm. We denote $\boldsymbol{A}[i,j]$ the $(i,j)$-th entry of matrix $\boldsymbol{A}$ and $\boldsymbol{a}_j$ its $j$-th column. We let $\mathcal{R}(\boldsymbol{A})$ and $\mathcal{N}(\boldsymbol{A})$ its range-space (aka column space) and null-space, respectively. $\langle \cdot, \cdot \rangle$ and $\|\cdot\|$ denote Euclidean inner product and norm, respectively. We use $\|\cdot\|_*$ to denote the nuclear-norm (i.e. sum of singular values). $\mathbb{I}_V$ represents the identity matrix of size $V$ and $\mathbb{1}_V$, the all ones vector of size $V \times 1$ (subscripts are removed when clear from context). $\Delta^{V-1}$ denotes the $V$-dimensional unit simplex and $\mathsf{S}(\cdot) : \mathbb{R}^V \to \Delta^{V-1}$ the softmax map: $\mathsf{S}(\boldsymbol{a}) = [\mathsf{S}_1(\boldsymbol{a}), \ldots, \mathsf{S}_V(\boldsymbol{a})]^\top$ with $\mathsf{S}_v(\boldsymbol{a}) = \exp(\boldsymbol{e}_v^\top \boldsymbol{a}) / \sum_{v' \in [V]} \exp(\boldsymbol{e}_{v'}^\top \boldsymbol{a})$, where $\boldsymbol{e}_v$ is the $v$-th standard basis vector in $\mathbb{R}^V$. We also denote $\widetilde{\boldsymbol{e}}_j$ the $j$-th standard basis vector in $\mathbb{R}^m$. All logarithms are natural logarithms (base $e$).

### 3.1 NTP objective as soft-label classification

We let $\mathcal{V} = [V] := \{1, \ldots, V\}$ represent a finite vocabulary of tokens (we use the terms 'word' and 'token' interchangeably). We denote by $\boldsymbol{z}_{1:t} = (z_1, \ldots, z_t)$ a sequence of $t$ tokens $z_t \in \mathcal{V}$ and focus, for simplicity, on prediction of the last $T$-th token $z := z_T$ given context $\boldsymbol{x} := \boldsymbol{z}_{1:T-1}$. For this, we assume access to a training set consisting of $n$ sequences $\mathcal{T}_n := \{(\boldsymbol{x}_i, z_i)\}_{i \in [n]}$, such that $\boldsymbol{x}_i \in \mathcal{X} := \mathcal{V}^{T-1}$ and $z_i \in \mathcal{V}$. This is used to train model $f_{\boldsymbol{\theta}'} : \mathcal{X} \to \mathcal{V}, f_{\boldsymbol{\theta}'}(\boldsymbol{x}) = \boldsymbol{W} \boldsymbol{h}_{\boldsymbol{\theta}}(\boldsymbol{x})$ parameterized by $\boldsymbol{\theta}' = \{\boldsymbol{W}, \boldsymbol{\theta}\}$, where $\boldsymbol{W} \in \mathbb{R}^{V \times d}$ is a decoding matrix and $\boldsymbol{\theta}$ parameterizes a map $\boldsymbol{h}_{\boldsymbol{\theta}} : \mathcal{X} \to \mathbb{R}^d$ from contexts to $d$-dimensional embeddings. We impose no restrictions on the specific form of the embedding map, which may, for example, be produced by an MLP, an LSTM, or a TF. We refer to row $\boldsymbol{w}_v, v \in \mathcal{V}$ of $\boldsymbol{W}$ as **word embedding** of token $v$ and $\boldsymbol{h}_{\boldsymbol{\theta}}(\boldsymbol{x})$ as **context embedding** of context $\boldsymbol{x}$. The model is found by minimizing the empirical CE loss $\mathrm{CE}(\boldsymbol{\theta}') = \frac{1}{n} \sum_{i \in [n]} - \log\left(\mathsf{S}_{z_i}(f_{\boldsymbol{\theta}'}(\boldsymbol{x}))\right)$.

Following Thrampoulidis (2024), we reframe the NTP training objective as classification over $m \leq n$ *distinct* contexts, each associated with a *sparse* probabilistic label vector $\hat{\boldsymbol{p}}_j \in \Delta^{V-1}$. Concretely, we denote $\bar{\boldsymbol{x}}_1, \ldots, \bar{\boldsymbol{x}}_m$ the $m \leq n$ *distinct* contexts among the (large number of) total $n$ contexts $\mathcal{T}_n$. Also, we let $\hat{\pi}_j = \frac{1}{n} \sum_{i \in [n]} \mathbb{1}[\boldsymbol{x}_i = \bar{\boldsymbol{x}}_j]$ denote the empirical probability of distinct context $\bar{\boldsymbol{x}}_j$. Accordingly, let $\hat{\boldsymbol{p}}_j \in \Delta^{V-1}$ denote the probability vector of conditional next-token distribution, i.e., for all $z \in \mathcal{V}$: $\hat{p}_{j,z} := \frac{1}{n} \sum_{i \in [n]: \boldsymbol{x}_i = \bar{\boldsymbol{x}}_j} \mathbb{1}[z_i = z]$, $j \in [m]$. In words, $n \cdot \hat{\pi}_j \cdot \hat{p}_{j,z}$ is the number of occurrences of token $z$ as a follow-up to context $\bar{\boldsymbol{x}}_j$. Define the support sets of these probability vectors as $\mathcal{S}_j := \{z \in \mathcal{V} \mid \hat{p}_{j,z} > 0\}$ and let $S_j := |\mathcal{S}_j|$.

It is also convenient to define the probability matrix $\boldsymbol{P} = [\hat{\boldsymbol{p}}_1, \ldots, \hat{\boldsymbol{p}}_m]$ and its corresponding **support matrix** $\boldsymbol{S} \in \{0,1\}^{V \times m}$, such that $\boldsymbol{S}[z, j] = 1$, *iff* $z \in \mathcal{S}_j$. In typical scenarios, $S_j < V$; thus, $\boldsymbol{P}$ and $\boldsymbol{S}$ are *sparse* matrices. For given context index $j \in [m]$, we say word $z$ is **in-support token** if $\boldsymbol{S}[z, j] = 1$ (eqv. $z \in \mathcal{S}_j$); otherwise, we say $z$ is **off-support token**.

With the above notation, the training loss becomes (Thrampoulidis, 2024),

$$\mathrm{CE}(\boldsymbol{\theta}') = - \sum_{j \in [m]} \hat{\pi}_j \sum_{z \in \mathcal{V}} \hat{p}_{j,z} \log\left(\mathsf{S}_z(\boldsymbol{W} \boldsymbol{h}_{\boldsymbol{\theta}}(\bar{\boldsymbol{x}}_j))\right) . \quad (7)$$

This is lower bounded by the empirical $T$-gram entropy (referred to hereafter as entropy) of the data (Shannon, 1948), i.e., for all $\boldsymbol{\theta}'$: $\mathrm{CE}(\boldsymbol{\theta}') \geq \mathcal{H} := - \sum_{j \in [m]} \hat{\pi}_j \sum_{z \in \mathcal{V}} \hat{p}_{j,z} \log\left(\hat{p}_{j,z}\right)$.

### 3.2 Unconstrained features model for NTP training

To gain insights into the geometry of solutions to CE minimization in Eq. (7), we assume sufficient model expressivity, allowing us to optimize embeddings freely, instead of abiding by their architecture-specific parameterization. We formalize this concept below.

**Definition 1** (NTP-UFM). *The unconstrained features model (UFM) for NTP training over training set $\mathcal{T}_m = \{\hat{\pi}_j, \hat{p}_j\}_{j\in[m]}$ refers to the following log-bilinear optimization problem:*

$$\min_{W,H} \quad \text{CE}(WH) + \frac{\lambda}{2}\|W\|^2 + \frac{\lambda}{2}\|H\|^2, \qquad \text{(NTP-UFM)}$$

*where minimization is over word and context embedding matrices $W \in \mathbb{R}^{V\times d}$ and $H := [h_1, \ldots, h_m] \in \mathbb{R}^{d\times m}$, and $\text{CE}(WH) := -\sum_{j\in[m]} \hat{\pi}_j \sum_{z\in\mathcal{S}_j} \hat{p}_{j,z} \log(\text{S}_z(Wh_j))$ .*

The key difference of NTP-UFM compared to Eq. (7) is that embeddings $h_\theta(\bar{x}_j)$ are now optimized unconstrainedly through free variables $h_j$. Furthermore, we have introduced ridge regularization. Although our focus is on understanding the geometry arising from minimizing the unregularized CE objective in Eq. (7), we utilize ridge regularization as a proxy to examine the behavior of gradient descent.[2] Specifically, we use the vanishing-regularization solution of NTP-UFM as a proxy for the parameters $(W_k, H_k)$ learned in large iteration $k \to \infty$ through GD on the unregularized objective (7).

### 3.3 Reformulation in terms of logits

We now introduce a reformulation of NTP-UFM utilizing the logit matrix $\mathbf{L} = \mathbf{WH}$. This is grounded on the well-known fact concerning the nuclear norm of a matrix (Srebro et al., 2004; Fazel, 2002): $\|\mathbf{L}\|_* = \min_{L=WH} \frac{1}{2}\|W\|^2 + \frac{1}{2}\|H\|^2$ .

**Lemma 1.** *Denote logit matrix $\mathbf{L} = [\ell_1, \ldots, \ell_m] \in \mathbb{R}^{V\times m}$ and let $\mathbf{L}_\lambda$ be a minimizer of*

$$\min_{L:\text{rank}(L)\leq d} \left\{ -\sum_{j\in[m]} \hat{\pi}_j \sum_{z\in\mathcal{S}_j} \hat{p}_{j,z} \log(\text{S}_z(\ell_j)) + \lambda\|L\|_* \right\}, \qquad (8)$$

*with SVD $\mathbf{L}_\lambda = \mathbf{U}\boldsymbol{\Sigma}\mathbf{V}^\top$, where $\mathbf{U} \in \mathbb{R}^{V\times r}, \boldsymbol{\Sigma} \in \mathbb{R}^{r\times r}, \mathbf{V} \in \mathbb{R}^{m\times r}$ and $r = \text{rank}(\mathbf{L}_\lambda) \leq d$. Then:*

*(i) The optimal cost of NTP-UFM is the same as that of Eq. (8).*

*(ii) $(\mathbf{W}_\lambda, \mathbf{H}_\lambda)$ minimizes NTP-UFM if and only if there exists a minimizer $\mathbf{L}_\lambda$ of Eq. (8) such that: $\mathbf{W}_\lambda \mathbf{H}_\lambda^\top = \mathbf{L}_\lambda, \ \mathbf{W}_\lambda \mathbf{W}_\lambda^\top = \mathbf{U}\boldsymbol{\Sigma}\mathbf{U}^\top$, and $\mathbf{H}_\lambda^\top \mathbf{H}_\lambda = \mathbf{V}\boldsymbol{\Sigma}\mathbf{V}^\top$ .*

The constraint $\text{rank}(L) \leq d$ in Eq. (8) ensures the logit matrix can be factorized as $WH$ with inner factor dimension being $d$. Thus, in general, Eq. (8) is non-convex.

## 4 Analysis of unconstrained features model for NTP training

This section analyzes the regularization path of Eq. (8) and, subsequently, of NTP-UFM. Throughout, we assume $d \geq V$, in which case the constraint $\text{rank}(L) \leq d$ becomes redundant. All proofs are deferred to the appendix and numerical evaluations to App. D.1.

### 4.1 NTP-SVM logits

The following nuclear norm-minimization problem plays a key role in our results.

**Definition 2.** *Define the NTP-SVM logit matrix $\mathbf{L}^{\text{mm}}$ as solution to the following optimization:*

$$\mathbf{L}^{\text{mm}} \in \arg\min_{L\in\mathbb{R}^{V\times m}} \|L\|_* \qquad \text{(NTP-SVM}_\star\text{)}$$

$$\text{subj. to} \ \ L[z,j] - L[z',j] = 0, \ \ \forall j \in [m], z \neq z' \in \mathcal{S}_j, \qquad (9)$$

$$L[z,j] - L[v,j] \geq 1, \ \ \forall j \in [m], z \in \mathcal{S}_j, v \notin \mathcal{S}_j . \qquad (10)$$

---

[2]The equivalence between GD and regularization paths has been rigorously established in certain convex settings (Ji et al., 2020), but it may not hold generally. Our setting is non-convex, so we use this equivalence as a heuristic proxy; see future work discussion in Sec. 6.

The optimization in NTP-SVM$_\star$ is a semidefinite program (SDP) and always feasible, as shown by the straightforward feasibility of the centered support matrix $\widetilde{S} = (\mathbb{I}_V - \frac{1}{V}\mathbb{1}_V\mathbb{1}_V^\top)S$. Its solution $L^{\mathrm{mm}}$ depends solely on the support matrix $S$ of the data. Interestingly, we prove in App. C that $\widetilde{S}$ solves NTP-SVM$_\star$ under symmetry assumptions on $S$. More generally, our empirical results show that $\widetilde{S}$ closely approximates the true solution $L^{\mathrm{mm}}$, supporting the use of Proxy **(P)**; see App. C.3.

Finally, when $S_j = 1, \forall j \in [m]$, then Eq. (9) vanishes and Eq. (10) lower bounds the margin between the correct token and the rest. This constraint appears in the classical hard-margin SVM, which explains our naming. Instead, the NTP setting results in additional equality constraints and minimizes nuclear-norm rather than Euclidean norm.

### 4.2 Regularization-path analysis of logits

We now characterize the solutions of (8) when regularization vanishes and $d \geq V$.

**Theorem 1.** *Suppose $d \geq V$. In the limit of vanishing regularization, the solution $L_\lambda$ of optimization (8) diverges in norm and converges in direction to $L^{\mathrm{mm}}$. Formally, $\lim_{\lambda \to 0} \|L_\lambda\| = +\infty$, and, there exists minimizer $L^{\mathrm{mm}}$ of NTP-SVM$_\star$ such that $\lim_{\lambda \to 0} \left\| \frac{L_\lambda}{\|L_\lambda\|_*} - \frac{L^{\mathrm{mm}}}{\|L^{\mathrm{mm}}\|_*} \right\| = 0$.*

Thus, with vanishing regularization, NTP-SVM$_\star$ captures the structure of the logits. By Lemma 1, it also captures the structure of $W$ and $H$. We explore this further in Sec. 4.3.

Now, recall that NTP-SVM$_\star$ only depends on the support matrix of the training data, raising the questions: How do the probabilities $\hat{p}_{j,z}$ influence the learned logit matrix and what implications this has for the loss? The theorem below complements Thm. 1 and addresses these questions. To state the result, define the following matrix subspace, as the span of rank-one matrices each being zero except entries $(z, j)$ and $(z', j)$, which equal 1 and $-1$ respectively for all context indices $j \in [m]$ and in-support word indices $z, z' \in \mathcal{S}_j$:

$$\mathcal{F} = \mathcal{F}(S) = \mathrm{span}\left(\left\{ (e_z - e_{z'})\widetilde{e}_j^\top \ : \ z \neq z' \in \mathcal{S}_j, j \in [m] \right\}\right) \subset \mathbb{R}^{V \times m}.$$

Note for any $L$, the projection $\mathcal{P}_\mathcal{F}(L)$ onto $\mathcal{F}$ is sparse with support matching $S$.

**Theorem 2.** *Under the setting of Theorem 1, the loss approaches its lower bound $\lim_{\lambda \to 0} \mathrm{CE}(L_\lambda) = \mathcal{H}$. Additionally, $\lim_{\lambda \to 0} \mathcal{P}_\mathcal{F}(L_\lambda) = L^{\mathrm{in}}$, where $L^{\mathrm{in}} \in \mathcal{F}$ is the unique $L \in \mathcal{F}$ that guarantees differences of logits equal log-odds, i.e., $L[z, j] - L[z', j] = \log(\hat{p}_{j,z}/\hat{p}_{j,z'}), \ \forall z \neq z' \in \mathcal{S}_j, j \in [m]$.*

The loss reaches the lower-bound $\mathcal{H}$ because on the subspace $\mathcal{F}$, $L_\lambda$ satisfies the log-odds linear equations of the theorem. This gives rise to Claim **(C4)**: the model outputs the correct probabilities for in-support tokens. There is no contradiction of this claim to Thm. 1 because $L^{\mathrm{mm}}$ belongs to the orthogonal complement $\mathcal{F}_\perp$ (which follows from Eq. (9); see App. B.3.1).

Combining the two theorems, $L_\lambda$ converges to $L^{\mathrm{in}}$ in $\mathcal{F}$ and diverges in $\mathcal{F}_\perp$, where it converges directionally to $L^{\mathrm{mm}}$. This justifies Claim **(C1)**.

### 4.3 Geometry of embeddings

In view of Lemma 1 we can obtain the embeddings by factorizing the logit matrix. Since logits diverge, the same is true for the embeddings, which also converge directionally.

**Corollary 1.** *Denote $(W_\lambda, H_\lambda)$ any minimizer of NTP-UFM. Consider the SVD $L^{\mathrm{mm}} = U\Sigma V^\top$. Then, using $\overline{\cdot}$ notation to denote normalized quantities, in the limit of vanishing regularization $\overline{W_\lambda W_\lambda^\top} \to U\overline{\Sigma}U^\top =: \overline{G}_W^{\mathrm{mm}}, \overline{H_\lambda^\top H_\lambda} \to V\overline{\Sigma}V^\top =: \overline{G}_H^{\mathrm{mm}}$ and $\overline{W_\lambda H_\lambda} \to U\overline{\Sigma}V^\top$.*

This corollary supports Claim **(C2)**. Another direct consequence of it is Claim **(C3)**: embeddings of contexts whose support sets are identical asymptotically collapse to the same embedding as $\lambda \to 0$. We formalize this in Proposition 2 in App. C.1.

## 5  Experiments

We empirically validate our analysis on text data. We start with two small-scale synthetic datasets to examine Claims **(C1)-(C4)**. We then experiment with a larger-scale dataset, where SVD calculation are computationaly expensive, we examine the Proxy **(P)**.

**Datasets.** For a complete description of the datasets, see App. D.2.

Synthetic. We manually create simple (context, next-token) pairs, with context length of size $T - 1 = 5$. The dataset is constructed so that all support-sets satisfy $S_j = 3$.

Simplified TinyStories. For a more realistic but still controlled setup, we curate the dataset from the TinyStories corpus: We derive contexts $\bar{x}_j$ and support sets $\mathcal{S}_j$ by choosing the most frequent word-level contexts with length $T - 1 = 5$.

TinyStories. For more standard data, we use 100 stories sampled from TinyStories. Unlike the other two datasets, we do not sample over the contexts and support sets.

**Models.** We train decoder-only TF with 4 layers and $d = 128$ for Synthetic and Simplified TinyStories, and 12 layers and $d = 256$ for TinyStories. For training details, see App. D.2.

We verify two claims made in Sec. 4: 1) We assess whether numerically optimizing both the TF and NTP-UFM in each setup yields the same embedding geometries, 2) We evaluate the global solution of NTP-UFM, as specified by Thm. 1 and Cor. 1, and check whether it respects the same implicit geometry as the TF and the NTP-UFM. For the first one, we train NTP-UFM directly on the label distribution of each dataset; this is computationally feasible only for the first two datasets. For the second claim, we find the global solution $L^{\mathrm{mm}}$ of NTP-SVM$_\star$ on each dataset using CVXPY (Diamond & Boyd, 2016) and $H^{\mathrm{mm}}$, $W^{\mathrm{mm}}$ using Cor. 1. If not computationally possible, we only use the Proxy **(P)** as reference.

**Metrics.** We define the following metrics for verifying our theoretical results:

*Visualization of embeddings' geometry.* To visualize the geometry of the word and context embeddings, we plot the heatmap of the normalized Gram matrix $\mathrm{CORR}(W^\top)$ and $\mathrm{CORR}(H)$, where for matrix $X$, $\mathrm{CORR}(X)$ is a matrix whose $(i, j)$-th entry is the cosine similarity between the respective columns of $X$, i.e., $\left[\mathrm{CORR}(X)\right]_{i,j} = x_i^\top x_j / (\|x_i\| \|x_j\|)$. To check Proxy **(P)**, we compare the context and word embeddings with $\mathrm{CORR}(\widetilde{S})$ and $\mathrm{CORR}(\widetilde{S}^\top)$.

*Quantifying geometric similarity.* To measure the structural similarity between $W$, $H$ and $\widetilde{S}^\top$, $\widetilde{S}$, we use $\mathrm{SIM}(X, Y) = (\sigma_{XY} + \epsilon) / (\sigma_X \sigma_Y + \epsilon)$, as the correlation matrix between two matrices.[3] Here, $\sigma_{XY}$ is the covariance, and $\sigma_X$ and $\sigma_Y$ are the standard deviations of $X$ and $Y$, respectively, and $\epsilon$ is a small constant for stable division. A value of $\mathrm{SIM}(X, Y) = 1$ indicates perfect structural correlation. To simplify the presentation, and with some abuse of notation, we denote the similarity metric for context and word embeddings as follows,

$$\mathrm{SIM}(H, \widetilde{S}) := \mathrm{SIM}(\mathrm{CORR}(H), \mathrm{CORR}(S)), \quad \mathrm{SIM}(W^\top, \widetilde{S}^\top) := \mathrm{SIM}(\mathrm{CORR}(W^\top), \mathrm{CORR}(S^\top)).$$

*Recovery of soft-labels.* To verify Thm. 2, we measure the distance $\|\mathcal{P}_{\mathcal{F}}(L_k) - L^{\mathrm{in}}\|$ of $L^{\mathrm{in}}$ from the projection of $L_k$ onto the subspace $\mathcal{F}$.

**Results.** In Sec. 1.1, we discussed Fig. 1, where we compared the logits, word and context embeddings trained on Simplified TinyStories by (a) TF, (b) NTP-UFM, and those predicted by (c) Claims **(C1)-(C3)**, and (d) Proxy **(P)**. Similar observations for the TinyStories and Synthetic datasets are shown in Figs. 2 and 10. For visualization details, see App. D.2.

Next, we discuss Fig. 3 that tracks our metrics during training for the Synthetic and Simplified TinyStories datasets. Fig. 3-(a) confirms that the loss is close to the empirical entropy $\mathcal{H}$. Fig. 3-(b) confirms the norm growth of the parameters. Note the slower growth of the context embeddings $H$ in the TF model, which we suspect is due to the layer normalization at the final layer. We also confirm Claim **(C4)**, the recovery of the soft labels, in Fig. 3-(c). At last, Fig. 3-(d) and (e) track the correlation of the learned geometries with the proxy **(P)** and with the theoretical predictions ($H^{\mathrm{mm}}$, $W^{\mathrm{mm}}$). Note that all similarity

---

[3]This is the Structural Similarity Index Measure (SSIM) used to measure the similarity between two images (Wang et al., 2004).

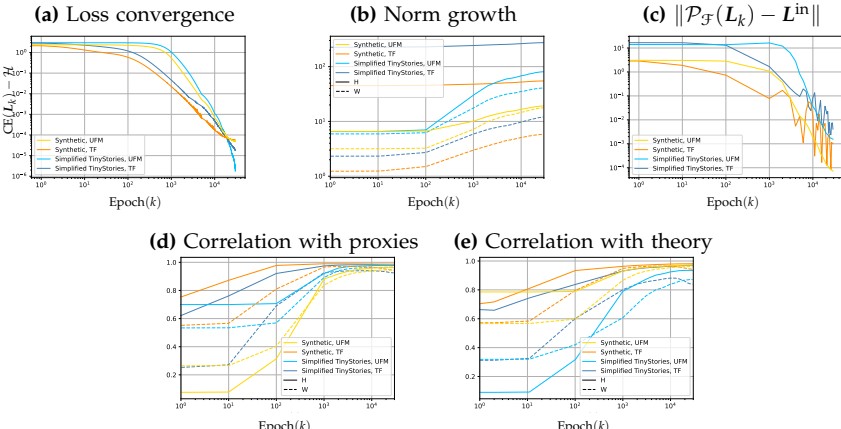

**Figure 3:** Experiments on Synthetic and Simplified TinyStories datasets. **(a):** CE approaches $\mathcal{H}$, **(b):** Parameters' norms grow during training. **(c):** $L_k$'s projection on the data subspace $\mathcal{F}$ converges to the sparse component $L^{\text{in}}$ specified by the soft-labels $\hat{p}_j$. **(d):** $\text{SIM}(H_k, \widehat{S})$ (solid) and $\text{SIM}(W_k^\top, \widehat{S}^\top)$ (dashed). At the final stage the learned embeddings exhibit high similarity with proxy **(P)**. **(e):** Same as (d), this time comparing with theory, i.e., $\text{SIM}(H_k, H^{\text{mm}})$ and $\text{SIM}(W_k^\top, W^{\text{mm}\top})$. Details in Sec. 5.

metrics increase towards 1 as training progresses. In Figs. 12 and 13 in the appendix, we display the evolution of the embeddings by capturing their heatmaps at various training checkpoints. Interestingly, the final patterns emerge early in the training process. Also, see App. D.2.3 for analogous plots for TinyStories.

Finally, in Figs. 14-17, we visualize the text values of the contexts and words along with their geometry heatmaps. We see that the embeddings' similarities are consistent with the linguistic properties of words/contexts. Embeddings of contexts such as "boy named Timmy. Timmy" and "kid called Lilly. She" align closely as they are likely followed by similar words. Also, word embeddings tend to cluster by their grammatical categories, with verbs and adjectives often showing high similarity. For instance, the verbs go, eat, sing, wear, play, and the adjectives clever, fat, furry, lively, little have high similarity.

## 6 Concluding remarks: Limitations and future work

Looking forward, we see several promising avenues for expanding upon our framework and addressing its present limitations. First, we believe it is feasible to broaden the regularization-path analysis to the non-convex realm of problem (8), potentially revealing a rank-constrained variant of NTP-SVM$_\star$. We hypothesize that the support matrices typical of natural language data might accommodate low-rank solutions that adhere to the constraints specified in Eqns. (9)-(10). This would allow us to relaxing the $d \geq V$ constraint in our analysis. See App. D.5 for more discussions. Second, while our initial focus was on establishing the framework, conducting preliminary analyses and proof-of-concept experiments, we encourage more extensive experimental exploration, especially with larger datasets and more complex model architectures with larger context window. Experiments in this paper focused on TF models due to their recent success in language. However, our theoretical results are independent of network design. Instead, they require the model to be over-parameterized enough to reach the entropy lower bound and ensure expressiveness in the embedding space. For a brief discussion, see App. D.3. Third, there is a compelling opportunity to delve into how the distributional characteristics of word/context pairs in natural language, represented by the probability matrix $P$, influence key properties like low-rankness and symmetry in the SVD factors of $L^{\text{mm}}$. Fourth, although the framework encompasses full autoregressive training, our experimental study has focused on prediction of the last token prediction, with the exception of preliminary experiments in App. D.4. Deeper studies are required with sequences of variable length to empirically evaluate the brittleness of our model's assumption, which enumerates distinct contexts regardless of their length or constituent token composition.

Ultimately, our goal is to spur further exploration into the geometries of context and word embeddings. More broadly, we aim to foster interest in deeper investigations into how the inductive bias of NTP influences the learning of linguistic patterns and regularities.

**Acknowledgments**

This work was funded by NSERC Discovery Grant No. 2021-03677, Alliance Grant ALLRP 581098-22 and an Alliance Mission Grant. TB also gratefully acknowledges support by UBC's Four Year Doctoral Fellowship and a scholarship by the Advanced Machine Leaning Training Network at UBC.

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

# Contents

# A Related work: Detailed version

This section expands upon the section on related work in the main body.

## A.1 Word embeddings as matrix factorization

Neural probabilistic language models are designed to predict the probability distribution of a target word $v \in \mathcal{V}$ within a given vocabulary $\mathcal{V}$, based on a sequence of preceding or surrounding words of (variable or fixed) length $t$ known as the context $z \in \mathcal{V}^t$. In these models, word and context representations (aka embeddings) are parameterized by their internal architecture (Bengio et al., 2000). The emergence of energy-based models like the Word2Vec family, particularly through the introduction of training methodologies such as the Skip-Gram with Negative Sampling (SGNS) objective by Mikolov et al. (2013b;a), marked a significant advancement in learning high quality embeddings; e.g., see Jurafsky & Martin (2023). SGNS aims to optimize the alignment between vectors of *target* words and the *context* words surrounding the target, while disassociating those of randomly drawn pairs. This is achieved by optimizing a log-bilinear model that learns unconstrained embeddings for words in two roles: as targets and as contexts. While intuitive, the objective remained somewhat mysterious until the seminal analysis by Levy & Goldberg (2014) offered a formal examination of embeddings generated by the SGNS objective, framing it as a form of weighted matrix factorization. Specifically for embeddings of large dimensionality, Levy and Goldberg demonstrated that SGNS implicitly factorizes a shifted version of the Pointwise Mutual Information (PMI) matrix, denoted as **PMI**, which is a measure of association between words and contexts by having entries $\mathbf{PMI}[z, j] = \log \frac{\Pr(z|j)}{\Pr(z)} = \log \frac{\hat{p}_{j,z}}{\Pr(z)}$ for target word $z$ and context word $j$. This result establishes a fundamental connection between count-based and predictive models (Pennington et al., 2014). However, this approach presumes non-zero occurrences of all context-word pairs in the training set, a condition not always met, leading to undefined factorization for pairs with zero co-occurrence (resulting in $\mathbf{PMI}[z, j] = -\infty$). To address the possibility of negative infinity values, an approximation involving a positive, sparse, and non-negative PMI (PPMI) matrix was proposed. While this adaptation offered an intriguing alternative to the original training objectives, the geometric characteristics of the resultant embeddings in comparison to those derived from the original loss remain unclear.

Our approach mirrors that of Levy & Goldberg (2014) to some extent (see also Pennington et al. (2014)), especially in treating NTP as a soft-label classification across distinct contexts and presuming unconstrained context embeddings (although these concepts are not clearly mentioned in those original works). At a very basic level our work distinguishes itself by focusing on the NTP objective, which, although akin to SGNS and NC objectives, employs cross-entropy loss instead of sigmoid functions. Aside from this distinction, we identify the following conceptual divergences, contributing to our novel perspective on this line of inquiry: **(i)** Firstly, we aim to characterize the geometry of embeddings as learned directly from the loss function, rather than through approximations. **(ii)** Secondly, although NTP-UFM, as a log-bilinear model, is in spirit to the log-bilinear models trained by Word2Vec, we only use such log-bilinear models as an analytically-tractable proxy to expressive deep neural probabilistic language models such as transformers. **(iii)** Thirdly, we confront the sparsity in the probabilistic labels of each context head-on, unlike other matrix-factorization methods that circumvent sparsity in a heuristic manner through smoothing or weighting techniques (Levy & Goldberg, 2014; Pennington et al., 2014). This helps us identify that sparsity leads to diverging embeddings, a scenario where setting the loss gradient to zero is infeasible with finite weights. **(iv)** Fourthly, by examining embeddings through the lens of the regularization path—a surrogate for gradient descent optimization—we unveil that word and context embeddings emerge from the factorization of a matrix $\boldsymbol{L}^{\text{in}} + R\boldsymbol{L}^{\text{mm}}$. Here, $\boldsymbol{L}^{\text{in}}$ captures the frequency of word-context occurrences akin to the PMI, setting non-occurring pairs to zero while retaining positive values for occurring pairs, much like the PPMI matrix. The directional component $\boldsymbol{L}^{\text{mm}}$, becoming dominant as weights diverge, reflects the explicit sparsity patterns of the context-word probability matrix.

The above-mentioned conceptual novelties can also be directly applied to the SGNS objective generalizing the result of Levy & Goldberg (2014) to sparse language patterns. We show this in App. A.4.

The seminal contribution of Levy & Goldberg (2014) has inspired numerous subsequent studies, leveraging the geometric insights of embeddings to uncover linguistic phenomena such as the linear relationships underlying word analogies (Allen & Hospedales, 2019; Hashimoto et al., 2016; Arora et al., 2016; Ethayarajh et al., 2018; Mikolov et al., 2013c). Other works, investigate the *anisotropy* of the learned word/context embeddings for both static models such as Word2Vec (Mu et al., 2017) and contextualized models such as pre-trained BERT and GPT-2 (Ethayarajh, 2019; Li et al., 2020). This property of the embeddings has been attributed to the imbalanced long-tail distribution of language (Gao et al., 2019; Li et al., 2020), shifts in the embedding space (Biś et al., 2021), and characteristic of the network architecture (Godey et al., 2024). Various techniques have been proposed to improve the isotropy of the embeddings, which yields embeddings with better language regularities (e.g., (Mu et al., 2017; Gao et al., 2019; Biś et al., 2021; Timkey & Van Schijndel, 2021)). We envision that our fresh perspective on geometric analysis and our findings could similarly motivate further research in this domain, extending the understanding and application of language model embeddings.

## A.2 Overparameterization and Implicit bias/regularization of GD

Recent studies on gradient-based optimization methods, such as SGD and its variants, have focused on overparameterized settings in empirical risk minimization without explicit regularization. This interest stems from observations that deep neural networks, especially in image classification, perform exceptionally well even with minimal or no weight decay. This research area, known as "implicit bias" or "implicit regularization" of GD emerged to explore which among the numerous potential minimizers in overparameterized settings are preferred by algorithms like GD. Notably, Soudry et al. (2018); Ji & Telgarsky (2018) showed that for linear logistic regression with one-hot labels, GD aligns with the hard-margin SVM solution, essentially becoming the max-margin classifier. Additionally, Ji & Telgarsky (2020) indicated that GD's optimization path in linear models is analogous to the regularization path, which looks at risk minimizers as regularization diminishes. This path is analytically more accessible and offers a useful framework to study GD's behavior. In our work, we apply this idea to investigate the regularization path of NTP training. These foundational results on implicit bias have spurred further research into stochastic and adaptive gradient methods (Nacson et al., 2019; Pesme et al., 2021; Sun et al., 2022; Azizan et al., 2021; Cattaneo et al., 2023), and more intricate architectures (Lyu & Li, 2020; Ji & Telgarsky, 2020; Gunasekar et al., 2018a;b)) including recent studies on transformers (Tarzanagh et al., 2023b;a); for a review, refer to Vardi & Shamir (2021). These insights have also paved the way for generalization analysis, linking GD's generalization to that of SVM-like solutions. This connection has been explored in high-dimensional contexts where traditional margin-based bounds are inadequate, revealing scenarios where no regularization yields optimal outcomes, a phenomenon known as 'harmless interpolation' or 'benign overfitting.' Most of these findings, whether on optimization or generalization, are confined to the one-hot classification framework.

In the language modeling literature, several theoretical works suggest that achieving the empirical entropy lower-bound at the pre-training stage results in better down-stream performance (Liu et al., 2023; Saunshi et al., 2020). Liu et al. (2023) attributes this better down-stream performance to the implicit bias of the pre-training algorithms, by showing that the optimizers prefer the more transferable models among the ones that achieve the same minimal loss value. A formal deviation from one-hot to soft-label classification for NTP is the work by (Thrampoulidis, 2024), which partially inspired our research. While their analysis, assuming fixed context embeddings, applies to linear models, our work extends this by optimizing both word and context embeddings, leading to a nonlinear model. We discover that this nonlinearity yields a preference towards minimizing the nuclear norm of the logits over the Euclidean norm of the word embedding matrix, as seen in (Thrampoulidis, 2024). Our theoretical contributions thus expand on their findings into the nonconvex domain.

### A.3 Neural collapse geometry

A third area of research closely related to our work is the investigation of neural-collapse (NC) geometry, which seeks to understand the geometry of last-layer features and classifier weights in deep networks optimized in the interpolating regime. Initiated by Papyan et al. (2020), this line of research uncovered a universal phenomenon across various architectures and datasets, where classifiers and last-layer features consistently converge to an equiangular tight frame (ETF). This convergence subsumes the aggregation of same-class features around their class mean, a phenomenon termed NC. The theoretical underpinnings for these empirical findings were later provided by analyzing the unconstrained features model (UFM) (Mixon et al., 2020; Fang et al., 2021; Zhu et al., 2021), suggesting that highly overparameterized networks can produce arbitrary last-layer features. This concept, reminiscent of the early framework used in Word2Vec models (Levy & Goldberg, 2014) as discussed above (see also Yang et al. (2017)), has since spurred extensive research in diverse contexts.

Among the vast body of work on NC geometry–an exhaustive reference of which is beyond our scope–, two studies stand out for their relevance to our research. Thrampoulidis et al. (2022) first adapted the UFM to describe geometries diverging from symmetric ETF by moving away from the balanced data assumption, demonstrating that logits optimizing the UFM with vanishing regularization align with a nuclear-norm SVM problem; see also Behnia et al. (2023). This approach also bridges NC geometry with the implicit bias literature previously discussed. Our work extends their principles to the NTP objective, differing from the one-hot encoding classification focus of their study. We take a similar approach, but this time applied to the NTP objective. This is unlike their result which only applies to one-hot encoding classification. In fact, our formulation in NTP-SVM$_\star$ extends their findings, encompassing them as a specific instance when each context is followed by only one token—a scenario less applicable to the linguistic contexts we focus on. Moreover, we demonstrate (see Theorem 2) that the NTP framework introduces a finite projection component to the regularization path, a feature absent in one-hot settings.

Our findings also resonate to certain extent with the recent multilabel NC geometry study in Li et al. (2023), where each example, typically an image, is tagged with multiple correct labels, represented as a $k$-hot vector. In contrast, our NTP framework, modeled as soft-label classification, associates each context with a probability vector. This distinction makes our framework more versatile, applicable beyond the restrictive case where soft-label classification aligns with multi-label classification—specifically, when all relevant tokens are equally likely. There are also crucial differences in the outcomes. Their analysis assumes an equal number of training samples of multiplicity $k = 1$ for each category, a condition that, in our context, would necessitate every vocabulary word to be followed by an identical number of contexts that are only followed by that word—a significantly limiting constraint. Our findings, however, are not bound by this assumption.

In an attempt to extend this line of work to language setups, contemporaneous work Wu & Papyan (2024), through experiments, suggests a correlation between the degree of convergence to the NC geometry and the validation loss for different language models trained on TinyStories for a few epochs. Different to us, they implicitly treat the NTP objective as one-hot classification and suggest that better *per-class* (i.e., next-token) collapse of the context embeddings leads to improved validation loss. In contrast, our analysis reveals a more intricate finding: within a sparse soft-label formulation of NTP training with language, it is not *per-class* collapse that minimizes the training objective. Instead, the objective is minimized when context embeddings with the same *support set* collapse after being projected onto *subspace* $\mathcal{F}_\perp$, which is defined by the sparsity pattern of the context-word co-occurrence matrix (see Sec. 4.3). Our approach also comes with an analysis framework rather than relying solely on experiments.

### A.4 Connections to Word2Vec

We apply our geometry analysis to Word2Vec. Consider the following Skip-gram objective:

$$\frac{1}{T} \sum_{t \in [T]} \sum_{-c \leq j \leq c, j \neq 0} \log p\left(z_{t+j} | z_t\right) = \frac{1}{T} \sum_{t \in [T]} \sum_{-c \leq j \leq c, j \neq 0} \log p\left(\mathsf{S}_{z_{t+j}}\left(\boldsymbol{H} \boldsymbol{w}_{z_t}\right)\right),$$

where $H \in \mathbb{R}^{V \times d}$ denotes the embedding matrix of *context* words $z_{t+j}, j \neq 0$ and $W \in \mathbb{R}^{V \times d}$ denotes the embedding matrix of *target words* $z_t$. As presented in Mikolov et al. (2013c), skip-gram actually implements a heuristic of this objective that aims to simplify the computation of the normalizing factor in the softmax. Second, consider the CBOW objective:

$$\frac{1}{T} \sum_{t \in [T]} \log p\left(z_t | \{z_{t+j}\}_{\{|j| \in [c], j \neq 0\}}\right) = \frac{1}{T} \sum_{t \in [T]} \log p\left(S_{z_t}(W \cdot \frac{1}{2c-1} \sum_{-c \leq j \leq c, j \neq 0} h_{z_{t+j}})\right),$$

where $\frac{1}{2c-1} \sum_{-c \leq j \leq c, j \neq 0} h_{z_{t+j}}$ is the continuous-bags-of-word embedding of the context $\{z_{t+j}\}_{\{|j| \in [c], j \neq 0\}}$.

Suppose the CBOW model with context consisting of only the previous word. To keep notation consistent with the rest of the paper, let $j \in [m] = [V]$ denote index of *distinct* contexts (aka previous words) and $z \in [V]$ denote index of (next) words. Note the embedding matrix $H \in \mathbb{R}^{d \times V}$ is now of same dimension as $W^\top$. Let $\hat{\pi}_j$ denote the marginal of context words and $\hat{p}_{j,z}$ denote the conditional probability of word $z$ following context word $j$. Then, the CBOW objective becomes equivalent to

$$\sum_{j \in [V]} \hat{\pi}_j \sum_{z \in [V]} \hat{p}_{j,z} \log\left(S_z\left(Wh_j\right)\right).$$

This now fits exactly our framework and the results for NTP apply mutatis mutandis.

In Mikolov et al. (2013c), instead of minimizing the CE objective, they actually use the Negative-Sampling (SGNS) objective. Levy & Goldberg (2014) rewrites the SGNS objective as follows:

$$\sum_{j \in [m]} \sum_{z \in [V]} - \Pr(j,z) \log\left(\sigma(w_z^\top h_j)\right) - k \Pr(j) \Pr(w) \log\left(1 - \sigma(w_z^\top h_j)\right),$$

where $\Pr(j,z) := \hat{\pi}_j \hat{p}_{j,z}$, $\Pr(j) := \hat{\pi}_j$, $\Pr(z) = \sum_{j \in [m]} \hat{\pi}_j \hat{p}_{j,z}$ and $k \in \mathbb{N}$ is a hyperparameter. Defining

$$q_{j,z} := \frac{\frac{\Pr(j,z)}{\Pr(j)\Pr(z)}}{k + \frac{\Pr(j,z)}{\Pr(j)\Pr(z)}} \in [0,1],$$

we can rewrite the SGNS objective as

$$\sum_{j \in [m]} \sum_{z \in [V]} \Pr(j) \Pr(w) \left(k + \frac{\Pr(j,z)}{\Pr(j)\Pr(z)}\right) \left(-q_{j,z} \log\left(\sigma(w_z^\top h_j)\right) - (1 - q_{j,z}) \log\left(1 - \sigma(w_z^\top h_j)\right)\right).$$

For each $(j,z)$-pair, the rightmost term in the expression above is lower bounded by $\mathcal{H}(q_{j,z}) := -q_{j,z} \log(q_{j,z}) - (1 - q_{j,z}) \log(1 - q_{j,z})$.

By following the same arguments developed for NTP, it can be checked that the lower bound is attained if and only if the following conditions hold:

A. There exists logit matrix $L_1 \in \mathbb{R}^{V \times m}$ (of rank $\leq d$) such that

$$L_1[z,j] = \text{PMI}(z,j) - \log(k) = \log\left(\frac{\Pr(j,z)}{\Pr(j)\Pr(z)}\right) - \log(k), \quad \forall j \in [m], z \in \mathcal{S}_j.$$

.

B. There exists logit matrix $L_2 \in \mathbb{R}^{V \times m}$ (of rank $\leq d$) such that

$$L_2[z,j] = 0, \quad \forall j \in [m], z \in \mathcal{S}_j,$$
$$L_2[v,j] < 0, \quad \forall j \in [m], v \notin \mathcal{S}_j.$$

The logit matrix $L_1$ is the shifted PMI matrix as also characterized in Levy & Goldberg (2014, Eq. (7)). This component plays a similar role to the finite component $L^{\text{in}}$ in the NTP setup of our paper. The second component, which is only non-zero if there exists a target-context

pair $(z, j)$ with zero co-occurrence in the training set $(\Pr(j, z) = 0)$, specifies the directional component, akin to the role of $L^{\text{mm}}$ in NTP. This component is absent in the analysis of Levy & Goldberg (2014), as they implicitly assume no zero entries in the co-occurence matrix when setting the derivative of the loss to zero. However, when the co-occurrence matrix is sparse, zero derivation of the loss is not achieved at any finite solution. Instead, $L_2$ identifies the infinite direction that drives the loss to its lower-bound. Levy & Goldberg (2014) address sparsity indirectly by discussing alternative matrix factorizations such as shifted positive-PMI (PPMI).

## B   Proofs

### B.1   Centering of word embeddings

**Lemma 2.** *For any $\lambda > 0$, any minimizer $\widehat{W}$ of NTP-UFM satisfies $\mathbb{1}_V^\top \widehat{W} = 0$. That is, word embeddings are centered.*

*Proof.* The proof is the same as Thrampoulidis et al. (2022, Lem. E.1) and is thus omitted. ∎

There is two consequences of Lemma 2:

A. The word embedding matrix $W$ is centered motivating the use of the *centered* support matrix $\widetilde{S}$ (rather than simply $S$) in Proxy **(P)**.

B. The constraint $\mathbb{1}^\top W = 0$ can be added to NTP-UFM without changing its optimal set. Thus, the non-convex rank constraint can be ignored, making the problem convex, provided $d \geq V - 1$. Finally, note this also implies the constraint $\mathbb{1}^\top L = 0$ can be added without any change in Eq. (8).

### B.2   Proof of Lemma 1

We prove a slightly stronger version of Lemma 1 which assumes the additional constraint $\mathbb{1}^\top L = 0$ in Eq. NTP-UFM; see Lemma 2. The proof follows identically if we ignore this centering property.

As mentioned in the main body, the proof of the lemma essentially relies on the well-known fact that

$$\|L\|_* = \min_{WH=L} \frac{1}{2}\|W\|^2 + \frac{1}{2}\|H\|^2 .$$

Firstly, fix any $L$ such that $r := \operatorname{rank}(L) \leq d$ and $\mathbb{1}_V^\top L = 0$. Let $L = U\Sigma V^\top$ be its SVD with $\Sigma \in \mathbb{R}^{r \times r}$. Choose

$$\hat{W} = U\Sigma^{1/2}\mathbf{R} \qquad \text{and} \qquad \hat{H} = \mathbf{R}^\top \Sigma^{1/2} V^\top$$

for some partial orthonormal matrix $\mathbf{R} \in \mathbb{R}^{r \times d}$ (i.e., $\mathbf{R}^\top \mathbf{R} = \mathbb{I}_d$). It then holds clearly that

$$\hat{W}\hat{H} = L \qquad \text{and} \qquad \|\hat{W}\|^2 + \|\hat{H}\|^2 = \operatorname{tr}\left(U\Sigma U^\top\right) + \operatorname{tr}\left(V\Sigma V^\top\right) = 2\operatorname{tr}(\Sigma) = 2\|L\|_*,$$

Thus,

$$\text{CE}(L) + \lambda\|L\|_* = \text{CE}(\hat{W}\hat{H}) + \frac{\lambda}{2}\left(\|\hat{W}\|^2 + \|\hat{H}\|^2\right) .$$

Because we can apply this for any $L$, it shows that[4]

$$\min_{\substack{L \in \mathcal{R}_d \\ \mathbb{1}^\top L = 0}} \text{CE}(L) + \lambda\|L\|_* \geq \min_{W,H} \text{CE}(WH) + \frac{\lambda}{2}\left(\|W\|^2 + \|H\|^2\right) , \tag{12}$$

---

[4]Here, use the fact that the minimizer of the LHS of (12) is attained because of coercivity of the nuclear norm.

where we define $\mathcal{R}_d := \{L \in \mathbb{R}^{V \times m} : \text{rank}(L) \leq d\}$ to be the manifold of rank-constrained matrices.

Secondly, consider any $W \in \mathbb{R}^{V \times d}, H \in \mathbb{R}^{d \times m}$ such that $\mathbb{1}_V^\top W = 0$. Let $L = WH$ and denote $L = U\Sigma V^\top$ its SVD. We then have $\mathbb{1}^\top L = 0$ and

$$
\begin{aligned}
\text{CE}(L) + \lambda \|L\|_* = \text{CE}(WH) &+ \lambda \, \text{tr}\left(U^\top WHV\right) \\
&\leq \text{CE}(WH) + \frac{\lambda}{2}\left(\|U^\top W\|^2 + \|HV\|^2\right) \\
&\leq \text{CE}(WH) + \frac{\lambda}{2}\left(\|W\|^2 + \|H\|^2\right),
\end{aligned}
\tag{13}
$$

where in the first line we used $\|L\|_* = \text{tr}(\Sigma) = \text{tr}(U^\top LV)$, in the second line we used Cauchy-Schwartz, and, in the third line we used that $\|U\|_2 \leq 1, \|V\|_2 \leq 1$. Thus, this proves

$$
\begin{aligned}
\min_{W,H} \text{CE}(WH) + \frac{\lambda}{2}\left(\|W\|^2 + \|H\|^2\right) &= \min_{\substack{W,H \\ \mathbb{1}^\top W = 0}} \text{CE}(WH) + \frac{\lambda}{2}\left(\|W\|^2 + \|H\|^2\right) \\
&\geq \min_{\substack{L \in \mathcal{R}_d \\ \mathbb{1}^\top L = 0}} \text{CE}(L) + \lambda \|L\|_*,
\end{aligned}
\tag{14}
$$

where the first equality follows by Lemma 2. Moreover, note that equalities in (13) hold if and only if the following three conditions hold

$$
\begin{aligned}
WH = L = U\Sigma V^\top &\implies U^\top WHV = \Sigma \\
U^\top W &= V^\top H^\top \\
\|U^\top W\|^2 = \|W\|^2, &\quad \|V^\top H^\top\|^2 = \|H\|^2.
\end{aligned}
$$

The first two conditions combined give

$$
U^\top WW^\top U = V^\top H^\top HV = \Sigma.
$$

Thus, $WW^\top = U\Sigma U^\top + U_\perp \Sigma_\perp U_\perp^\top$ for some non-negative diagonal matrix $\Sigma_\perp$ and $U_\perp$ the orthogonal complement of $U$. But, the third condition requires

$$
\text{tr}\left(U^\top WW^\top U\right) = \text{tr}\left(WW^\top\right) \implies \text{tr}(\Sigma) = \text{tr}(\Sigma) + \text{tr}(\Sigma_\perp) \implies \text{tr}(\Sigma_\perp) = 0 \implies \Sigma_\perp = 0.
$$

This gives

$$
WW^\top = U\Sigma U^\top,
\tag{15}
$$

and similarly

$$
H^\top H = V\Sigma V^\top.
\tag{16}
$$

Combining Eqns. (12) and (14) proves that

$$
\min_{W,H} \text{CE}(WH) + \frac{\lambda}{2}\left(\|W\|^2 + \|H\|^2\right) = \min_{\substack{L \in \mathcal{R}_d \\ \mathbb{1}^\top L = 0}} \text{CE}(L) + \lambda \|L\|_*.
$$

Moreover, the requirements for equalities in Eq. (13) established in Eqs. (15) and (16) show the desired about the form of gram matrices $W_\lambda W_\lambda^\top$ and $H_\lambda H_\lambda^\top$ as stated in the lemma.

### B.3   Proof of Theorems 1 and 2

The proof involves several components, and so, we proceed in steps.

### B.3.1 Notations recap

We begin by recaping the essential notation used throughout this section.

Recall NTP-SVM$_\star$: The constraints in Eq. (9) require in-support logits to be equal to each other for every context. For fixed $j \in [m]$, the set of equality constraints for all $z \neq z' \in \mathcal{S}_j$ is equivalent to the set of the same constraints for any anchor $z_j \in \mathcal{S}_j$ and $z' \neq z_j \in \mathcal{S}_j$. That is, there is an effective total of $S_j - 1$ linearly independent constraints for each $j \in [m]$.

**Definition 3** ($E_{\text{in,j}}$). *For each $j \in [m]$ fix an anchor index $z_j \in \mathcal{S}_j$ and define matrix $E_{\text{in,j}} \in \mathbb{R}^{(S_j-1)\times V}$ with $S_j - 1$ independent rows $(e_{z_j} - e_z)^\top, z \neq z_j, z \in \mathcal{S}_j$.*

With this definition, we can rewrite Eq. (9) since

$$E_{\text{in,j}}\ell_j = 0, \forall j \in [m] \iff L = [\ell_1, \dots, \ell_m] \text{ satisfies Eq. (9)}$$

The inequality constraints in Eq. (10), impose the requirement that in-support logits must be greater than out-of-support logits. Considering Eq. (9), we can again select any anchor $z_j \in \mathcal{S}_j$ and reformulate Eq. (10) as $E_{\text{out,j}}\ell_j \geq 1$, $j \in [m]$. Here, $E_{\text{out,j}} \in \mathbb{R}^{(V-S_j)\times V}$ has $V - S_j$ independent rows $(e_{z_j} - e_v)^\top, v \notin \mathcal{S}_j$. In total, there are $V - 1$ constraints per context in NTP-SVM$_\star$.

With the above definition of the matrices $E_{\text{in,j}} \in \mathbb{R}^{(S_j-1)\times V}, j \in [m]$, we can define the subspace $\mathcal{F}$ of $V \times m$ matrices that have their $j$-th column living on the range space $\mathcal{R}(E_{\text{in,j}}^\top)$ for all $j \in [m]$:

$$\mathcal{F} := \left\{ L \in \mathbb{R}^{V\times m} : \ell_j \in \mathcal{R}(E_{\text{in,j}}^\top), j \in [m] \right\} = \text{span}\left( \left\{ (e_z - e_{z'})^\top \widetilde{e}_j : z \neq z' \in \mathcal{S}_j, j \in [m] \right\} \right).$$

Note that this set depends on the support matrix $S$, but we suppress the dependence to simplify notation. Let $\mathcal{P}_{\mathcal{F}} : \mathbb{R}^{V\times m} \to \mathbb{R}^{V\times m}$ be the projection operator on this subspace. For concreteness, note that this can be explicitly defined as the matrix operator projecting each column $\ell_j$ of $L$ onto $\mathcal{R}(E_{\text{in,j}}^\top)$, i.e.,

$$\mathcal{P}_{\mathcal{F}}(L) = [\mathcal{P}_{\mathcal{F}}(L)_1 \quad \mathcal{P}_{\mathcal{F}}(L)_2 \quad \cdots \quad \mathcal{P}_{\mathcal{F}}(L)_m] \text{ with } \mathcal{P}_{\mathcal{F}}(L)_j := E_{\text{in,j}}^\top (E_{\text{in,j}}E_{\text{in,j}}^\top)^{-1} E_{\text{in,j}}\ell_j, j \in [m].$$

Note that $\mathcal{P}_{\mathcal{F}}$ is a projection operator since it is clearly linear and $\mathcal{P}_{\mathcal{F}}(\mathcal{P}_{\mathcal{F}}(L)) = \mathcal{P}_{\mathcal{F}}(L)$. Also, it is clear that $\mathcal{P}_{\mathcal{F}}(L) \in \mathcal{F}$ for all $L \in \mathbb{R}^{V\times m}$. Operationally, note that $\mathcal{P}_{\mathcal{F}}(L)$ has zero-entries on off-support elements. We will use this property indiscriminately throughout this section.

Finally, define the orthogonal complement to $\mathcal{F}$ as

$$\mathcal{F}_\perp := \left\{ L \in \mathbb{R}^{V\times m} : \ell_j \in \mathcal{N}(E_{\text{in,j}}), j \in [m] \right\} = \left\{ L \in \mathbb{R}^{V\times m} : E_{\text{in,j}}\ell_j = 0, j \in [m] \right\}. \tag{17}$$

Clearly, $\mathbb{R}^{V\times m} = \mathcal{F} \oplus \mathcal{F}_\perp$. Operationally, note that $L \in \mathcal{F}_\perp$ implies that on-support elements of every column are identical, that is, $L[z, j] = L[z', j]$ for all $z \in \mathcal{S}_j$ and $j \in [m]$. In other words, it holds

$$L \in \mathcal{F}_\perp \iff L \text{ satisfies (9)}, \tag{18}$$

which we will use without further explicit reference.

We also let $\mathcal{P}_\perp$ denote the projection operator to $\mathcal{F}_\perp$.

### B.3.2 When does the NTP loss attain its entropy lower-bound?

For word/context embeddings $W, H$ and logits $L = WH$, recall the NTP loss is written as

$$\text{CE}(WH) = \text{CE}(L) = \sum_{j\in[m]} \hat{\pi}_j \sum_{z\in\mathcal{S}_j} \hat{p}_{j,z} \log\left( 1 + \sum_{z'\neq z\in\mathcal{S}_j} e^{-(L[z,j]-L[z',j])} + \sum_{v\notin\mathcal{S}_j} e^{-(L[z,j]-L[v,j])} \right), \tag{19}$$

and is lower bounded by the empirical $T$-gram entropy of the data (Shannon, 1948), i.e., for all $\theta'$: $\mathrm{CE}(\boldsymbol{WH}) \geq \mathcal{H} := \hat{\mathbb{E}}_{(\boldsymbol{x},z) \sim \mathcal{T}_n} \left[ -\log\left( \hat{p}(z|\boldsymbol{x}) \right) \right]$.

We now state explicit conditions on the logit matrix $\boldsymbol{L}$ for which the lower bound is attained by a softmax model $\boldsymbol{L} = \mathrm{S}(\boldsymbol{WH})$ with unconstrained context embeddings $\boldsymbol{H}$. Due to sparsity of the conditional probability matrix $\boldsymbol{P}$, the lower bound is *not* attained for any finite logit matrix, but it can be reached asymptotically provided the conditions stated in the proposition below hold.

The statement below is a slight extension of Thrampoulidis (2024, Prop. 1).

**Proposition 1.** *The NTP objective of a softmax model $\mathrm{S}(\boldsymbol{WH})$ with $\boldsymbol{W} \in \mathbb{R}^{V \times d}, \boldsymbol{H} \in \mathbb{R}^{d \times m}$ can reach the empirical entropy lower-bound if the following three conditions hold simultaneously*

    *A. [NTP$_\mathcal{H}$-compatibility] There exists logit matrix $\boldsymbol{L}_1 \in \mathbb{R}^{V \times m}$ such that*

$$\forall j \in [m], z \neq z' \in \mathcal{S}_j \ : \ \boldsymbol{L}_1[j,z] - \boldsymbol{L}_1[j,z'] = \log\left( \frac{\hat{p}_{j,z}}{\hat{p}_{j,z'}} \right). \tag{20}$$

    *B. [NTP-separability] There exists logit matrix $\boldsymbol{L}_2 \in \mathbb{R}^{V \times m}$ that is feasible in NTP-SVM$_\star$.*

    *C. [Rank constraint] For all arbitrarily large positive scalars $\rho$,*

$$\mathrm{rank}\left( \boldsymbol{L}_1 + \rho \cdot \boldsymbol{L}_2 \right) \leq d.$$

*Concretely, if the above conditions hold, then for all arbitrarily large $\rho$, there exist $\boldsymbol{W}_\rho \in \mathbb{R}^{V \times d}$ and $\boldsymbol{H}_\rho \in \mathbb{R}^{d \times m}$ such that*

$$\lim_{\rho \to \infty} \mathrm{CE}(\boldsymbol{W}_\rho \boldsymbol{H}_\rho) = \mathcal{H}.$$

*Proof.* It suffices to prove that for $\boldsymbol{L}_\rho := \boldsymbol{L}_1 + \rho \cdot \boldsymbol{L}_2$:

$$\lim_{\rho \to \infty} \mathrm{CE}(\boldsymbol{L}_\rho) = \mathcal{H}. \tag{21}$$

To see that this is sufficient, note from condition $C$ of the theorem that for all arbitrarily large $\rho$, $\boldsymbol{L}_\rho$ admits a factorization $\boldsymbol{L}_\rho = \boldsymbol{W}_\rho \boldsymbol{H}_\rho$ where $\boldsymbol{W}_\rho \in \mathbb{R}^{V \times d}$ and $\boldsymbol{H}_\rho \in \mathbb{R}^{d \times m}$. Since this holds for all large $\rho$, Eq. (21) implies the desired.

Thus, we now prove Eq. (21). For any $j \in [m]$, it is easy to check, using conditions $A$ and $B$ of the theorem, that the following two statements are true for all $\rho > 0$:

    A. For all $z \neq z' \in \mathcal{S}_j$:

$$\begin{aligned} \boldsymbol{L}_\rho[z,j] - \boldsymbol{L}_\rho[z',j] &= (\boldsymbol{L}_1[z,j] - \boldsymbol{L}_1[z,j]) + \rho \cdot (\boldsymbol{L}_2[z,j] - \boldsymbol{L}_2[z',j]) \\ &= \log(\hat{p}_{j,z}/\hat{p}_{j,z'}) \end{aligned}$$

    B. For all $z \in \mathcal{S}_j, v \notin \mathcal{S}_j$:

$$\begin{aligned} \boldsymbol{L}_\rho[z,j] - \boldsymbol{L}_\rho[v,j] &= (\boldsymbol{L}_1[z,j] - \boldsymbol{L}_1[v,j]) + \rho \cdot (\boldsymbol{L}_2[z,j] - \boldsymbol{L}_2[v,j]) \\ &\geq (\boldsymbol{L}_1[z,j] - \boldsymbol{L}_1[v,j]) + \rho \\ &\geq -2\|\boldsymbol{L}_1\|_2 + \rho. \end{aligned}$$

In the last inequality above, we used the loose bound $\|\boldsymbol{L}_1\|_2 \geq \max_{j \in [m], z \in [V]} |\boldsymbol{L}_1[z,j]|$ for the spectral norm of $\boldsymbol{L}_1$.

Using these in Eq. (19) gives:

$$
\begin{aligned}
\text{CE}(\boldsymbol{L}_\rho) &\leq \sum_{j\in[m]} \hat{\pi}_j \sum_{z\in\mathcal{S}_j} \hat{p}_{j,z} \log\left(1 + \sum_{z'\neq z\in\mathcal{S}_j} \frac{\hat{p}_{j,z'}}{\hat{p}_{j,z}} + Ve^{-\rho}e^{-2\|L_1\|_2}\right) \\
&= \sum_{j\in[m]} \hat{\pi}_j \sum_{z\in\mathcal{S}_j} \hat{p}_{j,z} \log\left(\frac{1}{\hat{p}_{j,z}} + Ve^{-\rho}e^{-2\|L_1\|_2}\right) \\
&= \sum_{j\in[m]} \hat{\pi}_j \sum_{z\in\mathcal{S}_j} \hat{p}_{j,z} \log\left(\frac{1}{\hat{p}_{j,z}}\right) + \sum_{j\in[m]} \hat{\pi}_j \sum_{z\in\mathcal{S}_j} \hat{p}_{j,z} \log\left(1 + \hat{p}_{j,z}Ve^{-\rho}e^{-2\|L_1\|_2}\right). \\
&\leq \mathcal{H} + Ve^{-2\|L_1\|_2}e^{-\rho}.
\end{aligned}
\tag{22}
$$

The last inequality uses $\log(1+x) \leq x, x > 0$.

Clearly the bound above converges to $\mathcal{H} = \sum_{j\in[m]} \hat{\pi}_j \sum_{z\in\mathcal{S}_j} \hat{p}_{j,z} \log\left(\frac{1}{\hat{p}_{j,z}}\right)$ as $\rho \to \infty$. From this and $\text{CE}(\boldsymbol{L}_\rho) \geq \mathcal{H}$, the desired Eq. (21) follows. This completes the proof. $\square$

Note that the first two conditions of the theorem are easy to satisfy.

On the one hand, Eq. (20) always has a solution. For later use, we also note that the solution is unique on the subspace $\mathcal{F}$ and we denote this solution as $\boldsymbol{L}^{\text{in}}$. This is easy to check: for any $j \in [m]$, $\boldsymbol{\ell}_j$ is a solution of Eq. (20) if and only if it takes the form $\boldsymbol{\ell}_j = \boldsymbol{E}_{\text{in,j}}^\top \left(\boldsymbol{E}_{\text{in,j}}\boldsymbol{E}_{\text{in,j}}^\top\right)^{-1} \boldsymbol{a}_j + \boldsymbol{v}, \boldsymbol{v} \in \mathcal{N}(\boldsymbol{E}_{\text{in,}})$ where $\boldsymbol{a}_j \in \mathbb{R}^{S_j-1}$ has entries $\log(\hat{p}_{j,z}/\hat{p}_{j,z'}), z' \neq z \in \mathcal{S}_j$. Thus, $\boldsymbol{L}^{\text{in}}$ is explicitly defined as follows.

**Lemma 3** ($\boldsymbol{L}^{\text{in}}$). *Let $\boldsymbol{L}^{\text{in}}$ be the unique solution of Eq. (20) on the subspace $\mathcal{F}$. Then, $\boldsymbol{L}^{\text{in}}$ has columns*

$$
\boldsymbol{\ell}_j^{\text{in}} = \boldsymbol{E}_{\text{in,j}}^\top \left(\boldsymbol{E}_{\text{in,j}}\boldsymbol{E}_{\text{in,j}}^\top\right)^{-1} \boldsymbol{a}_j, \ \forall j \in [m],
$$

*where $\boldsymbol{a}_j \in \mathbb{R}^{S_j-1}$ has entries $\log(\hat{p}_{j,z_j}/\hat{p}_{j,z'})$ for $z' \neq z_j \in \mathcal{S}_j$.*

On the other hand, as mentioned in Sec. 4.1, NTP-SVM$_\star$ is always feasible.

The challenge is in satisfying conditions *A-C* simultaneously.

However, when $d \geq V$, then the rank condition $C$ of the theorem is trivial. Thus, we have the following corollary.

**Corollary 2.** *Suppose $d \geq V$, then the entropy lower bound can be asymptotically attained by the NTP loss.*

### B.3.3 Ball-constrained CE minimization and main result

Instead of the regularized problem (8), it is more convenient (but, equivalent due to convexity) to study the following ball-constrained version:

$$
\widehat{\boldsymbol{L}}_B \in \arg\min_{\|L\|_*\leq B} \text{CE}(\boldsymbol{L}),
\tag{23}
$$

parameterized by $B > 0$. We will study properties of the minimizers $\widehat{\boldsymbol{L}}_B$ of (23) as $B \to \infty$. The theorem below subsumes Theorems 1 and 2.

**Theorem 3.** *Suppose $d \geq V$. Denote $\widehat{\boldsymbol{L}}_B$ any solution of (23) for regularization B. Then, in the limit of diverging regularization the following statements are true.*

*(i)* $\lim_{B\to\infty} \text{CE}(\widehat{\boldsymbol{L}}_B) = \mathcal{H}$.

*(ii)* $\lim_{B\to\infty} \mathcal{P}_\mathcal{F}(\widehat{\boldsymbol{L}}_B) = \boldsymbol{L}^{\text{in}}$, *where $\boldsymbol{L}^{\text{in}}$ is defined in Lemma 3.*

*(iii) The solution diverges in norm:* $\lim_{B\to\infty} \|\widehat{\boldsymbol{L}}_B\| = +\infty$.

*(iv) The solution converges in direction to the optimal set of NTP-SVM$_\star$. That is, there exists minimizer $\boldsymbol{L}^{\mathrm{mm}}$ of NTP-SVM$_\star$ such that*

$$\lim_{B \to \infty} \left\| \frac{\widehat{\boldsymbol{L}}_B}{\|\widehat{\boldsymbol{L}}_B\|_*} - \frac{\boldsymbol{L}^{\mathrm{mm}}}{\|\boldsymbol{L}^{\mathrm{mm}}\|_*} \right\| = 0.$$

### B.3.4 The in-support NTP loss and its minimizer $L^{\mathrm{in}}$

Define the in-support NTP loss as follows:

$$\mathrm{CE}_{\mathrm{in}}(\boldsymbol{L}) = \sum_{j \in [m]} \hat{\pi}_j \sum_{z \in \mathcal{S}_j} \hat{p}_{j,z} \log \left( 1 + \sum_{z' \neq z \in \mathcal{S}_j} e^{-(L[z,j] - L[z',j])} \right)$$

Contrary to (19), the summation inside the logarithm is only over word indices $z$ that belong to the support set of context $j$.

By non-negativity of log, for all $\boldsymbol{L}$ it holds $\mathrm{CE}(\boldsymbol{L}) \geq \mathrm{CE}_{\mathrm{in}}(\boldsymbol{L})$. In fact, because of the sparsity ansatz ($\exists j \in [m]$ such that $S_j < V$) and strict nonegativity of exponential, the inequality is strict for all *finite* $\boldsymbol{L}$, i.e.

$$\mathrm{CE}(\boldsymbol{L}) > \mathrm{CE}_{\mathrm{in}}(\boldsymbol{L}), \ \forall \boldsymbol{L} \in \mathbb{R}^{V \times d}. \tag{24}$$

Moreover, observe that for any $\boldsymbol{L}$, it holds that

$$\mathrm{CE}_{\mathrm{in}}(\boldsymbol{L}) = \mathrm{CE}_{\mathrm{in}}(\mathcal{P}_{\mathcal{F}}(\boldsymbol{L})). \tag{25}$$

To see this, decompose $\boldsymbol{L}$ in orthogonal components onto $\mathcal{F}$ and $\mathcal{F}_{\perp}$, respectively, i.e., $\boldsymbol{L} = \mathcal{P}_{\mathcal{F}}(\boldsymbol{L}) + \mathcal{P}_{\perp}(\boldsymbol{L})$. By Eq. (18), $\boldsymbol{L}_{\perp} := \mathcal{P}_{\perp}(\boldsymbol{L})$ satisfies Eq. (9), i.e. $\boldsymbol{L}_{\perp}[z,j] - \boldsymbol{L}_{\perp}[z',j] = 0$ for all $z, z' \in \mathcal{S}_j$ and $j \in [m]$.

The lemma below uses this to show that $\mathrm{CE}_{\mathrm{in}}$ has a unique minimizer in $\mathcal{F}$.

**Lemma 4.** *The in-support loss $\mathrm{CE}_{\mathrm{in}}(\boldsymbol{L})$ has a unique finite minimizer in $\mathcal{F}$, which we denote $\boldsymbol{L}^{\mathrm{in}}$:*

$$\boldsymbol{L}^{\mathrm{in}} = \arg\min_{\boldsymbol{L} \in \mathcal{F}} \mathrm{CE}_{\mathrm{in}}(\boldsymbol{L}). \tag{26}$$

*The minimizer $\boldsymbol{L}^{\mathrm{in}}$ satisfies the NTP$_{\mathcal{H}}$-compatibility condition in Eq. (20) and*

$$\mathrm{CE}_{\mathrm{in},\star} := \mathrm{CE}_{\mathrm{in}}(\boldsymbol{L}^{\mathrm{in}}) = \mathcal{H}.$$

*Proof.* Note that $\boldsymbol{L} \in \mathcal{F}$ is a column-wise separable constraint $\boldsymbol{\ell}_j \in \mathcal{R}(\boldsymbol{E}_{\mathrm{in},j}^\top)$ that restricts the non-zero entries $\boldsymbol{L}[z,j] = \ell_{j,z}$ to indices $z \in \mathcal{S}_j$. The objective is also separable with respect to columns. Without loss of generality fix $j = 1$ and assume $\mathcal{S}_j = [S_j]$. It then suffices proving that the function $f : \mathbb{R}^{S_1} \to \mathbb{R}$:

$$f(\boldsymbol{\ell}) = -\sum_{z \in \mathcal{S}_1} \hat{p}_{1,z} \log \left( \frac{\exp(\ell_z)}{\sum_{z' \in \mathcal{S}_1} \exp(\ell_{z'})} \right), \tag{27}$$

has a unique minimizer in $\mathcal{R}(\widetilde{\boldsymbol{E}}_{\mathrm{in},1}^\top)$, where $\widetilde{\boldsymbol{E}}_{\mathrm{in},1} := \boldsymbol{E}_{\mathrm{in},1}[:, \mathcal{S}_1] \in \mathbb{R}^{(S_1-1) \times S_1}$ is the non-zero block of $\boldsymbol{E}_{\mathrm{in},1}$.

By non-negativity of the KL divergence,

$$f(\boldsymbol{\ell}) \geq -\sum_{z \in \mathcal{S}_1} \hat{p}_{1,z} \log(\hat{p}_{1,z})$$

with equality if and only if,

$$\forall z \in \mathcal{S}_1 : \frac{\exp(\ell_z)}{\sum_{z' \in \mathcal{S}_1} \exp(\ell_{z'})} = \hat{p}_{1,z}. \tag{28}$$

In turn, this is equivalent to

$$\forall z, z' \in \mathcal{S}_1 \; : \; \ell_z - \ell_{z'} = \log(\hat{p}_{1,z}/\hat{p}_{1,z'}) \,. \tag{29}$$

To see the equivalence note the following: On the one hand, starting with Eq. (28) for $z, z' \in \mathcal{S}_1$, dividing both sides of the two equalities, and taking the logarithm, we arrive at Eq. (29). On the other hand, if Eq. (29) holds, then Eq. (28) holds since

$$\frac{\sum_{z' \in \mathcal{S}_1} \exp(\ell_{z'})}{\exp(\ell_z)} = \sum_{z' \in \mathcal{S}_1} \exp(\ell_{z'} - \ell_{z'}) = \frac{\sum_{z' \in \mathcal{S}_1} \hat{p}_{1,z'}}{\hat{p}_{1,z}} = \frac{1}{\hat{p}_{1,z}} \,.$$

Now, observe that (29) is the same as the $\mathrm{NTP}_{\mathcal{H}}$-compatibility equation for $j = 1$. Moreover, note that a logit vector $\ell \in \mathbb{R}^{\mathcal{S}_1}$ solves (29) if and only if for

$$\ell = \widetilde{\boldsymbol{E}}_{\mathrm{in},1}^{\top} \left( \widetilde{\boldsymbol{E}}_{\mathrm{in},1} \widetilde{\boldsymbol{E}}_{\mathrm{in},1}^{\top} \right)^{-1} \boldsymbol{a}_1 + \boldsymbol{v}, \; \boldsymbol{v} \in \mathcal{N} \left( \widetilde{\boldsymbol{E}}_{\mathrm{in},1} \right),$$

where $\boldsymbol{a} \in \mathbb{R}^{\mathcal{S}_1 - 1}$ with entries $\log(\hat{p}_{1,z_1}/\hat{p}_{1,z'})$ for $z' \neq z_1 \in \mathcal{S}_1$. This proves that there is a unique minimizer $\ell_1^{\mathrm{in}} := \widetilde{\boldsymbol{E}}_{\mathrm{in},1}^{\top} \left( \widetilde{\boldsymbol{E}}_{\mathrm{in},1} \widetilde{\boldsymbol{E}}_{\mathrm{in},1}^{\top} \right)^{-1} \boldsymbol{a}_1$ on $\mathcal{R} \left( \widetilde{\boldsymbol{E}}_{\mathrm{in},1}^{\top} \right)$.

Since the choice of $j = 1$ above was arbitrary, this proves the lemma. $\qquad \square$

### B.3.5 Auxiliary lemmas

This section proves a sequence of auxiliary lemmas that are used to prove Thm. 3 in the next section.

The first lemma shows that entropy can only be approached asymptotically, i.e., provided the logit matrix diverges.

**Lemma 5.** *For all $\boldsymbol{L} \in \mathbb{R}^{V \times m}$ and all $\boldsymbol{L}' \in \mathcal{F}_{\perp}$ that additionally satisfy $\boldsymbol{E}_{\mathrm{out},j} \ell_j' \geq 1, \forall j \in [m]$ (i.e., $\boldsymbol{L}'$ is NTP-SVM$_\star$ feasible), it holds*

$$\mathrm{CE}_{\mathrm{in}}(\boldsymbol{L}) = \lim_{R \to \infty} \mathrm{CE}(\boldsymbol{L} + R\boldsymbol{L}').$$

*Using this, it holds for all $\boldsymbol{L} \in \mathbb{R}^{V \times m}$ that*

$$\lim_{R \to \infty} \mathrm{CE}(\boldsymbol{L}^{\mathrm{in}} + R\boldsymbol{L}') = \mathrm{CE}_{\mathrm{in}}(\boldsymbol{L}^{\mathrm{in}}) = \mathrm{CE}_{\mathrm{in},\star} = \mathcal{H} \leq \mathrm{CE}_{\mathrm{in}}(\boldsymbol{L}) < \mathrm{CE}(\boldsymbol{L}). \tag{30}$$

*Proof.* To see the first claim note that $\boldsymbol{L}' \in \mathcal{F}_{\perp}$ implies

$$\forall j \in [m], z \neq z' \in \mathcal{S}_j \; : \; \boldsymbol{L}'[j, z] = \boldsymbol{L}'[j, z'] \implies$$

$$\forall j \in [m], z \in \mathcal{S}_j \; : \; \sum_{z' \in \mathcal{S}_j} e^{-(\boldsymbol{L}[z,j] - \boldsymbol{L}[z',j] + R(\boldsymbol{L}'[j,z] - \boldsymbol{L}'[j,z']))} = \sum_{z' \in \mathcal{S}_j} e^{-(\boldsymbol{L}[z,j] - \boldsymbol{L}[z',j])}$$

and $\boldsymbol{E}_{\mathrm{out},j} \ell_j' \geq 1, \forall j \in [m]$ implies

$$\forall j \in [m], z \in \mathcal{S}_j, v \notin \mathcal{S}_j \; : \; \boldsymbol{L}'[j, z] - \boldsymbol{L}'[j, v] > 0 \implies$$

$$\forall j \in [m], z \in \mathcal{S}_j \; : \; \sum_{v \notin \mathcal{S}_j} e^{-(\boldsymbol{L}[z,j] - \boldsymbol{L}[v,j] + R(\boldsymbol{L}'[j,z] - \boldsymbol{L}'[j,v]))} \xrightarrow{R \to \infty} 0 \,.$$

The second claim applies the first claim for $\boldsymbol{L} = \boldsymbol{L}^{\mathrm{in}}$ and uses Eq. (24). $\qquad \square$

We now use the previous lemma to show that the ball-consrtained minimizer diverges as the ball constraint approaches infinity.

**Lemma 6.** *The norm of the ball-constrained minimizer diverges, that is*

$$\lim_{B \to \infty} \|\widehat{\boldsymbol{L}}_B\|_* = \infty$$

*Proof.* Suppose on the contrary that $\lim_{B \to \infty} \|\widehat{L}_B\|_* < \infty$. From Eq. (30), we have

$$\mathrm{CE}(\widehat{L}_B) > \lim_{R \to \infty} \mathrm{CE}(L^{\mathrm{in}} + R L^{\mathrm{mm}}).$$

Moreover, by triangle inequality for nuclear norm $\|L_R\|_* \leq \|L^{\mathrm{in}}\|_* + R\|L^{\mathrm{mm}}\|_*$. Thus, for any arbitrarily large $R > 0$, there exists large enough $B$ such that $\|L_R\|_* \leq B$, making $L_R := L^{\mathrm{in}} + R L^{\mathrm{mm}}$ feasible in (23). These together, contradict optimality of $\widehat{L}_B$, proving that $\lim_{B \to \infty} \|\widehat{L}_B\|_* = \infty$. Finally, recalling the norm inequality $\|\widehat{L}_B\| \geq \|\widehat{L}_B\|_*/\sqrt{V}$, it also holds that $\lim_{B \to \infty} \|\widehat{L}_B\| = \infty$. □

The last lemma constructs a sequence of logits approaching the entropy lower bound while remaining feasible in problem (23) as $B \to \infty$.

**Lemma 7.** *The objective function approaches the loss lower bound, i.e.,*

$$\lim_{B \to \infty} \mathrm{CE}(\widehat{L}_B) = \mathrm{CE}_{\mathrm{in},\star} = \mathcal{H}.$$

*Concretely, there exists logit matrix $L_R$ parameterized by $R = R(B)$ that is feasible in (23) and sastisfies:*

$$\mathrm{CE}(L_R) \leq \mathrm{CE}_{\mathrm{in},\star} + V e^{2\|L^{\mathrm{in}}\|_2} e^{-R}. \tag{31}$$

*Moreover, $\lim_{B \to \infty} R = \infty$ and $\lim_{B \to \infty} \frac{R}{\|\widehat{L}_B\|_*} = \frac{1}{\|L^{\mathrm{mm}}\|_*}$ where $\|L^{\mathrm{mm}}\|_*$ is the optimal cost of* NTP-SVM$_\star$.

*Proof.* We evaluate the loss achieved by the following candidate good point

$$L_R := L^{\mathrm{in}} + R(B) \cdot L^{\mathrm{mm}}.$$

Here, (for large enough $B > \|L^{\mathrm{in}}\|_*$) we set

$$R := R(B) = \frac{\|\widehat{L}_B\|_* - \|L^{\mathrm{in}}\|_*}{\|L^{\mathrm{mm}}\|_*}.$$

Note this is chosen so that $\|L_R\|_* \leq \|\widehat{L}_B\|_* \leq B$ towards making $L_R$ feasible in (23). Also, recall from Lemma 6 that $\|\widehat{L}_B\|_* \to \infty$ as $B \to \infty$; thus, also $R \to \infty$.

It remains to prove Eq. (31). This follows directly from the sequence of equations in (22).

□

### B.3.6 Proof of Theorem 3

We prove each statement separately.

**Statement (i).** Since for all $L$, it holds $\mathrm{CE}(L) \geq \mathcal{H}$, it suffices to prove that $\mathcal{H}$ can be asymptotically attained. Since $d \geq V$, this follows from Corollary 2. (For a more explicit proof, see Lemma 7).

**Statement (ii).** Since for all $L$ it holds $\mathrm{CE}(L) \geq \mathrm{CE}_{\mathrm{in}}(L) \geq \mathrm{CE}_{\mathrm{in},\star} = \mathcal{H}$, it follows from Statement (i) that

$$\lim_{B \to \infty} \mathrm{CE}_{\mathrm{in}}(\widehat{L}_B) = \mathrm{CE}_{\mathrm{in},\star}.$$

But $\mathrm{CE}_{\mathrm{in}}(L) = \mathrm{CE}_{\mathrm{in}}(\mathcal{P}_{\mathcal{F}}(L))$ (see Eq. (25)). Thus,

$$\lim_{B \to \infty} \mathrm{CE}_{\mathrm{in}}(\mathcal{P}_{\mathcal{F}}(\widehat{L}_B)) = \mathrm{CE}_{\mathrm{in},\star}.$$

The desired now follows ince $L^{\mathrm{in}}$ is unique minimizer of $\mathrm{CE}_{\mathrm{in}}$ on $\mathcal{T}$ by Lemma 4.

**Statement (iii).** Follows directly from Lemma 6.

**Statement (iv).** From Lemma 7, there exists $L_R$ for which the loss is close to $\mathcal{H}$ (see Eq. (31)).

Towards arriving at a contradiction, we will show that if $\widehat{L}_B$ is not in the direction of $L^{\mathrm{mm}}$, then it incurs a loss that is larger than the upper bound of $\mathrm{CE}(L_R)$ computed in Eq. (31).

To do this, assuming the statement of the theorem is not true, we will lower bound

$$\mathrm{CE}(\widehat{L}_B) - \mathcal{H} = \sum_{j \in [m]} \hat{\pi}_j \sum_{z \in \mathcal{S}_j} \hat{p}_{j,z} \log \left( \hat{p}_{j,z} \left( \sum_{z' \in \mathcal{S}_j} e^{-(\widehat{L}_B[j,z] - \widehat{L}_B[j,z'])} + \sum_{v \notin \mathcal{S}_j} e^{-(\widehat{L}_B[j,z] - \widehat{L}_B[j,v])} \right) \right).$$
(32)

By our assumption, there exists $\epsilon > 0$, such that there exists arbitrarily large $B$ satisfying:

$$\left\| \frac{\|L^{\mathrm{mm}}\|_*}{\|\widehat{L}_B\|_*} \widehat{L}_B - L^{\mathrm{mm}} \right\| > \epsilon,$$
(33)

for all minimizers $L^{\mathrm{mm}}$ of NTP-SVM$_\star$. Define

$$\widehat{L} = \frac{1}{R'(B)} (\widehat{L}_B - L^{\mathrm{in}}),$$

where, $R' := R'(B) > 0$ is chosen so that $\|\widehat{L}\|_* < \|L^{\mathrm{mm}}\|_*$. Concretely, this can be ensured by setting (to see this use the triangle inequality for nuclear norm):

$$R' = \frac{(\|\widehat{L}_B\|_* + \|L^{\mathrm{in}}\|_* + \zeta)}{\|L^{\mathrm{mm}}\|_*},$$

for some $\zeta > 0$. Recall that $\|\widehat{L}_B\|_* \to \infty$ as $B \to \infty$. On the other hand $\|L^{\mathrm{in}}\|_*$ is not dependent on $B$. Thus, in the large limit $B \to \infty$, it holds $\left| \frac{\|L^{\mathrm{mm}}\|_*}{\|\widehat{L}_B\|_*} - \frac{1}{R'} \right| \to 0$ and $\frac{\|L^{\mathrm{in}}\|_*}{\|\widehat{L}_B\|_*} \to 0$. These combined with (33) show that we can always choose $B$ large enough so that Eq. (33) guarantees, for some $\epsilon' > 0$, that

$$\|\widehat{L} - L^{\mathrm{mm}}\| \geq \epsilon'.$$

But $\widehat{L}$ achieves the optimal cost of NTP-SVM$_\star$, since $\|\widehat{L}\|_* < \|L^{\mathrm{mm}}\|_*$. Thus, there exists $\delta \in (0,1)$ and $j \in [m]$ such that at least one of the following is true

(i) $\exists z$ and $z' \neq z \in \mathcal{S}_j$ such that

$$|\widehat{L}[j,z] - \widehat{L}[j,z']| \geq \delta,$$
(34)

(ii) $\exists z \in \mathcal{S}_j, v \notin \mathcal{S}_j$ such that

$$\widehat{L}[j,z] - \widehat{L}[j,v] \leq 1 - \delta.$$
(35)

*Case (i):* Without loss of generality $(e_z - e_{z'})^\top \widehat{\ell}_j \leq -\delta$ (otherwise, flip $z, z'$). Thus, ignoring all but one term in (32) gives

$$\mathrm{CE}(\widehat{L}_B) - \mathcal{H} \geq \hat{\pi}_j \hat{p}_{j,z} \log \left( \hat{p}_{j,z} e^{-(\widehat{L}_B[j,z] - \widehat{L}_B[j,z'])} \right).$$
(36)

But,

$$\widehat{L}_B[j,z] - \widehat{L}_B[j,z'] = R'(\widehat{L}[j,z] - \widehat{L}[j,z']) + (L^{\mathrm{in}}[j,z] - L^{\mathrm{in}}[j,z']) \leq -R'\delta + 2\|L^{\mathrm{in}}\|_2 \quad (37)$$

Put in (32) and using $\hat{p}_{j,z} \geq 1/V$, $\hat{\pi}_j \geq 1/m$ shows

$$\mathrm{CE}(\widehat{L}_B) \geq \mathcal{H} + \frac{1}{mV} \log \left( \frac{1}{V} e^{R'\delta} \exp(-2\|L^{\mathrm{in}}\|_2) \right) \geq \mathcal{H} + \frac{1}{mV} \log \left( \frac{e^{R'\delta}}{V \exp(2\|L^{\mathrm{in}}\|_2)} \right)$$

Compare this with (31), it is clear that as $B \to \infty$, so that $R' \to \infty$, it holds $\frac{1}{mV} \log \left( \frac{e^{R'\delta}}{V \exp(2\|L^{\mathrm{in}}\|_2)} \right) > 1 > V \exp(2\|L^{\mathrm{in}}\|_2) e^{-R}$. Thus, $\mathrm{CE}(\widehat{L}_B) > \mathrm{CE}(L_R^{\mathrm{in}})$, a contradiction.

_Case (ii):_ We can assume $E_{\text{in},j}\widehat{\ell}_j = 0$ for all $j \in [m]$, since otherwise we are in Case (i). Now, again ignoring all but the $(j,z)$ term in the CE loss for which (35) holds for some $v \notin \mathcal{S}_j$, we find

$$\text{CE}(\widehat{L}_B) - \mathcal{H} \geq \hat{\pi}_j \hat{p}_{j,z} \log \Big( p_{j,z} \Big( \sum_{z' \in \mathcal{S}_j} e^{-(\widehat{L}_B[j,z] - \widehat{L}_B[j,z'])} + e^{(\widehat{L}_B[j,v] - \widehat{L}_B[j,z])} \Big) \Big).$$

On the one hand, using $E_{\text{in},j}\widehat{\ell}_j = 0$, we have

$$\sum_{z' \in \mathcal{S}_j} e^{-(\widehat{L}_B[j,z] - \widehat{L}_B[j,z'])} = \sum_{z' \in \mathcal{S}_j} e^{-(L^{\text{in}}[j,z] - L^{\text{in}}[j,z])} = \sum_{z' \in \mathcal{S}_j} \frac{\hat{p}_{j,z'}}{\hat{p}_{j,z}} = \frac{1}{\hat{p}_{j,z'}}.$$

On the other hand, by (35):

$$e^{\widehat{L}_B[j,v] - \widehat{L}_B[j,z]} \geq e^{-R'(1-\delta)} e^{L^{\text{in}}[j,v] - L^{\text{in}}[j,z]} \geq e^{-R'(1-\delta)} \exp(-2\|L^{\text{in}}\|_2),$$

Putting the above together yield:

$$\text{CE}(\widehat{L}_B) - \mathcal{H} \geq \hat{\pi}_j \hat{p}_{j,z} \log \Big( 1 + \frac{e^{-R'(1-\delta)} \exp(-2\|L^{\text{in}}\|_2)}{V} \Big) \geq \frac{e^{-R'(1-\delta)}}{mV^2 \exp(2\|L^{\text{in}}\|_2) + mV}.$$

where the second inequality uses $\log(1 + x) \geq \frac{x}{1+x}, x > 0$.

Compare this with (31). As $B \to \infty$, and noting that $R, R'$ grow at the same rate, it holds $\frac{e^{-R'(1-\delta)}}{mV^2 \exp(2\|L^{\text{in}}\|_2) + mV} > V \exp(2\|L^{\text{in}}\|_2) e^{-R}$. Thus, $\text{CE}(\widehat{L}_B) > \text{CE}(L_R)$, a contradiction.

In either case, we arrive at a contradiction, which completes the proof.

## C   On the solution of NTP-SVM$_\star$

To better understand how the SVD factorization informs the geometry, we investigate further how the structure of $L^{\text{mm}}$ (solution to NTP-SVM$_\star$) depends on the support matrix $S$.

### C.1   Subspace-collapse

**Proposition 2** (Subspace collapse). *Assume NTP-SVM$_\star$ is feasible. There exists minimizer $H = [h_1, \ldots, h_m]$ of NTP-UFM such that for all contexts $j, j' \in [m]$ with same support set $\mathcal{S}_j = \mathcal{S}_{j'}$, their embeddings are same in direction in the limit of vanishing regularization, i.e., $\lim_{\lambda \to 0} \left\| \overline{h_j} - \overline{h_{j'}} \right\| = 0$.*

This is consistent with our empirical observations in Figs. 1 and 2: As the loss approaches its lower bound, context embeddings with the same support set converge in the same direction and exhibit maximum correlation. This is indicated by the bright diagonal blocks in CORR($H$) heatmaps.

The proposition can be proved by inspecting NTP-SVM$_\star$: Since the affine constraints (9) for each sample $j \in [m]$ depend solely on the support set $\mathcal{S}_j$, it can be shown that there exists a minimizer $L^{\text{mm}} = [\ell_1^{\text{mm}}, \cdots, \ell_m^{\text{mm}}]$ such that $\ell_j^{\text{mm}} = \ell_{j'}^{\text{mm}}$ whenever $\mathcal{S}_j = \mathcal{S}_{j'}$. This implies that the corresponding columns $j, j'$ of $V^\top$ are identical. As a result, by Cor. 1, $\lim_{\lambda \to 0} \left\| \overline{h_j} - \overline{h_{j'}} \right\| = 0$.

It is important to reiterate that Prop. 2 does not prevent the minimizer $L_\lambda = W_\lambda H_\lambda$ from accurately recovering the distinct soft-label values $\hat{p}_j, \hat{p}_{j'}$ and achieving the empirical entropy lower-bound $\mathcal{H}$ as $\lambda \to 0$: The collapse occurs on subspace $\mathcal{F}_\perp$, orthogonal to the data subspace $\mathcal{F}$ on which the logit matrix satisfies the log-odds constraints indicated in Thm. 2; See Secs. D.1 and 5 for numerical verifications.

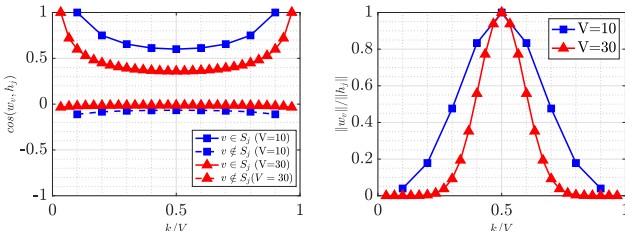

**Figure 4:** Geometry properties illustration for Proposition 3. Shown two values of $V$ for varying values of $k \in [V-1]$. (Left) angles between context and word embeddings. (Right) normalized norm ratio of word to context embeddings.

## C.2  Special case: support sets of equal sizes

To gain insights, we start with an idealized symmetric setting.

**Proposition 3.** *Fix $k \in [V-1]$ and suppose $\boldsymbol{S}$ contains all $m = \binom{V}{k}$ support sets of size $k$. Then,*

*(i) The logit matrix takes the form $\boldsymbol{L}^{\mathrm{mm}} = (\mathbb{I}_V - \frac{1}{V}\mathbb{1}\mathbb{1}^\top)\boldsymbol{S} \triangleq \widetilde{\boldsymbol{S}}$.*

*(ii) Word embeddings form equiangular tight frame (ETF) being equinorm and maximally separated.*

*(iii) Context embeddings are equinorm and the embedding $\boldsymbol{h}_j$ of context $j$ is co-linear to $\sum_{z \in \mathcal{S}_j} \boldsymbol{w}_z$.*

In the symmetric setting of the proposition, we can analytically solve NTP-SVM$_\star$ giving $\boldsymbol{L}^{\mathrm{mm}}$ as expressed in the statement (i). In fact, it is possible to obtain an explicit characterization of the SVD of $\boldsymbol{L}^{\mathrm{mm}}$, which enables the precise definition of embeddings' geometries in statements (ii) and (iii). This explicit characterization leads to closed-form formulas for the angles and norms of embeddings, detailed in Prop. 4.

**Proposition 4.** *Assume the setting of Proposition 3. The geometry of context and word embeddings is described by the following relations:*

$$\forall v \neq v' \in [V] : \cos(\boldsymbol{w}_v, \boldsymbol{w}_{v'}) = \frac{-1}{V-1} \quad and \quad \|\boldsymbol{w}_v\| = \|\boldsymbol{w}_{v'}\| \tag{38a}$$

$$\forall j \neq j' \in [m] : \cos(\boldsymbol{h}_j, \boldsymbol{h}_{j'}) = \frac{|\mathcal{S}_j \cap \mathcal{S}_{j'}| - \frac{k^2}{V}}{k - \frac{k^2}{V}} \quad and \quad \|\boldsymbol{h}_j\| = \|\boldsymbol{h}_{j'}\| \tag{38b}$$

$$\forall j \in [m], v \in [V] : \cos(\boldsymbol{w}_v, \boldsymbol{h}_j) = \begin{cases} \sqrt{\frac{V-1}{k(V-k)}} & v \in \mathcal{S}_j \\ \frac{-1}{\sqrt{k(V-k)(V-1)}} & v \notin \mathcal{S}_j \end{cases} \tag{38c}$$

$$\forall j \in [m], v \in [V] : \frac{\|\boldsymbol{w}_v\|^2}{\|\boldsymbol{h}_j\|^2} = \frac{(V-1)\binom{V-2}{k-1}}{k(V-k)}. \tag{38d}$$

For $k = 1$ the embeddings recover the ETF geometry that has been previously shown for the setting of one-hot classification (Papyan et al., 2020). For general $k > 1$, word embeddings continue forming an ETF, but the geometry of context embeddings changes although they remain equinorm. Embedding $\boldsymbol{h}_j$ forms the same *acute* angle with all in-support word vectors $\boldsymbol{w}_v, v \in \mathcal{S}_j$ and the same *obtuse* angle with all out-of-support word vectors $\boldsymbol{w}_v, v \notin \mathcal{S}_j$. Additionally, for $k > 1$, norms of word embeddings become larger than norms of context embeddings. See also Fig. 4 for a visualization of these properties. We defer the proof of these two propositions to App. C.5.3.

## C.3  A simple candidate solution

Achieving an analytic expression for $\boldsymbol{L}^{\mathrm{mm}}$, as was possible in Prop. 3, might not always be possible. This challenge primarily arises from the combinatorial complexity of the

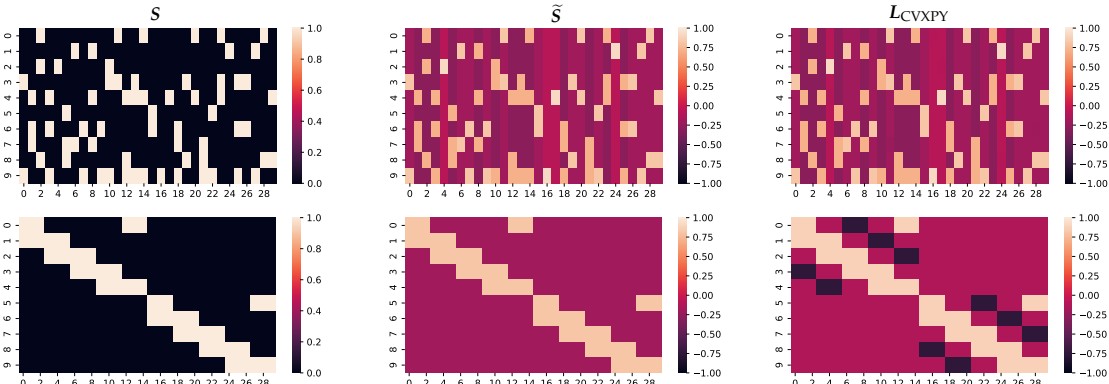

**Figure 5:** Comparison between the solution found by CVXPY for NTP-SVM$_\star$ and $\widetilde{S}$ defined in App. C for two realizations of the support set matrix $S$. See text for details.

constraints that adhere to the sparsity pattern of $S$. To mitigate this issue and potentially circumvent the need to solve the semidefinite program in NTP-SVM$_\star$—which becomes computationally intensive for large $V$ and $m$—we introduce a strategy that allows for a numerical verification of whether $\widetilde{S}$ of Prop. 3 solves NTP-SVM$_\star$.

$\widetilde{S}$ readily satisfies the feasibility conditions for NTP-SVM$_\star$, meeting all inequality constraints with equality. Below, we further introduce a dual-certificate condition that, when fulfilled by $\widetilde{S}$, ensures optimality.

**Proposition 5.** *Let $\widetilde{S} = U\Sigma V^\top$ denote the SVD of $\widetilde{S} = (\mathbb{I}_V - \frac{1}{V}\mathbb{1}\mathbb{1}^\top)S$. Define, $A := UV^\top$. If $\forall j \in [m]$ and all $v \notin \mathcal{S}_j$, it holds that $A[v,j] < 0$, then $\widetilde{S}$ solves NTP-SVM$_\star$.*

Verifying the certificate's conditions only requires an SVD of $\widetilde{S}$. In situations where these SVD factors can be analytically determined, the certificate also enables a formal proof of optimality (e.g. Prop. 3).

In more complex settings than that of Prop. 3, whether $\widetilde{S}$ satisfies the dual certificate in Prop. 5 or not, and thus its optimality, depends on the sparsity pattern of $S$. Our empirical evaluations show that $\widetilde{S}$ successfully solves NTP-SVM$_\star$ in numerous realizations of $S$ we have examined. However, this is not the case in all realizations of the support set structure.

In Fig. 5, we present an example of two realizations of $S$, where the condition in Prop. 5 is met in one instance but not in the other. The second and third column of the figure display the heatmap of $\widetilde{S}$ and the optimal solution of NTP-SVM$_\star$ as determined by CVXPY (Grant & Boyd, 2014) for the support sets $S$ shown in the first column. Notably, although $\widetilde{S}$ is not optimal in the second row, it provides a close and simple approximation of the optimal $L^{mm}$ which almost captures the overall structure of the optimal solution found by CVXPY. Additionally, in Fig. 6, we show similar heatmaps of the optimal logits for the setup of the Simplified TinyStories dataset in Sec. 5. Once again, while $\widetilde{S}$ is not necessarily optimal, it serves as a close proxy for the optimal solution. Note that in the case of one-hot classification under STEP imbalanced data, $\widetilde{S}$ is previously shown to be the unique minimizer of NTP-SVM$_\star$ (Thrampoulidis et al., 2022).

Although not necessarily optimal for all configurations of the support set, we conjecture that $\widetilde{S}$, gives an approximation with almost similar geometric properties of the optimal solution $L^{mm}$. This conjecture motivates a more efficient proxy for predicting the implicit geometry in the next section.

## C.4 A proxy for NTP-SVM$_\star$ solutions

Considering $\widetilde{S} = (\mathbb{I}_V - \frac{1}{V}\mathbb{1}\mathbb{1}^\top)S$, i.e., the column-wise centered support set, as a close proxy of the optimal solution of NTP-SVM$_\star$, we provide a connection between the support sets

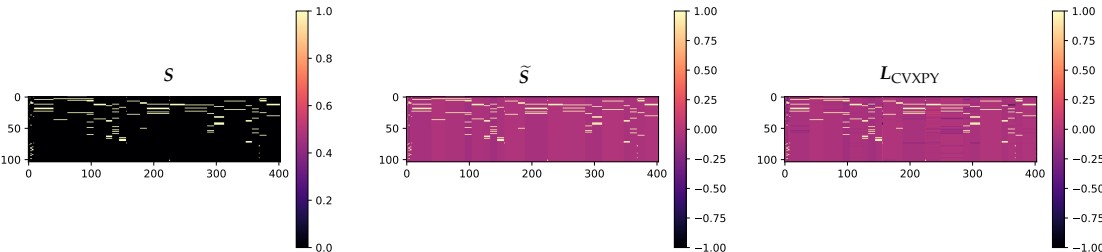

**Figure 6:** Same as Fig. 5 for the support set configuration in the `Simplified TinyStories` of Sec. 5.

and embeddings geometry. Suppose $L^{\mathrm{mm}} \approx \widetilde{S}$. Then, by Thm. 1,

$$W^{\mathrm{mm}}W^{\mathrm{mm}\top} = (L^{\mathrm{mm}}L^{\mathrm{mm}\top})^{\frac{1}{2}} \approx (\widetilde{S}\widetilde{S}^{\top})^{\frac{1}{2}},$$

$$H^{\mathrm{mm}\top}H^{\mathrm{mm}} = (L^{\mathrm{mm}\top}L^{\mathrm{mm}})^{\frac{1}{2}} \approx (\widetilde{S}^{\top}\widetilde{S})^{\frac{1}{2}}. \tag{39}$$

Computing the matrix square roots in Eq. (39) is as expensive as an SVD decomposition which can be prohibitive for large matrices. Instead, we suggest the following more computationally efficient proxies for estimating the directional component of the implicit geometries:

$$W^{\mathrm{mm}}W^{\mathrm{mm}\top} \approx \widetilde{S}\widetilde{S}^{\top} = (\mathbb{I}_V - \frac{1}{V}\mathbb{1}_V\mathbb{1}_V^{\top})SS^{\top}(\mathbb{I}_V - \frac{1}{V}\mathbb{1}_V\mathbb{1}_V^{\top}),$$

$$H^{\mathrm{mm}\top}H^{\mathrm{mm}} \approx \widetilde{S}^{\top}\widetilde{S} = S^{\top}(\mathbb{I}_V - \frac{1}{V}\mathbb{1}_V\mathbb{1}_V^{\top})S, \tag{40}$$

This leads to Proxy **(P)** introduced in Sec. 1.2.

Ignoring for simplicity the projection to the $\mathbb{I}_V - \frac{1}{V}\mathbb{1}_V\mathbb{1}_V^{\top}$ subspace, Eq. (40) gives a simple explanation for similarity between embeddings: For given context $j \in [m]$, the corresponding embedding has higher correlation with contexts $j'$ whose support set intersection $|\mathcal{S}_j \cap \mathcal{S}_{j'}|$ is larger. Similarly, word embeddings are closer if the respective words appear together in more support sets: word embedding $w_z$ is more similar to $w_{z'}$ when $|\{j : w_z, w_{z'} \in \mathcal{S}_j, j \in [m]\}|$ is larger. The projection then fixes the bias of the parameters: since ridge-regularization is strongly convex and softmax is invariant to shift, any minimizer of NTP-UFM satisfies $\mathbb{1}_V^{\top}W = 0$ ( see Lemma 2).

### C.5 Proofs

#### C.5.1 Proof of Proposition 2

We will prove that for $j, j' \in [m]$ with same support set $\mathcal{S}_j = \mathcal{S}_{j'}$ the $j$-th and $j'$-th columns of $L^{\mathrm{mm}} = [\boldsymbol{\ell}_1, \ldots, \boldsymbol{\ell}_m]$ are same. For the sake of contradiction, suppose that this is not the cases, i.e. $\boldsymbol{\ell}_j \neq \boldsymbol{\ell}_{j'}$. Consider the following three candidate solutions of NTP-SVM$_{\star}$:

$$L^{\mathrm{mm}} = [\boldsymbol{\ell}_1, \ldots, \boldsymbol{\ell}_j, \ldots, \boldsymbol{\ell}_{j'}, \ldots \boldsymbol{\ell}_m]$$

$$L_{\mathrm{flip}}^{\mathrm{mm}} = [\boldsymbol{\ell}_1, \ldots, \boldsymbol{\ell}_{j'}, \ldots, \boldsymbol{\ell}_j, \ldots \boldsymbol{\ell}_m]$$

$$L_{\mathrm{avg}}^{\mathrm{mm}} = \left[\boldsymbol{\ell}_1, \ldots, \frac{\boldsymbol{\ell}_j + \boldsymbol{\ell}_{j'}}{2}, \ldots, \frac{\boldsymbol{\ell}_j + \boldsymbol{\ell}_{j'}}{2}, \ldots \boldsymbol{\ell}_m\right].$$

By assumption $L^{\mathrm{mm}}$ is optimal in NTP-SVM$_{\star}$. We now show that the other two matrices are also optimal.

Firstly, note that $L_{\mathrm{flip}}^{\mathrm{mm}}$ is feasible in NTP-SVM$_{\star}$ because: (i) the affine constraints only depend on the support sets, which are same for $j, j's$, (ii) rank $\left(L_{\mathrm{flip}}^{\mathrm{mm}}\right) = \mathrm{rank}\left(L^{\mathrm{mm}}\right) \leq d$ since permuting columns is rank-preserving operation. Moreover, permuting columns also

preserves nuclear-norm of a matrix. To see this note that $L_{\text{flip}}^{\text{mm}} = L^{\text{mm}} P = U \Sigma (P^\top V)^\top$ for some permutation matrix $P \in \mathbb{R}^{m \times m}$. Since $V$ is partial orthonormal, the same is true for $P^\top V$. Thus, $\|L_{\text{flip}}^{\text{mm}}\|_* = \text{tr}(\Sigma) = \|L^{\text{mm}}\|_*$, which proves optimality of $\text{rank}\left(L_{\text{flip}}^{\text{mm}}\right)$.

Similarly, we can argue that $L_{\text{avg}}^{\text{mm}}$ is feasible in NTP-SVM$_\star$ because: (i) the affine constraints only depend on the support sets, which are same for $j, j's$, (ii) $\text{rank}\left(L_{\text{avg}}^{\text{mm}}\right) = \text{rank}\left(L^{\text{mm}}\right) \leq d$ since adding columns and multiplying them by scalars are rank-preserving operations. But now, from convexity of the objective function, and noting that $L_{\text{avg}}^{\text{mm}} = \frac{1}{2}\left(L^{\text{mm}} + L_{\text{flip}}^{\text{mm}}\right)$ we have

$$\|L_{\text{avg}}^{\text{mm}}\|_* \leq \frac{1}{2}\|L^{\text{mm}}\|_* + \frac{1}{2}\|L_{\text{flip}}^{\text{mm}}\|_* = \|L^{\text{mm}}\|_* \,.$$

Thus, $L_{\text{avg}}^{\text{mm}}$ is also optimal.

Now, from Lemma 1, we know that for some partial orthonormal matrix $\mathbf{R}$, in the limit of vanishing regularization: $H_\lambda \propto \mathbf{R}^\top \Sigma^{\frac{1}{2}} V^\top$ where $V, \Sigma$ are SVD factors of $L_{\text{avg}}^{\text{mm}}$. Since the $j, j'$ columns of $L_{\text{avg}}^{\text{mm}}$ are the same by construction, the same is true for the $j, j'$ columns of $V^\top$. This proves the desired.

### C.5.2 Proof of Proposition 5

To prove the proposition, we first derive the dual of NTP-SVM$_\star$.

We begin with a more convenient, but equivalent, formulation of NTP-SVM$_\star$. One way to do this is by recalling that NTP-SVM$_\star$ is a convex relaxation to

$$\min_{W,H} \frac{1}{2}\|W\|_F^2 + \frac{1}{2}\|H\|_F^2, \quad \text{s.t.} \quad \forall j \in [m], z_j \neq z \in \mathcal{S}_j \ : \ (e_{z_j} - e_{z'})^\top W h_j = 0$$
$$\forall j \in [m], v \notin \mathcal{S}_j \ : \ (e_{z_j} - e_v)^\top W h_j \geq 1 \,.$$

where, for each sample $j$, we choose an anchor label $z_j \in \mathcal{S}_j$. We can now relax this in terms of the matrix $X = \begin{bmatrix} W \\ H^\top \end{bmatrix} \begin{bmatrix} W^\top & H \end{bmatrix}$. (Note that $L$ essentially corresponds to the off-diagonal blocks of $X$). This gives us the following reformulation of NTP-SVM$_\star$. (Formally, it is easy to show that (41) is the dual of the dual of NTP-SVM$_\star$. This result about nuclear minimization is classic (e.g., Recht et al. (2010)) and we omit the details.):

$$\min_{X \succcurlyeq 0} \frac{1}{2}\text{tr}(X) \quad \text{s.t.} \quad \forall j \in [m], z_j \neq z \in \mathcal{S}_j \ : \ X[z_j, V + j] - X[z', V + j] = 0 \qquad (41)$$
$$\forall j \in [m], v \notin \mathcal{S}_j \ : \ X[z_j, V + j] - X[v, V + j] \geq 1 \,.$$

Define $A_{V \times m}$ such that $A[z, j] = -\lambda_{j,z}$, $z \neq z_j$ and $A[z_j, j] = \sum_{z \neq z_j} \lambda_{j,z}$. ($\mathbf{1}_V^\top A = 0$).

The dual problem is

$$\max_A \min_{X \succcurlyeq 0} \sum_{j \in [m]} \sum_{v \notin \mathcal{S}_j} \lambda_{j,v} + \frac{1}{2}\text{tr}\left(\begin{bmatrix} \mathbb{I} & -A \\ -A^\top & \mathbb{I} \end{bmatrix} X\right) \quad \text{s.t.} \quad \mathbf{1}_V^\top A = 0 \quad A[v, j] \leq 0, v \notin \mathcal{S}_j$$
$$= \max_A \sum_{j \in [m]} \sum_{v \notin \mathcal{S}_j} \lambda_{j,v} \quad \text{s.t.} \quad \mathbf{1}_V^\top A = 0 \quad A[v, j] \leq 0, v \notin \mathcal{S}_j, \quad \|A\|_2 \leq 1$$
$$= \max_A \ \text{tr}(A^\top \widetilde{S}) \quad \text{s.t.} \quad \mathbf{1}_V^\top A = 0 \quad A[v, j] \leq 0, v \notin \mathcal{S}_j, \quad \|A\|_2 \leq 1$$

where here $\widetilde{S}$ is defined in Prop. 3. The last line above holds since,

$$\sum_{j\in[m]}\sum_{v\notin\mathcal{S}_j}\lambda_{j,v} = \sum_{j\in[m]}\sum_{v\notin\mathcal{S}_j}\lambda_{j,v} + \sum_{j\in[m]}\sum_{\substack{v\in\mathcal{S}_j\\v\neq z_j}}\lambda_{j,v} - \sum_{j\in[m]}\sum_{\substack{v\in\mathcal{S}_j\\v\neq z_j}}\lambda_{j,v}$$

$$= \sum_{j\in[m]}\sum_{v\neq z_j}\lambda_{j,v} - \sum_{j\in[m]}\sum_{\substack{v\in\mathcal{S}_j\\v\neq z_j}}\lambda_{j,v}$$

$$= \sum_{j\in[m]}(\frac{1-|\mathcal{S}_j|}{V} + \frac{|\mathcal{S}_j|}{V})\sum_{v\neq z_j}\lambda_{j,v} - \sum_{j\in[m]}\sum_{\substack{v\in\mathcal{S}_j\\v\neq z_j}}\lambda_{j,v}$$

$$= \sum_{j\in[m]}\frac{1-|\mathcal{S}_j|}{V}A[z_j,j] - \sum_{j\in[m]}\frac{|\mathcal{S}_j|}{V}\sum_{v\neq z_j}A[v,j] + \sum_{j\in[m]}\sum_{\substack{v\in\mathcal{S}_j\\v\neq z_j}}A[v,j]$$

$$= \sum_{j\in[m]}\widetilde{S}[z_j,j]A[z_j,j] + \sum_{j\in[m]}\sum_{v\notin\mathcal{S}_j}\widetilde{S}[v,j]A[v,j] + \sum_{j\in[m]}\sum_{\substack{v\in\mathcal{S}_j\\v\neq z_j}}\widetilde{S}[v,j]A[v,j]$$

$$= \sum_{j\in[m]}\sum_{v\in[V]}\widetilde{S}[v,j]A[v,j].$$

We remark the following about the dual problem derived above. First, because $\widetilde{S}$ is feasible in NTP-SVM$_\star$ and constraints are linear, Slater's conditions hold, thus, we have strong duality. Second, by complementary slackness, if the nonegativity conditions $A[v,j] \leq 0, v \notin \mathcal{S}_j$ are strict, then, at optimality, $L$ satisfies the inequality constraints (10) with equality.

Now, regarding solving the dual, observe that if we remove the constraint $A[v,j] \leq 0, v \notin \mathcal{S}_j$, then the solution to the dual is simply $A = UV^\top$ where $U\Sigma V^\top$ is the SVD of $\widetilde{S}$. In fact, it is not hard to show that this is the unique solution of the relaxed dual; see for example (Thrampoulidis et al., 2022, Lemma C.1) for an analogous result. Therefore, if $UV^\top$ satisfies the condition of the proposition, then it is clearly the unique optimal solution to the dual problem.

From this and complementary slackness discussed above, it must be that any minimizer of NTP-SVM$_\star$ must satisfy the inequality constraints with equalities. Formally, the optimal solution $\hat{L}$ satisfies simultaneously:

$$\hat{L}^\top \mathbb{1}_V = 0_m \quad \text{and} \quad \forall j \in [m] : E_{\text{in},j}\hat{\ell}_j = \mathbb{1}_{S_j}, \quad E_{\text{out},j}\hat{\ell}_j = 0_{V-1-S_j}.$$

Note that this is a system of $m + \sum_{j\in[m]}(S_j + V - 1 - S_j) = Vm$ linear equations. It can also be easily verified that these equations are linearly independent. Thus, their unique solution is $\widetilde{S}$. This completes the proof of the proposition.

### C.5.3 Proof of Propositions 3 and 4

At the heart of proving both Propositions 3 and 4 is the following result, which determines the SVD factors $U$, $\Sigma$, and $V$ of $\widetilde{S} = (\mathbb{I}_V - \frac{1}{V}\mathbb{1}\mathbb{1}^\top)S$. This shows immediately that word embeddings form an ETF. It also forms the basis for computing the geometry of context embeddings. We show this after proving the lemma.

**Lemma 8.** *Fix any $k \in [V]$ and suppose $S$ contains all $\binom{V}{k}$ support sets of size $k$, i.e. $m = \binom{V}{k}$. Let $P \in \mathbb{R}^{V\times(V-1)}$ be orthonormal basis of the subspace orthogonal to $\mathbb{1}_V$. Then, the $U, \Sigma$ factors of the SVD of $\widetilde{S}$ are: $U = P$ and $\Sigma = \sqrt{\binom{V-2}{k-1}}\mathbb{I}_{V-1}$. Thus,*

$$U\Sigma U^\top \propto \left(\mathbb{I}_V - \frac{1}{V}\mathbb{1}\mathbb{1}^\top\right).$$

*Moreover, the matrix $\boldsymbol{V} \in \mathbb{R}^{m \times (V-1)}$ of right-singular vectors is such that its rows $\boldsymbol{v}_j, j \in [m]$ can be expressed as follows with respect to the rows $\boldsymbol{u}_z, z \in [V]$ of $\boldsymbol{U}$:*

$$\boldsymbol{v}_j = \frac{1}{\sqrt{\binom{V-2}{k-1}}} \sum_{z \in \mathcal{S}_j} \boldsymbol{u}_z \quad j \in [m].$$

*Proof.* We will compute explicitly $\widetilde{S}\widetilde{S}^\top$. Start with computing $SS^\top$. For diagonal elements, the dot product is between a row of $S$ and itself, counts the number of 1s in that row. Since each element is included in $\binom{V-1}{k-1}$ support sets (choosing the remaining $k-1$ elements from the other $V-1$ elements), each diagonal entry of $SS^\top$ is $\binom{V-1}{k-1}$. For off-diagonal elements, the dot product counts the number of support sets in which both elements of the corresponding rows appear. This requires choosing the remaining $k-2$ elements from $V-2$ elements, giving $\binom{V-2}{k-2}$ for each off-diagonal entry of $SS'$. Overall, after algebraic simplification, we find that

$$SS^\top = \binom{V-2}{k-1} \left( \mathbb{I}_V + \frac{k-1}{V-k} \mathbb{1}\mathbb{1}^\top \right).$$

We may now use the closed form of $\widetilde{S}$ and few more algebra simplifications to find that

$$\widetilde{S}\widetilde{S}^\top = \binom{V-2}{k-1} \left( \mathbb{I}_V - \frac{1}{V} \mathbb{1}\mathbb{1}^\top \right).$$

Let $\boldsymbol{P} \in \mathbb{R}^{V \times (V-1)}$ denote an orthonormal basis of the subspace orthogonal to $\mathbb{1}_V$. Since $\widetilde{S}\widetilde{S}^\top = \binom{V-2}{k-1} \boldsymbol{P}\boldsymbol{P}^\top$ and $\widetilde{S}\widetilde{S}^\top = \boldsymbol{U}\boldsymbol{\Sigma}^2\boldsymbol{U}^\top$, we have shown that $\boldsymbol{U} = \boldsymbol{P}$ and $\boldsymbol{\Sigma} = \sqrt{\binom{V-2}{k-1}} \mathbb{I}_V$. $\square$

Using $\mathbb{1}^\top S = k\mathbb{1}_m^\top$, we can compute

$$\widetilde{S}^\top\widetilde{S} = S^\top S - \frac{1}{V}(S^\top \mathbb{1}_V)(S^\top \mathbb{1}_V)^\top = S^\top S - \frac{k^2}{V} \mathbb{1}_m\mathbb{1}_m^\top.$$

From this and the fact that $\mathrm{diag}(S^\top S) = k\mathbb{I}_m$, and recalling that $\widetilde{S}^\top\widetilde{S} = \boldsymbol{V}\boldsymbol{\Sigma}^2\boldsymbol{V}^\top$, we find that all context embeddings are equinorm with squared norm (i.e. diagonal entries of $\boldsymbol{V}\boldsymbol{\Sigma}\boldsymbol{V}^\top$) equal to

$$\|\boldsymbol{h}_j\|^2 = \frac{k - \frac{k^2}{V}}{\sqrt{\binom{V-2}{k-1}}}, \quad \forall j \in [m].$$

Recall also that the diagonal entries of $\boldsymbol{U}\boldsymbol{\Sigma}\boldsymbol{U}^\top$ are

$$\|\boldsymbol{w}_v\|^2 = \left(1 - \frac{1}{V}\right)\sqrt{\binom{V-2}{k-1}}, \quad \forall v \in [V].$$

From the off-diagonal entries of $\widetilde{S}^\top\widetilde{S}$, we can infer the angles between different context embeddings. Let $j \neq j' \in [m]$, then the $(j, j')$ of $\widetilde{S}^\top\widetilde{S}$ is $|\mathcal{S}_j \cap \mathcal{S}_{j'}| - \frac{k^2}{V}$. Thus, the $(j, j')$ off-diagonal entries of $\boldsymbol{V}\boldsymbol{\Sigma}\boldsymbol{V}^\top$ are

$$\boldsymbol{h}_j^\top \boldsymbol{h}_{j'} = \frac{|\mathcal{S}_j \cap \mathcal{S}_{j'}| - \frac{k^2}{V}}{\sqrt{\binom{V-2}{k-1}}} \implies \cos(\boldsymbol{h}_j, \boldsymbol{h}_{j'}) = \frac{|\mathcal{S}_j \cap \mathcal{S}_{j'}| - \frac{k^2}{V}}{k - \frac{k^2}{V}} \quad j \neq j' \in [m].$$

Recalling that $\boldsymbol{W}\boldsymbol{H}^\top = \boldsymbol{L}^{\mathrm{mm}}$, we may now compute the angle between word and context embeddings as follows:

$$\cos(\boldsymbol{w}_v, \boldsymbol{h}_j) = \begin{cases} \sqrt{\frac{V-1}{k(V-k)}} & v \in \mathcal{S}_j \\ \frac{-1}{\sqrt{k(V-k)(V-1)}} & v \notin \mathcal{S}_j \end{cases}. \tag{42}$$

This completes the proof of Proposition 4 and the proof of statements (ii) and (iii) of Proposition 3.

It remains to prove statement (i) of Proposition 3, i.e. proving that $\widetilde{S} = (\mathbb{I}_V - \frac{1}{V}\mathbb{1}\mathbb{1}^\top)S$ is the unique minimizer of NTP-SVM$_\star$.

To prove this, we appeal to Proposition 5. First, it is straightforward checking that $L^{\mathrm{mm}}$ satisfies the SVM constraints (in fact, with equality). Thus, from Proposition 5, it suffices that the matrix $A := UV^\top$ satisfies $A[v, j] < 0$ for all $v \notin \mathcal{S}_j$ and all $j \in [m]$. But, in our case $A = UV^\top = \binom{V-2}{k-1}^{-\frac{1}{2}}\widetilde{S}$ because $\Sigma = \mathbb{I}_{V-1}$. Thus, the desired condition holds for $A$ since $\widetilde{S}[v, j] = -k/V < 0$ for all $j \notin \mathcal{S}_j$ and $j \in [m]$.

## D  Numerical experiments

### D.1  Numerical simulation: NTP-UFM

Here, our aim is to confirm, through numerical experiments, that the solution derived from (stochastic) gradient descent when optimizing NTP-UFM is consistent with the analysis presented in Sec. 4. We train NTP-UFM with vocabulary size $V = 10$, embedding dimension $d = 10$, and $m = 95$ training samples whose (centered) support matrix $\widetilde{S}$ is shown in Fig. 7-(c). We set $\hat{\pi}_j = 1/m$ and for this fixed support matrix $S$, we generate random soft labels $\hat{p}_j \in \Delta^{V-1}$ such that $\hat{p}_{j,v} = 0, v \notin \mathcal{S}_j$.

We display the training evolution in Fig. 8. We train NTP-UFM with SGD and small weight decay $\lambda = 10^{-5}$ until reaching the empirical entropy lower-bound $\mathcal{H} \approx 1.04$ within an order of $10^{-4}$ as shown in Fig. 8-(a). This decrease in the loss is accompanied by a consistent increase in the parameter norms, as displayed in Panel (b). Despite the increase in norms, the projection of $L_k$ onto the subspace $\mathcal{F}$ remains close to the finite component $L^{\mathrm{in}}$ specified by Thm. 2. This is illustrated in Panel (c) and ensures the recovery of the soft-labels on the in-support tokens.

In Fig. 7, we visualize the logits $L_k = W_k H_k$, and cosine similarities $\mathrm{CORR}(W_k^\top)$ and $\mathrm{CORR}(H_k)$ of word and context embeddings, respectively. As shown in Panel (a) of Fig. 7, as training continues the parameters recover the implicit geometry in Panel (b), which visualizes the prediction made by Thm. 1 and Cor. 1. In Panel (c), we also visualize the cosine similarity of $\widetilde{S}$ and $\widetilde{S}^\top$, which we introduced as a less expensive proxy of $W^{\mathrm{mm}}$ and $H^{\mathrm{mm}}$ (see App. C.4). While the theory and proxy are not exactly the same, their heatmaps display similar structure. To verify the close relationship between the embeddings and the proxies quantitatively, we measure $\mathrm{SIM}(H, \widetilde{S})$ and $\mathrm{SIM}(W^\top, \widetilde{S}^\top)$, where $\mathrm{SIM}(X, Y)$ measures the structural correlation between the two matrices (see Sec. 5). Fig. 8-(d) confirms the high correlation between the proxy **(P)** and the embeddings' implicit geometry. We also note that in the special case where the support sets of two contexts are identical, the heatmaps also verify the subspace collapse Claim **(C3)**: Along the diagonal blocks of $\mathrm{CORR}(H)$, where the contexts with similar support sets lie, the samples have maximum correlation and are aligned.

We finally note that in this experiment $L^{\mathrm{mm}} = \widetilde{S}$ (we verified that the dual-certificate condition in Prop. 5 holds).

### D.2  Additional details and results: Sec. 5

#### D.2.1  Datasets

We use a total of three datasets. We first employ two smaller-scale datasets, which are well-suited for verifying our theoretical solutions. Then, we use one larger-scale dataset designed to investigate the geometric properties of text in conditions that approximate real-world text scenarios.

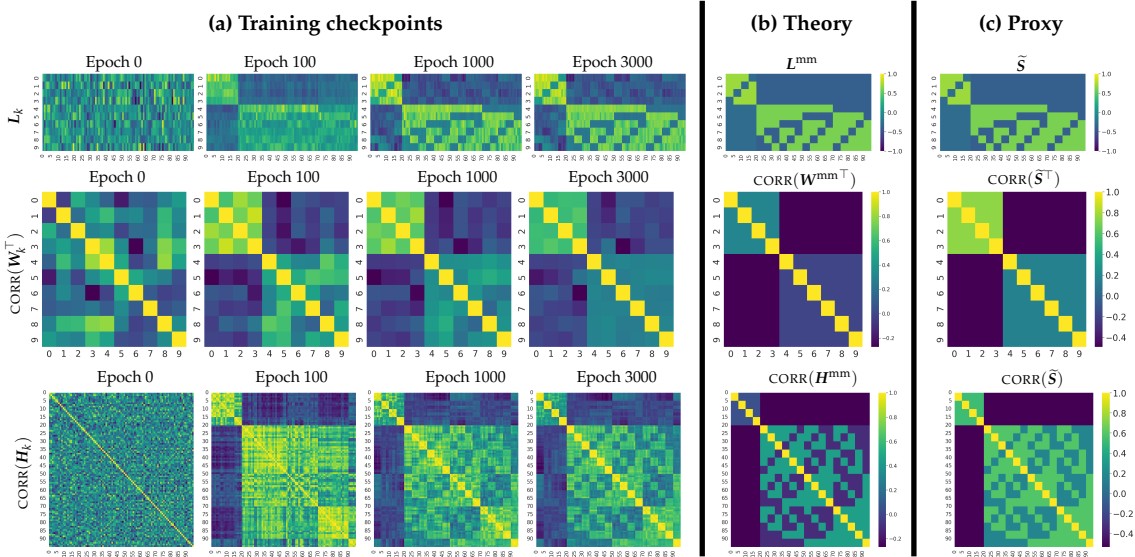

**Figure 7:** Evolution of NTP-UFM parameters $L_k$, $\mathrm{CORR}(W_k^\top)$ and $\mathrm{CORR}(H_k)$ when training close to convergence to the empirical entropy $\mathcal{H}$ (See Fig. 8). At the end of the training, the parameters align with the prediction of Thm. 1 ((a) vs (b)). Additionally, the correlation patterns between the embeddings closely follow the similarities between the support sets ((a) vs (c)). See App. D.1 for details.

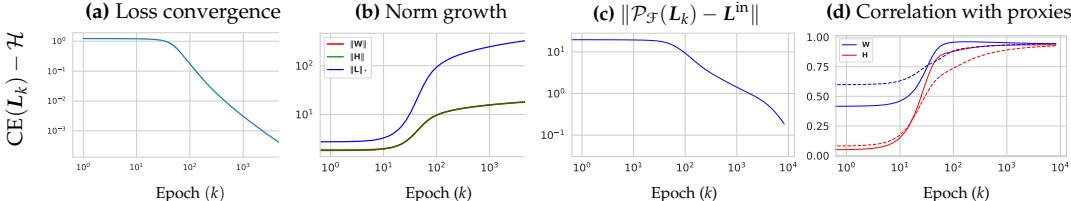

**Figure 8:** Numerical experiments on NTP-UFM. **(a):** CE converges closely to the empirical entropy $\mathcal{H}$, **(b):** Norms of the parameters grow, **(c):** While the parameters converge directionally to $L^{\mathrm{mm}}$ (Fig. 7), the projection of $L_k$ on $\mathcal{F}$ converges to the finite component $L^{\mathrm{in}}$ specified by the soft-labels $\hat{p}_j$, **(d):** Structural correlation between the learned embeddings and the proxy, i.e., $\mathrm{SIM}(H, \widetilde{S})$, $\mathrm{SIM}(W^\top, \widetilde{S}^\top)$. The correlation with $H^{\mathrm{mm}}$ and $W^{\mathrm{mm}\top}$ (instead of $\widetilde{S}$, $\widetilde{S}^\top$) is displayed with dashed lines for reference.

Synthetic. We generate each context $\bar{x}_j$ as follows: We select $T - 1 = 5$ words (tokens), and manually come up with the $T = 6$-th tokens that are consistent with the given context. To model the probabilities $\hat{\pi}_j$ and $\hat{p}_j$, we sample each support independently to a number of repeats to emulate the behavior of repeated context in natural language. For instance, the context $\bar{x}_j =$ "Lily wants to try the" is followed by $\mathcal{S} = \{\text{soup}, \text{game}, \text{movie}\}$ with respective probabilities $[0.5, 0.25, 0.25]$. The dataset consists of $n = 116$ samples, containing $m = 16$ distinct contexts, and a vocabulary size of $V = 30$. Each context has a fixed support set length of $|\mathcal{S}_j| = 3$. In Fig. 9, we show the $m = 16$ unique contexts and the soft labels on their next token. In this dataset, the tokenization is done at the word level and the empirical entropy lower-bound is $\mathcal{H} = 1.6597$.

Simplified TinyStories. Advancing towards a more realistic but still controlled dataset, we use contexts and support sets from the TinyStories corpus. We derive contexts $\bar{x}_j$ and support sets $\mathcal{S}_j$ from the TinyStories dataset and we create the training data from the most frequent word-level contexts with length 5, such as $\bar{x}_j = [\text{"once"}, \text{"upon"}, \text{"a"}, \text{"time"}, \text{","}]$. For the support sets $\mathcal{S}_j$, we record all next tokens of $\bar{x}_j$ in the original dataset. Then, we replace the words in the contexts with their synonyms to generate new contexts with identical support sets. This allows us to have multiple contexts sharing common support sets, while controlling the vocabulary size, which is set to $V = 104$. The final dataset consists

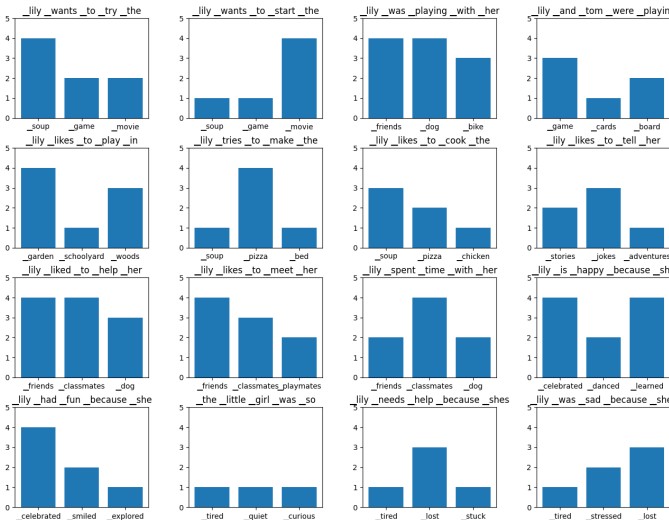

**Figure 9:** The contexts, support sets, and soft labels in the Synthetic dataset.

of $n \approx 3050$ samples with $m \approx 400$ unique contexts. The frequencies $\hat{p}_j$ are determined by independently sampling each next token from $\mathcal{S}_j$ several times. Here, $\mathcal{H} = 1.0166$.

TinyStories. To experiment with more a standard dataset, we use 100 stories sampled from TinyStories. Here, we do not make any sample selections for contexts and support sets, unlike the previously mentioned datasets. We use a tokenizer with vocabulary size $V = 128$, and fixed context length $T - 1 = 6$. We choose a small context window and vocabulary size to make tracking the distinct contexts and their support sets computationally manageable. In this setup, the number of distinct contexts $m \sim 10^5$ and $\mathcal{H} = 0.3112$.

### D.2.2 Training details

**Synthetic and TinyStories experiemtns.** We train the models long enough to ensure that the loss approached the empirical entropy lower-bound $\mathcal{H}$ within an order of $10^{-4}$. In all experiments, we use AdamW optimizer with a weight decay $\lambda = 10^{-6}$. The learning rate is initially set to $10^{-4}$, with a step-decay schedule.

**TinyStories experiemtns.** We train TF with AdamW optimizer until it reaches the empirical entropy lower-bound within an order $10^{-3}$. We use warm-up over the first 5 epochs to increase the learning rate to $5 \times 10^{-4}$ and use a cosine learning rate scheduler to decay its values over the course of the training. We set the weight decay to $10^{-5}$.

### D.2.3 Additional results

Figs. 10-17 complement the discussions and experimental results in Sec. 5. Below we only discuss the missing details of the TinyStories experiments.

**Visualization details of TinyStories experiments.** For visualizing the heatmap of embeddings similarity in Fig. 2, we choose 10 unique support sets in the dataset with $|\mathcal{S}_j| > 2$. For each of the chosen support sets $\mathcal{S}$, we choose 10 distinct contexts $j$ such that $\mathcal{S}_j = \mathcal{S}$ (a total of 100 samples), and illustrate the Gram matrix of the centered support matrix $\tilde{S}$ and context embeddings $H$ for this subset of the contexts. Given that the support sets are sparse, we choose to use contexts with the largest support set size which would result in a more pronounced pattern on the correlation matrices. We skip visualizing the word embeddings $W$ in this experiment, as the heatmaps did not display any noticeable visual structures.

For the metrics displayed in Fig. 11, due to the larger number of context embeddings, we only compute the metrics such as $\|\mathcal{P}_{\mathcal{F}}(L_k) - L^{\text{in}}\|$, norm growth and correlation with proxies using a sample of 1000 token/context pairs chosen randomly from the training set. We observe qualitatively similar behavior compared to other experiments, with the exception of the norm growth of $H$, which we conjecture to be a consequence of layer normalization in the final embedding layer.

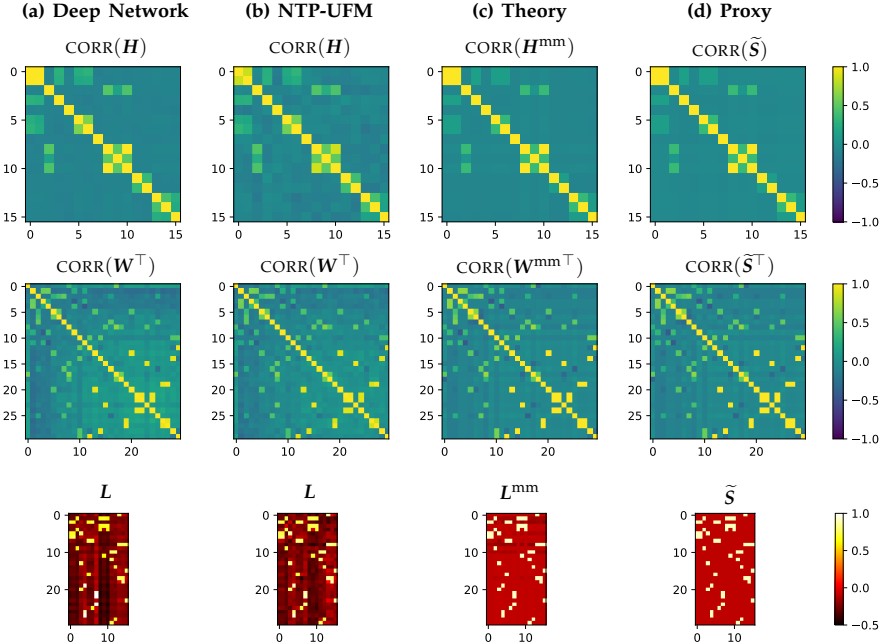

**Figure 10:** Same as Fig. 1, this time for the Synthetic dataset.

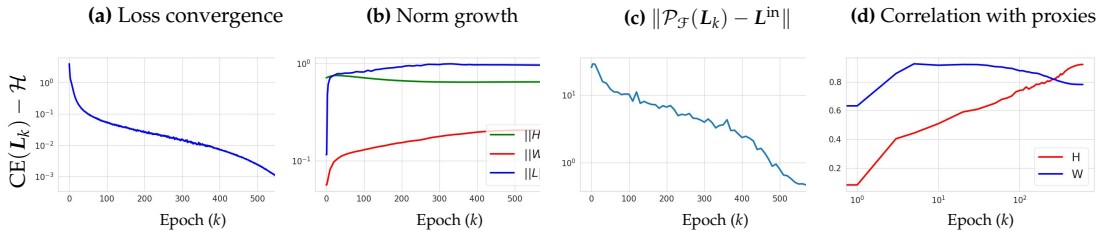

**Figure 11:** Similar to Fig. 7, this time on a 12-layer TF trained on a subset of 100 stories from the `TinyStories` dataset.

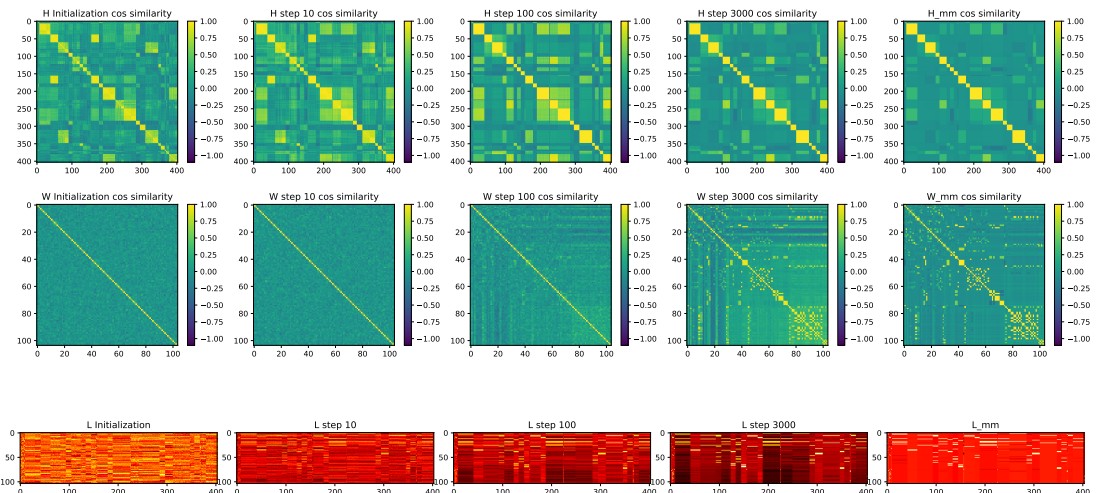

**Figure 12:** Evolution of $H_k$, $W_k$ and $L_k$ to NTP-SVM$_\star$ solution $H^{\mathrm{mm}}$, $W^{\mathrm{mm}}$ and $L^{\mathrm{mm}}$ throughout training. The embeddings are trained by the transformer on the `Simplified TinyStories` dataset. See Sec. 5.

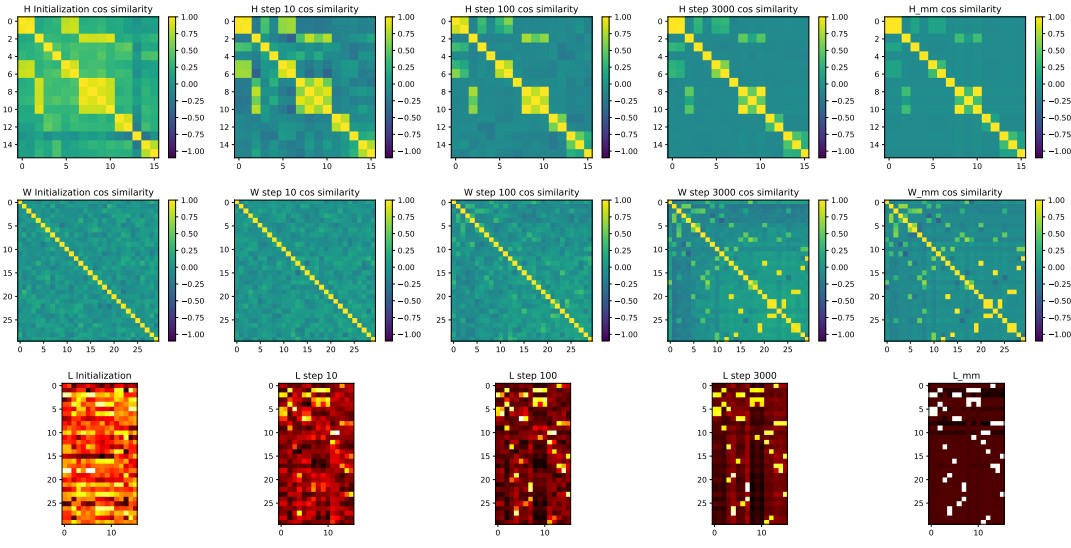

**Figure 13:** Same as Fig. 12, this time for the `Synthetic` dataset. See Sec. 5.

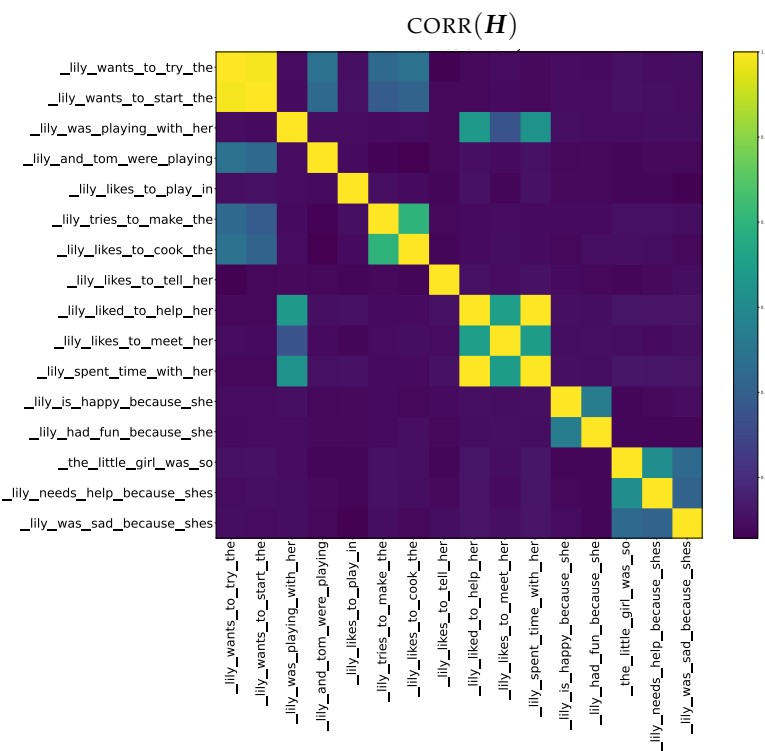

**Figure 14:** Geometry of context embeddings for Synthetic dataset. A lighter color indicates higher similarity in the embedding space.

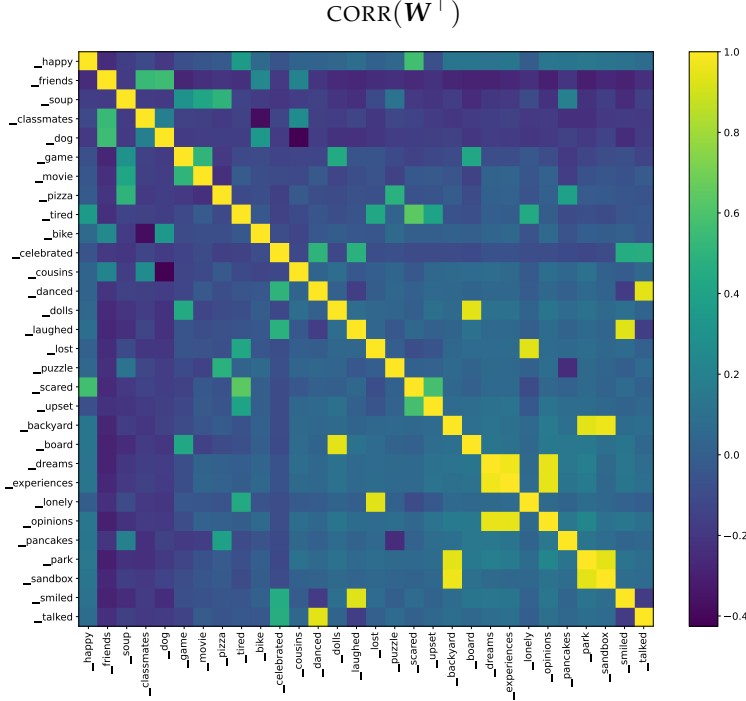

**Figure 15:** Geometry of word embeddings for Synthetic dataset. A lighter color indicates higher similarity in the embedding space.

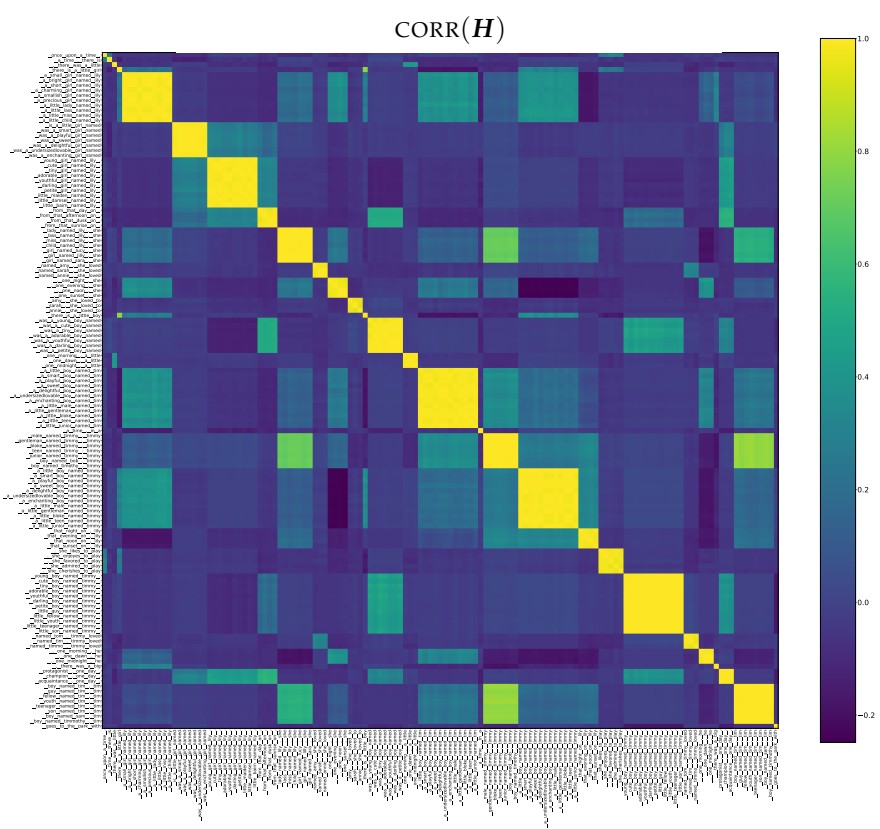

**Figure 16:** Geometry of context embeddings for `Simplified TinyStories` dataset. A lighter color indicates higher similarity in the embedding space.

$$\text{CORR}(\boldsymbol{W}^\top)$$

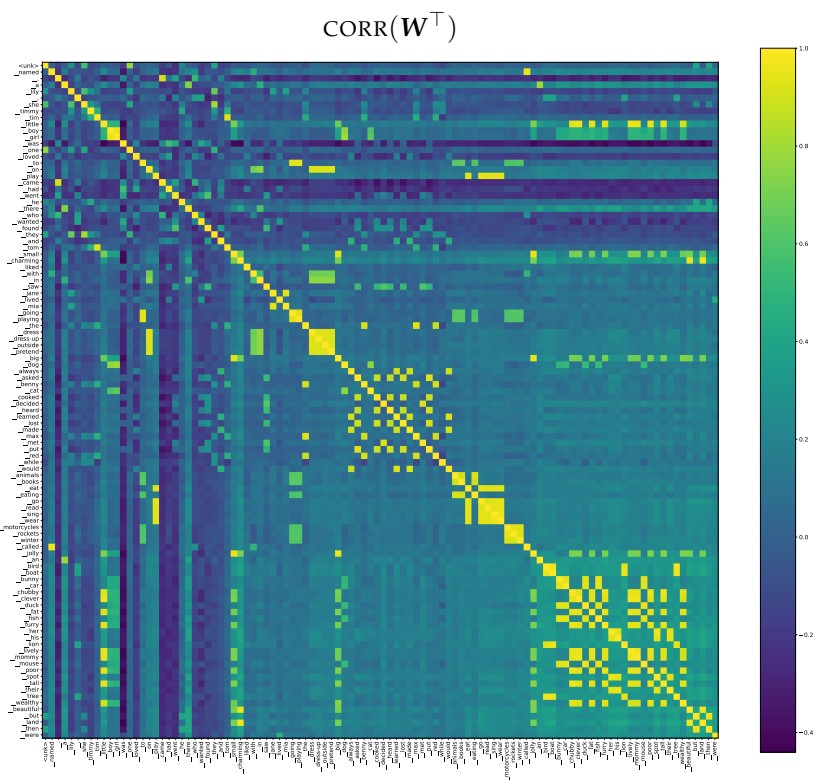

**Figure 17:** Geometry of word embeddings for `Simplified TinyStories` dataset. A lighter color indicates higher similarity in the embedding space.

### D.3 Other architectures

The results in Thms. 1 and 2 hold provided the model is expressive enough to generate (approximately) unconstrained embeddings and the loss can be minimized close to the lower-bound. Depending on the expressiveness of a specific network design, the size of the network required for achieving the entropy lower-bound can vary significantly.

To explore this, we hereby repeat our experiments on `Simplified TinyStories` by replacing the TF model with a multi-layer perceptron (MLP). We use an MLP consisting of 20 layers organized into four blocks, each containing five layers, with hidden dimension sizes 1024, 512, 256, and 128. This leads to a network with around 8 times more parameters than the TF model. The final embeddings geometry is depicted in Fig. 18. We observe that the geometrical patterns appearing in $\text{CORR}(\boldsymbol{H})$ and $\text{CORR}(\boldsymbol{W}^\top)$ at a coarse level are similar to those of the TF model. However, the MLP, even with $\sim 10$ times larger size, struggles to recover the fine-grained patterns. In terms of loss convergence, we find the MLP converges to the empirical entropy in the order of $10^{-2}$ but TF converges in the order of $10^{-4}$ on the same dataset. We conjecture that an even larger MLP might be able to achieve better convergence to our theoretical prediction, but we also caution of potential optimization bottlenecks. We encourage additional experiments with other architectures such as LSTMs or state-space models as future work.

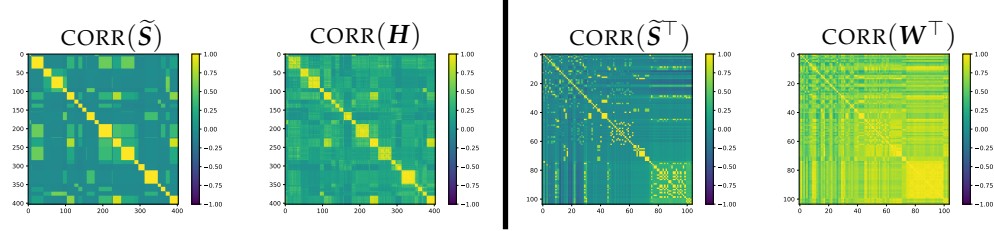

**Figure 18:** Embedding geometries trained by MLP. **Left:** Geometry comparison between $\text{CORR}(\widetilde{\boldsymbol{S}})$ and $\text{CORR}(\boldsymbol{H})$; **Right:** Geometry comparison between $\text{CORR}(\widetilde{\boldsymbol{S}}^\top)$ and $\text{CORR}(\boldsymbol{W}^\top)$. See App. D.3 for details.

### D.4 Auto-regressive training

Here, we train the model auto-regressively. In the previous settings, we focused on training samples that were all of fixed length $T - 1 = 5$. In this section, we let the model learn the embeddings for different sequence lengths $1 \le T - 1 \le 16$. Note that the length of the context does not have an impact on our theoretical analysis other than affecting the sparsity pattern $\boldsymbol{S}$ of the training set which in turn influences the implicit geometry as we have seen.

We train an 8-layer TF on 200 stories from `TinyStories` using character-level tokenizer, which limits the vocabulary to approximately 40 characters and promotes higher entropy in next-token distribution. We denote the loss value across different positions $T \in \{2, \cdots, 17\}$ at iteration $k$ by $\text{CE}(\boldsymbol{L}_{T,k})$. In Fig. 20, we display the distance of each loss component $\text{CE}(\boldsymbol{L}_{T,k})$ from its empirical entropy lower-bound $\mathcal{H}_T$, i.e., the T-gram entropy of the training set. We observe generally better convergence for shorter sequence lengths $T$. In Fig. 19, we illustrate the context embeddings similarities. For visualization, we select 5 contexts per each sequence length that end in token "y_" or "_t". The context embeddings and the proxy **(P)** show similar patterns at a coarse level. However, at finer scale, the learned context embeddings that end with the same tokens exhibit strong correlation on average, even when their support sets do not align. We defer further investigation into whether the embeddings' dependence is due to insufficient network capacity or an optimization bottleneck to future work.

### D.5 Discussion on large vocabulary setting

Throughout our analysis, we require the embedding dimension to be larger than the vocabulary size, i.e., $d \ge V$. This condition was necessary to make the non-convex problem in (8) convex. However, if there exists a low-rank optimal solution in NTP-SVM$_\star$, i.e., $\text{rank}(\boldsymbol{L}^{\text{mm}}) < V$, the condition $d \ge V$ can be relaxed in the analysis. In general, the smallest

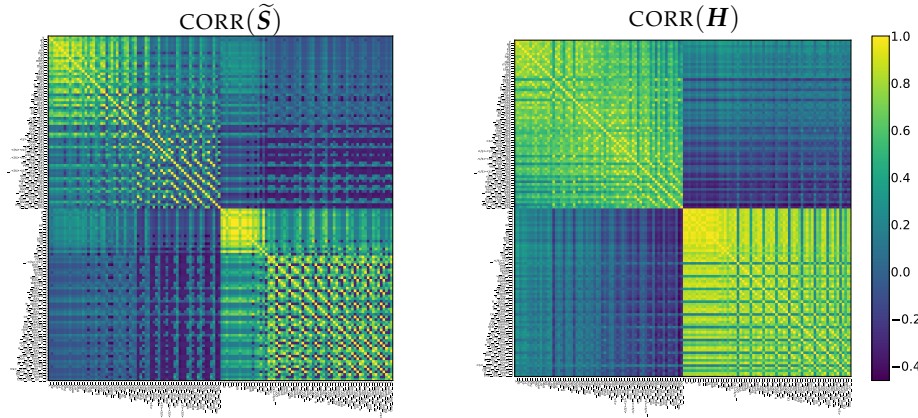

**Figure 19:** Geometry of context embeddings $\text{CORR}(\boldsymbol{H})$ and the heuristic proxy **(P)**, $\text{CORR}(\widetilde{\boldsymbol{S}})$, in the autoregressive experiment of App. D.4.

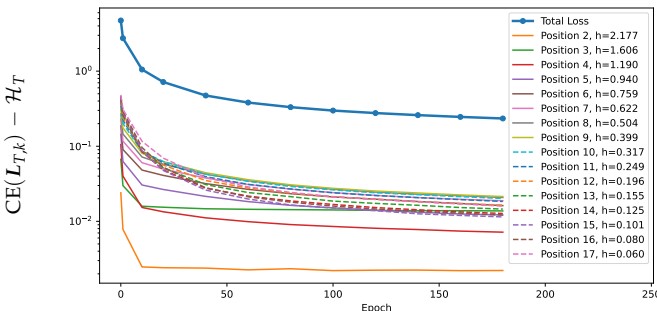

**Figure 20:** TF trained autoregressively on a subset of `TinyStories` dataset: Loss convergence to its empirical entropy lower-bound for each sequence length $T = 2, \cdots, 17$. See App. D.4 for details.

rank among all minimizers depends on the sparsity pattern of the language. It is intriguing to investigate this dependence further.

This section examines the network's performance when the decoder dimension $d$ is smaller than the vocabulary size $V$. To ensure a fair comparison between different setups, we fix the transformer blocks' inner embedding dimension $d_{\text{TF}} = 64$ to maintain comparable expressivity for all networks. To vary the final layer embedding dimension $d$, we add an additional linear layer on top of the network to adjust the context embedding dimension.

In Fig. 21, we show the learned context embeddings for $d = 128$ and $d = 64$, for a 10-layer TF trained on the `Simplified TinyStories` dataset with $V = 104$. We observe that with smaller $d$, the speed of convergence to the entropy lower-bound $\mathcal{H}$ decreases. However, the learned context embeddings exhibit relatively similar geometry with moderate values of $d < V$. We leave more theoretical and empirical exploration of this setting to future works.

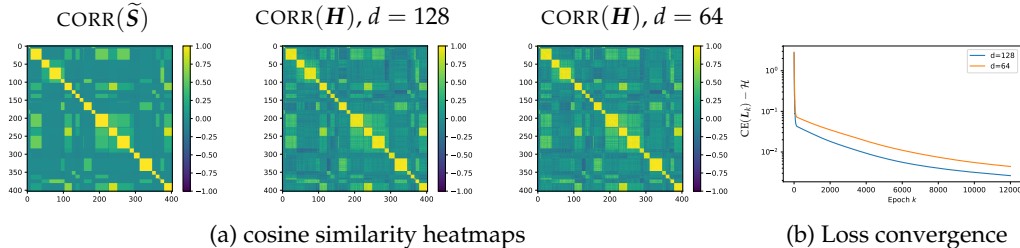

(a) cosine similarity heatmaps            (b) Loss convergence

**Figure 21:** Varying embedding dimension $d$ in the `Simplified TinyStories` experiment. **(a)** Left: Proxy **(P)** for context embeddings. Middle and Right: Geometry of context embeddings trained with TF with $d = 128 > V$ and $d = 64 < V$, respectively, **(b)** Loss convergence to the empirical entropy lower bound $\mathcal{H}$. See App. D.5 for details.

