# OpenReview forum: "Implicit Geometry of Next-token Prediction: From Language Sparsity Patterns to Model Representations"
_colmweb.org/COLM/2024/Conference — COLM_

### Official Review · Reviewer_AKL7 · 2024-05-10

**Rating:** 6
**Confidence:** 3
**Ethics Flag:** 1

**Summary:**

The authors provide an analysis of the geometry of word and context embeddings in language models learned with next-token prediction.  Specifically they model the learning problem as a soft-label classification of the next token, and assume that context embeddings can be learned in an unrestricted way (i.e. not constrained to the parametric form imposed by the model).  This leads to an interpretation of the problem as a nuclear-norm regularized optimization, similar to support vector machines, in which the geometry (similarity matrices) of the learned word and context embeddings align with that of the support sets of the corresponding contexts.  This is a nice and intuitive result, and it is also demonstrated empirically (qualitatively) on a synthetic toy dataset and on the Tiny Stories dataset.

Quality:  This is generally high-quality, well-motivated work.  The empirical work is lower-quality, in the sense that the results are qualitative and limited to the restrictive setting where the model dimensionality is higher than the vocabulary size.

Clarity:  The paper is generally very clear.

Originality:  As far as I know, this is original work.

Significance:  The significance is limited by the restrictive assumptions and limited empirical work in the paper.  Like many papers on the science of LMs, it is not clear what the implications of the work on how practitioners train or use LMs might be.  Although an analysis with more obvious implications would have higher significance, I think this purely scientific analysis is significant enough.

**Questions To Authors:**

- There seems to be a typo in the definition of $f_{\theta'}$, as its output is not in $\mathcal{V}$ as stated (Section 2, first paragraph).

- In the definition of CE(\theta'), what is $n$?  I believe there is a missing subscript $i$ on $x$, and $S_{z_i}$ seems to be undefined (Section 2, first paragraph).

- Spell out "ETF" (Section 3.4.1).

- I believe the reference to Harris 1954 should be Firth, J. R., Studies in Linguistic Analysis, 1957. (Section 4.1)

- I believe the finding that "Contexts with a larger intersection in the suppost set ... exhibit larger correlation" (Section 4.2) is a qualitative statement based on visual inspection of Fig. 3. Is that correct?  If so, then that should be stated explicitly.  The mention of "larger correlation" suggests that a correlation was actually measured.

- There are a couple of missing citations:
  * "the linear relationships underlying word analogies" (Section 5)
  * "recent promising studies ... that leverage the neural-collapse geometry" (Section 6)

- Typos, grammar, etc.:
  * "their synonymous" --> "their synonyms".  How did you derive the synonyms?  (Section 4.2)
  * "Different to them" --> "Unlike them" (Section 5)

**Reasons To Accept:**

+ The idea is well-motivated and the paper is generally well-written.
+ The results, while preliminary, suggest that the analysis approach is promising.

**Reasons To Reject:**

- The findings are somewhat anecdotal, based on toy settings and qualitative results
- The assumption d > V (the hidden dimensionality is larger than the vocabulary) is very unrealistic, and it's not clear how it might be relaxed in future work (the authors don't address this)

---

> ### Author Rebuttal · Authors · 2024-05-31
>
> Thanks for your overall positive feedback. In the final version of the paper, we will make sure to address the clarity issues and typos pointed out. Thank you!
>
> **d>V**:  Unlike static models, contextualized models involve two levels of compression: mapping 1) V words, and 2) m contexts (sequences of words) to d-dim space. Despite the requirement d>V, we allow the number of contexts to be much larger, i.e. m>>d. So, from the perspective of contexts, it remains interesting and nontrivial to formalize how the model compresses them into a lower dimensional space. Besides, it is not a priori clear to us, even under the assumption of d>V, how the geometry depends on the language statistics as encoded in the sparsity pattern.  That said, we provide preliminary results for d<V at https://ibb.co/Yb9S5Tr as motivation for future work.
>
> We also remark that the rank-constrained logit relaxation (see Lem1) holds for any d. For example, this directly yields a necessary and sufficient condition for the loss to reach the entropy lower bound, which is the feasibility of the rank-constrained NTP-SVM problem. We also hypothesize that the sparsity pattern of language promotes low-rank solutions to the NTP-SVM problem which could allow extensions of the parameter convergence results to the setting d<V.
>
> **Visual inspections**: Thank you for commenting on that. Following your suggestion, we now verify the correlation quantitatively as well by computing the normalized covariance of the heatmaps. The plot of this metric vs epoch is available at https://ibb.co/Yb9S5Tr . We hope this addresses your concern and thanks again for the suggestion.

---

> > ### Comment · Reviewer_AKL7 · 2024-06-05
> >
> > Thank you for the responses!  The additional explanation and results do help.  If the paper is accepted, I hope that you can add the additional results.  That being said, the experimental settings are still toy-ish, and that more realistic experiments could have been done (if that is not the case, I would be happy to be corrected!).  For this reason I continue to feel that the paper is only marginally above the acceptance threshold.

---

> > > ### Author Response · Authors · 2024-06-05
> > >
> > > Thank you for your careful read and suggestions, and for endorsing our paper. We appreciate your time.

---

### Official Review · Reviewer_wYxA · 2024-05-10

**Rating:** 4
**Confidence:** 2
**Ethics Flag:** 1

**Summary:**

The paper sets up the autoregressive nature of language models as a prediction task from complete contexts (a very sparse huge set) to words in the vocabulary (~only the one-hot case of gold data is discussed~ including probabilistic targets) and characterizes that prediction through a series of assumption-driven decompositions into matrices and objectives.
I found the paper very hard to follow, though that may be due to my lack of background in the area, but as the reasons to reject below show, even so I think there are definitive weaknesses standing in the way of publication at this conference.

**Questions To Authors:**

- please use parentheses around citations where appropriate (\citep{})
- typo: empiriical -> empirical

**Reasons To Accept:**

The clear stating of assumptions and rigorous treatment are appreciated, as is the extensive appendix.

**Reasons To Reject:**

- The focus on d > V - 1 makes the entire analysis not fit the standard language modeling setup where d <<< V - 1 (in fact the whole motivation for neural modeling)
- Notation used is not just explained but even just introduced in the appendix
- Relevance to practical language modeling and potential for useful insights derived from the described characterization are not given

---

> ### Author Rebuttal · Authors · 2024-05-31
>
> Thanks for your review. Before addressing your concerns:
> 1) We are puzzled by your comment “only the one-hot case…”. In fact, we explicitly account for the fact that multiple tokens (labels) can follow a given sequence of tokens (context) in the training corpus which results in a soft-label setup.
> 2) Allow us to clarify our contributions: Our high-level goal is to identify how the information from the training corpus gets encoded in language models. This is influenced by several factors: 1) optimization objective, 2) model architecture, and 3) model/data scale. We decouple these factors by focusing on first one: interactions of context/word embeddings $H,W$ in the objective at optimality. We find them as matrix factorization of a logit that decomposes to a finite PMI-like matrix $L_{in}$ and a directional component $L_{mm}$, which is driven by the sparsity pattern of the conditional probability of next tokens. Overall, this a) gives a new perspective to classical static word embedding results such as [LG14], b) connects contextualized embeddings to the sparsity pattern of language, c) establishes conditions for NTP loss reaching lower-bound, d) confirms the log-bilinear NTP-UFM as a good analytical proxy for complex models, e.g. deep transformers.
>
> **d>V**:  Unlike static models, contextualized models involve two levels of compression: mapping 1) V words, and 2) m contexts (sequences of words) to d-dim space. Despite the requirement d>V, we allow for the number of contexts to be much larger, i.e. m>>d. So, from the perspective of contexts, it remains nontrivial to formalize how the model compresses them into a lower dimensional space. That said, we provide preliminary results for d<V at https://ibb.co/Yb9S5Tr as motivation for future work.
>
> **Practical insights**: We believe that our perspective opens avenues to counteract the negative impacts of biases and extreme imbalances specific to language. Better understanding of the context/tokens distributions and how they correlate with the model embeddings may help combat unfairness in large language models. For example, we envision using the geometry of embeddings to reverse engineer the design of more robust loss or to devise posthoc robust sampling methods that account for imbalances in the token/context distributions of language data. Our work is the first step in making these connections more concrete but further investigations are needed to fully achieve these practical goals.

---

> > ### Comment · Reviewer_wYxA · 2024-06-05
> >
> > Re 1: Oh, you're right, I must have misunderstood that!
> >
> > Re d>V: Additional results on this seen to a proper conclusion like in the submitted paper may raise this to be a more worthwhile contribution, but I don't think that would be ready in time for a camera-ready...
> >
> > With the clarified practical insights, I am raising my score slightly.

---

> > > ### Author Response · Authors · 2024-06-05
> > >
> > > Thank you for acknowledging our response.
> > >
> > > We are glad that the misunderstanding about our main contribution—explicitly accounting for the fact that multiple tokens (labels) can follow a given sequence of tokens (context) in the training corpus, resulting in a **sparse** soft-label setup—has been addressed.
> > >
> > > We reiterate that combining this viewpoint with unconstrained features offers a new perspective in studying word/context embeddings. Given the originality and timeliness (word embedding geometry studies date back to the word2vec era and have recently surged in the context of LLMs), we believe our contribution is valuable to the community.
> > >
> > > Additionally, beyond the motivational figures on embedding geometry, some of our theoretical analysis already applies to the regime d<V. For instance, in Appendix C.3.1 (see definition 4 and in the paragraph below), we show that the feasibility of NTP-SVM with an additional constraint rank(L)≤d is necessary and sufficient for the model to reach the entropy lower bound. We consider this a fundamental result. Besides, the results remain non-trivial even when d>V. Therefore, we respectfully disagree with the assessment that our work lacks in becoming a “more worthwhile” contribution.

---

### Official Review · Reviewer_HSEc · 2024-05-11

**Rating:** 6
**Confidence:** 1
**Ethics Flag:** 1

**Summary:**

Similar to Levy & Goldberg (2014), who framed the Skip-Gram with Negative Sampling (SGNS) training objective of Word2Vec (Mikolov et al., 2013b;a) as weighted matrix factorization, this work frame next-work prediction (NTP) training as soft-label classification over sparse probabilistic label vectors coupled with an analytical approximation that allows for the unrestricted generation of context embeddings by the language model. This combination establishes a link between NTP training and rank-constrained, nuclear-norm regularized optimization in the logit domain, offering a framework to analyze the embeddings’ geometry.

This framework allows the examination of how the embeddings’ geometry varies with the distributional properties of the training data, connecting geometric features to textual structures. The authors validate their theoretical insights through experiments on synthetic and
small-scale real language datasets

I think this paper leans a bit towards theory, which is somewhat outside my area of expertise. Therefore, please take my review with a grain of salt.

**Questions To Authors:**

Please see above.

**Reasons To Accept:**

I think this work can deepen our understanding of why the next-word prediction objective works so well.

**Reasons To Reject:**

This is not really a reason to reject the paper, but could you briefly summarize your work for a researcher who mainly works in empirical NLP rather than in the ML theory domain? Additionally, I am curious about the potential practical implications and applications of this work.

---

> ### Author Rebuttal · Authors · 2024-05-31
>
> Thanks for your positive feedback. As a matter of fact, your summary very well captures the message of our work!
>
> Repeating here for completeness: Our high-level goal is to identify how the information from the training corpus gets encoded in prediction-based language models. This is influenced by several factors: 1) optimization objective, 2) model architecture, and 3) scale of model and data. We decouple these factors by focusing on the first one: we study the interactions of context embeddings $H$ and word embeddings $W$ in the objective at optimality. We find that they occur as matrix factorization of a logit matrix that decomposes to a finite PMI-like matrix $L_{in}$ and a directional component $L_{mm}$. This decomposition is driven by the sparsity pattern of the conditional probability distributions of next tokens.
>
> Overall, this a) gives a fresh perspective to classical matrix factorization results for static word embeddings such as [LG14], b) establishes a connection of the contextualized embeddings to the sparsity pattern of language, c) establishes conditions under which the NTP training loss reaches its lower-bound, d) confirms the log-bilinear model (NTP-UFM) is a good analytical proxy for more complex prediction-based models, e.g. deep transformers.
>
>
> Looking ahead we believe that capturing such influences of how the geometry depends on language statistics such as the conditional probability distributions of next tokens, opens avenues to counteract the negative impacts of biases and extreme imbalances in the training set, as text data is known to follow a heavily long-tailed distribution by Zipf's law. Better understanding of the distribution of context/tokens and how they correlate with the model embeddings may help combat unfairness in large language models. For example, we envision using the geometry of embedding information to reverse engineer the design of more robust loss or to devise posthoc robust sampling methods that account for imbalances in the token/context distributions of language data. We acknowledge that further investigations are needed to fully achieve this goal. However, we consider our framework a first step in making these connections more concrete. Furthermore, we believe these foundations could help develop interpretability metrics for LLMs in order to better analyze their properties.

---

> > ### Comment · Reviewer_HSEc · 2024-06-04
> >
> > This explanation looks good to me. Thank you for your response!

---

> > > ### Author Response · Authors · 2024-06-04
> > >
> > > Thank you very much for acknowledging our response. If any more questions arise, we are happy to discuss!

---

### Official Review · Reviewer_4aLd · 2024-05-15

**Rating:** 7
**Confidence:** 2
**Ethics Flag:** 1

**Summary:**

This paper aims to understand the geometric properties of word/context embeddings learned in the process of next-token-prediction in training large language models. To do this, the paper designs a formulation of the next-token-prediction task, that mimics the original model but provides a framework for analyzing the geometry of the embeddings. Through this framework the authors of this work are able to draw specific conclusions about the embedding's geometry. Experiments conducted on two (one synthetic, one real, small-scale) datasets to validate the theories from this paper.

**Reasons To Accept:**

How this work benefits the community:

* This paper clearly has some interesting ideas on a methodology for analyzing word embeddings, which would be valuable to the community.

* The paper provides strong theoretical support for claims and hypotheses, that would be very interesting to researchers working on analyzing large language models.

* Based on the observations and learnings from this work, potentially impactful research directions are outlined for the audience.

**Reasons To Reject:**

How this work could be further strengthened:

* This paper would greatly benefit from more clarity in its exposition of the work. There is a *lot* packed into this work. Much of it is better understood by referring to details in the appendix. As such, this paper appears to be more like a long journal article -- which has been modified to fit this format. As a result, much of the interesting and important content is split out into the appendix.

* While the focus of the paper has been clearly stated as being interested in uncovering the geometric properties of word embeddings, the motivation/benefit of this is not explicitly spelled out. It is unclear, at the outset, why and how the geometric properties of embeddings could help further research in this field.

---

> ### Author Rebuttal · Authors · 2024-05-31
>
> We thank you for your positive feedback. We would leverage the extra page of the camera-ready to address your concern about the density of information in the current form. We will also include a clearer statement on the motivation/implications, as briefly discussed below.
>
> Our high-level objective is to understand what information from the training corpus gets encoded in the embedding space of contexts and words. In other words, which structures or statistics from the training set influence the similarity and arrangements of embeddings and how this correlates with the semantic similarity of words and contexts. This is influenced by several factors: 1) optimization objective, 2) model architecture, and 3) scale of model and data. We decouple these factors by focusing on the first one: we study the interactions of context embeddings $H$ and word embeddings $W$ in the objective at optimality.
>
> We find that the statistical properties that are important are encoded in the sparsity patterns of the conditional probability of the next tokens. For example, this is reflected in our experiments by the similarities between the Gram matrices of word/context embeddings and $SS^T, S^TS$. Conceptually, these findings relate prediction-based models that operate on contextualized word embeddings to models based on co-occurrence statistics, much like the classical connection of word2vec, which operated on static word embeddings, to count-based models [LG14]. The latter classical study motivated and resulted in improved models such as GloVe. We anticipate that our work provides analogous opportunities for contextualized word embedding models.
>
> We also believe that such connections can open avenues to counteract the negative impacts of biases and extreme imbalances in the training set, as text data is known to follow a heavily long-tailed distribution by Zipf's law. Better understanding of the distribution of context/tokens and how they correlate with the model embeddings may help combat unfairness in large language models. We acknowledge that further investigations are needed to fully achieve this goal. However, we consider our framework a first step in making these connections more concrete. Furthermore, we believe these foundations could help develop interpretability metrics for LLMs in order to better analyze their properties.

---

> > ### Comment · Reviewer_4aLd · 2024-06-05
> > **Thank you for the additional insights**
> >
> > These follow up comments help give me a better understanding of your contributions.

---

> > > ### Author Response · Authors · 2024-06-05
> > >
> > > Thank you very much for your positive feedback! We appreciate your time.

---

### Official Review · Reviewer_6jDj · 2024-05-23

**Rating:** 4
**Confidence:** 4
**Ethics Flag:** 1

**Summary:**

This paper studies the implicit bias of next-token prediction (i.e., language modeling) with the goal of characterizing how the contextual and word embeddings will behave. The theoretical characterization relies on tracing the evolution of the logit matrix L of the model, which is |V| x (number of contexts), over the course of gradient descent. In particular, they formulate language modeling as a semidefinite optimization problem, and then they identify that the logit matrix will converge to something that maximizes the margin between words co-occurring in a given context and words excluded from a given context.

The logit matrix L is the product of W, the word embeddings, and H, the contextual embeddings, so this result on L can be translated to describe how W and H behave. The conclusion then is that the embedding for a given context forms an acute angle (i.e. has a positive dot product with) the word embeddings in the context and an obtuse angle (i.e. has a negative dot product with) the word embeddings outside of the context. These theoretical results are confirmed through carefully controlled experiments on TinyStories and synthetic data.

**Questions To Authors:**

1. Is any part of the analysis unique to next-token prediction? I believe it's possible to formulate masked language modeling in the same way, though one would have to be a little bit careful about what to do with the other masked tokens in a sequence when forming a prediction.

2. Can you elaborate on what the technical challenges are of using a sparse high-dimensional label vector? I am not sure how the original regularization path analysis in Ji & Telgarsky would struggle to accommodate the NTP problem. Similarly, can you clarify how this paper is different from (Thrampoulidis, 2024)? It seems that paper did most of the work in formulating the next-token prediction problem and even discussed the same ideas around the regularization path. From my understanding, the key difference is that that paper had fixed $H$ and only trained $W$, whereas this paper seeks to train both together.

3. Is there any insight into word embedding geometry that has not been shown by prior papers? As I see it, this idea that word embeddings are related to the SVD of the co-occurrence matrix is not too surprising (and your formulation essentially describes the logit matrix as a co-occurrence matrix). I would ideally have liked to have seen something more related to the generative capabilities of really large models (hallucinations, in-context learning, etc), so that one could use this framework of analysis to quantify why next-word prediction can cultivate surprising behaviors.

I am an active reviewer and will take your answers into account to update my score! Also, it is entirely possible that the weaknesses I raised are incorrect -- please correct me if that's the case!

**Reasons To Accept:**

1. Understanding the geometry of the contextual and word embeddings is a well-studied area and is generally useful for interpreting and predicting language model's behavior.
2. The paper clearly considers many related works and thoroughly discusses how their results fit in with prior ones. I would, however, suggest that the authors include more works on interpretability in language models.
3. The theory can accommodate arbitrarily complex data and the training of transformers, which are the popular choice for language models these days (there are some caveats, see below).
4. This approach to analyzing next-token prediction is thought-provoking. I found the future work section to be very interesting, and I think that extending these ideas to accommodate those more realistic settings could be valuable.

**Reasons To Reject:**

**Model is unrealistic**

The theoretical results are in an unrealistic setting where the model's embedding dimension $d$ is larger than the vocabulary size $V$. As I understand it, this allows for the logit matrix $L$ to be uniquely factorized into $WH$, which is essential to the conclusion about the embedding geometry. This does not reflect the standard transformer construction -- in fact, the most recently released language model Gemma exhibits a massive gap between vocabulary size and hidden dimension.

I also want to point out how this assumption weakens the conceptual takeaways from the paper. In most cases, we are interested in studying the model's low-dimensional representation to see how it compresses high-dimensional information -- the assumption that $d\geq V$ removes the opportunity to do this. For example, the takeaway from Proposition 1 is that the contextual embedding will have the same (positive) dot product with all words contained in the context (Fig 4). In that case, how can the model even predict the next token correctly? It would assign equal probabilities to all tokens it has seen in this context (agnostic to their ordering), and that makes it substantially weaker than the standard next-token prediction setting. This setting thus does not provide any more insight than past works that have modeled contextual embeddings as averages of word embeddings [1, 2, 3] (note: the authors do cite these works in passing in their related works). One can also see this by observing that $L^{mm}$ places uniform weights on all terms in the vocabulary.

[1] Carl Allen and Timothy Hospedales. Analogies explained: Towards understanding word embeddings. In International Conference on Machine Learning, pp. 223–231. PMLR, 2019.

[2] Tatsunori B Hashimoto, David Alvarez-Melis, and Tommi S Jaakkola. Word embeddings as metric recovery in semantic spaces. Transactions of the Association for Computational Linguistics, 4:273–286, 2016.

[3] Sanjeev Arora, Yuanzhi Li, Yingyu Liang, Tengyu Ma, and Andrej Risteski. A latent variable model approach to pmi-based word embeddings. Transactions of the Association for Computational Linguistics, 4:385–399, 2016.

**Writing and formulation are hard to follow**

The value of building such a complex formalism is predicated on it being comprehensible and easy to extend. The writing in this paper is unfortunately quite difficult to follow. For example, "in-support tokens" are never defined -- I have just assumed them to be tokens that occur within a given context. But they could also feasibly be "all tokens that could follow a given context". This is just one example of the haphazard writing in the paper that made it difficult for me to follow, even though I have experience reading and working with these theoretical topics.

**Minimal insight can be derived from theoretical results**

I have already mentioned the issue with the embedding dimension and the vocabulary size, but I want to raise other issues with deriving insights from this theory.

1. The geometry of the embeddings can only be described when the support matrix (i.e., the set of contexts used for training) contains all possible sequences of length k that can be constructed from the vocabulary. Realistically, if the vocabulary was a standard (maybe even sub-word) one, most of these sequences would be gibberish, containing no semantic or syntactic coherence.

2. The generative model used for language essentially amounts to treating every language model as a n-gram model. While this has some merit, I think it again goes against the ideas we have about how representation learning works. Relatedly, the max-margin result is pretty standard from the regularization path analysis, but max-margin does not shed much insight when considering _generative_ models instead of the usual classifiers.

---

> ### Author Rebuttal · Authors · 2024-05-31
>
> We appreciate your thorough review.
>
> **In-support tokens**: For context j, a token z is in-support if $ p_{j,z}>0 $, i.e. it appears after the context within the dataset. This is important as it relates to a few of your comments. We apologize for the confusion.
>
> **d>V**:  Unlike static models, contextualized models involve two levels of compression: mapping 1) V words, and 2) m contexts (sequences of words) to d-dim space. Despite the requirement d>V, we allow the number of contexts to be much larger, i.e. m>>d. So, from the perspective of contexts, it remains interesting and nontrivial to formalize how the model compresses them into a lower dimensional space. That said, we provide preliminary results for d<V at https://ibb.co/Yb9S5Tr as motivation for future work.
>
> **Fresh perspective**: The classic [LG14], interprets minimizers of word2vec as matrix factorization of PMI matrix. This relies on a subtle assumption: the PMI matrix has well-defined entries, i.e. co-occurrence probability of any word given context is non-zero, typically enforced heuristically by smoothing. Instead, we account explicitly for sparsity which drives our decomposition of L in two orthogonal components: 1) $L_{in}$, a finite component conceptually similar to PMI, 2) $L_{mm}$, a directional component that only depends on the sparsity pattern.
>
> **Correct prediction**: This is subtle but important: Prop1 describes the dominant component $L_{mm}$. But, recall that the logits have a second finite component $L_{in}$ (living on an orthogonal subspace). This is the one that accounts for the conditional probability of in-support tokens ensuring a correct prediction. See Fig5 for numerical verification of convergence to the two components. We will re-emphasize this after Prop1. Thanks!
>
> **Beyond NTP**  Great question. The framework is indeed more general. MLM is an example: it applies directly to simplified setting of masking a single token, but for the practical setting of random masking of several tokens one might need to be more careful to capture all the intricacies. While interesting, we leave this for future work.
>
> **[JT19,T24]**: The basis of our contribution is incorporating the UFM framework within the formalism of [T24]. Technically, training both $W$ and $H$ makes the problem non-convex, but convex relaxation (4) allows us to leverage regularization path ideas from prior works [JT19] back to [RZH03]. We also go beyond [T24] in our empirical evaluations on language data.

---

> > ### Author Response · Authors · 2024-06-06
> >
> > Thank you again for your detailed review and feedback. We appreciate it. We have made every effort to respond to your questions and clarify misunderstandings in terminology (such as the meaning of in-support words) that may have led to misconceptions. We hope you are satisfied with our responses and look forward to your reply. We are also happy to engage in further discussion and answer any questions before the discussion period ends tomorrow.

---

### Author Response · Authors · 2024-06-05

Dear Reviewers,

Thank you for reviewing our submission. We have tried to address all your comments while staying within the character limit.

We hope our responses are clear. If so, we would appreciate if this is reflected to your evaluation.
If you have further questions, we are more than happy to elaborate on our initial responses during the discussion period.

Thank you again.

---

### Decision · Program_Chairs · 2024-07-10

**Decision:**

Accept

**Comment:**

I find the paper to be very interesting. While I mostly agree with Reviewer 6jDj on the weaknesses of the paper, I think we can expect the paper's result being useful for future work. For instance, while now the assumption d > V seems unrealistic, the size of language models (and thus d) is increasing whereas V is unlikely to increase by a lot (in monolingual setting). So when the LM becomes large enough, this assumption will hold true. Given that the theoretical part of deep learning and LMs is in a very early stage, I personally think we should approach the field in a bottom-up manner just like other science areas in their early days; that is, we first enumerate findings even if they may not be immediately useful, hoping that some future work will be able to connect them. Also, given the nature of CoLM, which is a more experimental conference than existing venues in NLP and machine learning, I think there will be sufficient audience who will be interested in this paper.